# Oxidative photocatalysis on membranes triggers non-canonical pyroptosis

Chaiheon Lee [1,2,3,10], Mingyu Park [1,2,10], W. C. Bhashini Wijesinghe[1,10], Seungjin Na [4], Chae Gyu Lee[1,2], Eunhye Hwang [1,2,3], Gwangsu Yoon [1,2], Jeong Kyeong Lee [1,2], Deok-Ho Roh[1,2], Yoon Hee Kwon[3], Jihyeon Yang[3], Sebastian A. Hughes[5,6], James E. Vince [5,6], Jeong Kon Seo[3,7] ✉, Duyoung Min [1,2] ✉ & Tae-Hyuk Kwon [1,2,3,8,9] ✉

Intracellular membranes composing organelles of eukaryotes include membrane proteins playing crucial roles in physiological functions. However, a comprehensive understanding of the cellular responses triggered by intracellular membrane-focused oxidative stress remains elusive. Herein, we report an amphiphilic photocatalyst localised in intracellular membranes to damage membrane proteins oxidatively, resulting in non-canonical pyroptosis. Our developed photocatalysis generates hydroxyl radicals and hydrogen peroxides via water oxidation, which is accelerated under hypoxia. Single-molecule magnetic tweezers reveal that photocatalysis-induced oxidation markedly destabilised membrane protein folding. In cell environment, label-free quantification reveals that oxidative damage occurs primarily in membrane proteins related to protein quality control, thereby aggravating mitochondrial and endoplasmic reticulum stress and inducing lytic cell death. Notably, the photocatalysis activates non-canonical inflammasome caspases, resulting in gasdermin D cleavage to its pore-forming fragment and subsequent pyroptosis. These findings suggest that the oxidation of intracellular membrane proteins triggers non-canonical pyroptosis.

The intracellular membrane in eukaryotes serves as a reaction platform for biochemical processes, such as organelle interactions, signalling, metabolic reactions, and biomolecular productions[1–3]. These physiological reactions are mediated by membrane proteins and lipids[3], which are the functional constituents of the intracellular membranes. However, these components are susceptible to oxidative damage caused by the endo- and exogenous reactive oxygen species (ROS)[4,5]. This oxidative damage to the intracellular membrane can disrupt the organellar function and trigger programmed cell death, which is implicated in the pathogenesis of various diseases[6–8]. Discovering cellular processes in response to oxidative stress on intracellular membranes is essential for devising strategies to control cell death signalling and treat diseases related to membrane protein oxidation. It has been reported that lipids constituting intracellular membranes can be peroxidised by radical propagation reactions through cytosolic iron-induced hydroxyl radical (•OH) generation,

[1]Department of Chemistry, School of Natural Science, Ulsan National Institute of Science and Technology (UNIST), Ulsan, Republic of Korea. [2]X-Dynamic Research Center, UNIST, Ulsan, Republic of Korea. [3]Research Center, O2MEDi inc., Ulsan, Republic of Korea. [4]Research Center for Bioconvergence Analysis, Korea Basic Science Institute, Cheongju, Republic of Korea. [5]The Walter and Eliza Hall Institute of Medical Research, Parkville, VIC, Australia. [6]Department of Medical Biology, University of Melbourne, Parkville, VIC, Australia. [7]UNIST Central Research Facility, UNIST, Ulsan, Republic of Korea. [8]Graduate School of Carbon Neutrality, UNIST, Ulsan, Republic of Korea. [9]Graduate School of Semiconductor Materials and Device Engineering, UNIST, Ulsan, Republic of Korea. [10]These authors contributed equally: Chaiheon Lee, Mingyu Park, W. C. Bhashini Wijesinghe. ✉e-mail: jkseo6998@unist.ac.kr; dymin@unist.ac.kr; kwon90@unist.ac.kr

which eventually causes caspase-independent ferroptosis[9]. However, cell death signalling in response to the intracellular membrane-localised ROS generation and oxidative stress on intracellular membrane proteins have not been fully elucidated.

Oxidative photocatalysis can be a promising approach to control oxidative stress at desired points inside cells spatiotemporally[10]. Recent studies have investigated cellular responses to organelle-targeted oxidative stress and subsequent cell death signalling using the photocatalysts generating ROS at specific organelles[11,12]. However, a photocatalyst capable of simultaneous localisation within the intracellular membranes enclosing organelles (such as the endoplasmic reticulum, Golgi apparatus, mitochondria, vesicles, and nucleus) has not been developed. This limitation hinders the exploration of cellular

responses to membrane protein oxidation within the intracellular membranes. Nevertheless, the development of an intracellular membrane-localised photocatalyst presents a significant challenge, as these photocatalysts should possess high lipophilicity to localise within the intracellular membrane while still being able to penetrate the plasma membrane. Therefore, developing intracellular membrane-focused oxidative photocatalysis is essential for analysing cellular responses to intracellular membrane oxidation.

In this work, we develop an intracellular membrane-localised organic photocatalyst, BTP, a fatty acid-like molecule consisting of hydrophobic linear π-conjugation and a hydrophilic head (Fig. 1a). Owing to its amphiphilic structure, BTP is localised in the intracellular membrane. Furthermore, its photocatalysis oxidises

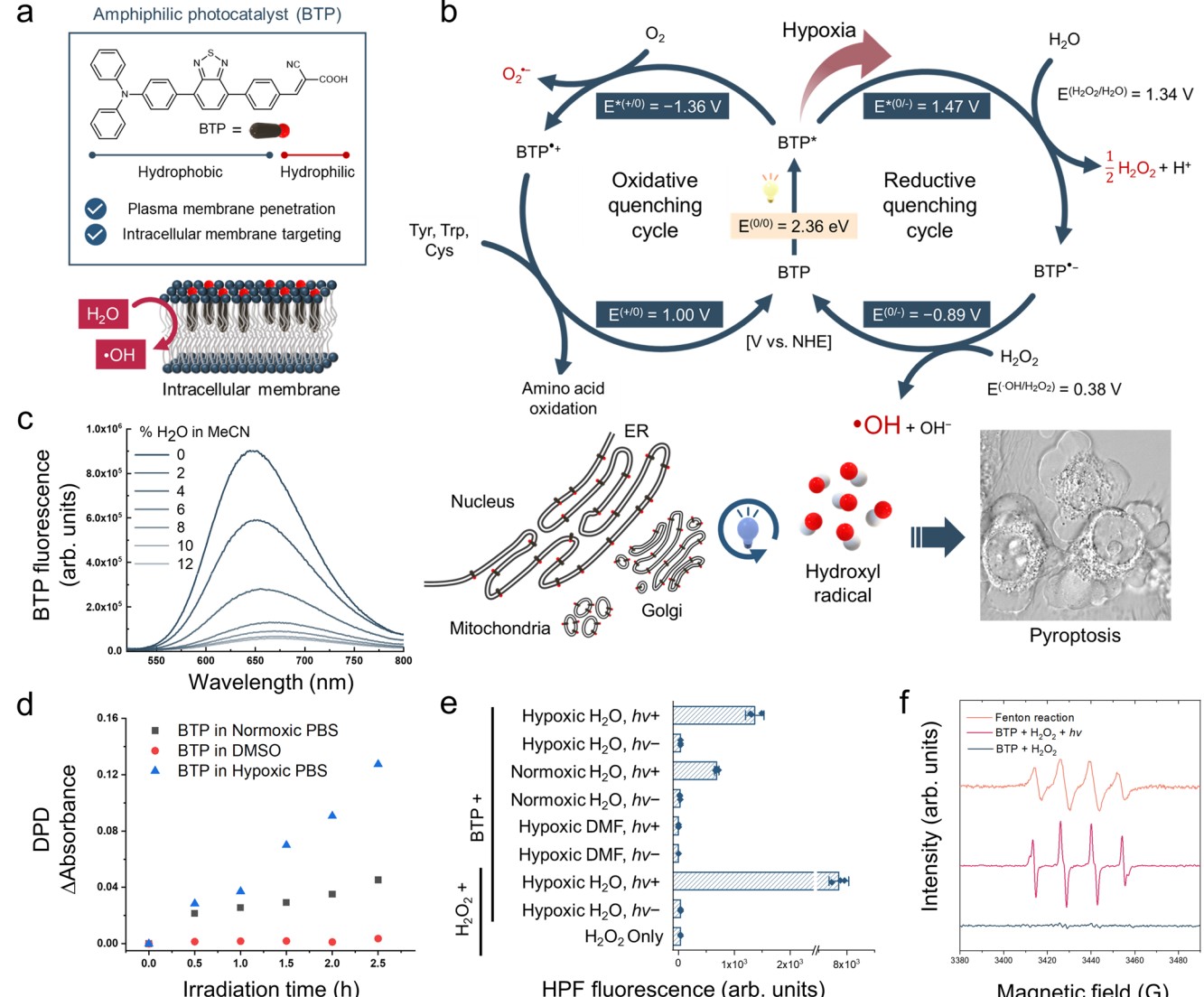

**Fig. 1 | Photocatalytic cycles of BTP for inducing pyroptosis via membrane oxidation. a** Amphiphilic molecular structure of BTP. **b** Photocatalytic cycles of BTP. The right circle represents the reductive quenching cycle that produces $H_2O_2$ and •OH, and the left circle represents oxidative quenching cycle that induces $O_2^{•-}$ generation and amino acid oxidation. This oxidative photocatalysis triggers non-canonical pyroptosis. The inserted cell image shows the pyroptotic morphology of HeLa cells with BTP (5 μM) photocatalysis. **c** Quenching of the fluorescence of BTP by photoinduced electron transfer from $H_2O$. The spectra represent the variation in BTP fluorescence with %$H_2O$ (0 − 12%) in acetonitrile. **d** $H_2O_2$ generation assay with DPD and horseradish peroxidase. BTP (50 μM) in normoxic PBS, Ar-bubbled PBS (hypoxic PBS), and DMSO were irradiated by the blue LED ($\lambda_{max}$ = 450 nm,

66.7 mW·cm$^{-2}$) for 150 min. At 30 min intervals, the change in absorbance at 551 nm was measured to indicate $H_2O_2$ generation. **e** •OH generation assay with HPF. The results represent HPF fluorescence measured under various conditions (See method section) with/without light exposure (blue LED, $\lambda_{max}$ = 450 nm, 2 J·cm$^{-2}$) ($n$ = 3 independent experiments). **f** Electron paramagnetic resonance (EPR) spectroscopy with 10 mM BMPO. A spectrum of •OH spin adduct, BMPO-OH, was observed after BTP photocatalysis with $H_2O_2$ ([BTP] = 1 mM, [$H_2O_2$] = 10 mM) and Fenton reaction as positive control ([$Fe_2SO_4$] = 1 mM, [$H_2O_2$] = 10 mM). Data are presented as mean ± s.d. **$P$ = 0.00049. Student's two-tailed $t$ test. Source data are provided as a Source Data file.

water molecules, resulting in the generation of highly oxidising radical species (i.e., hydroxyl radicals) and induction of intracellular membrane-focused oxidative stress even under hypoxia. This oxidative damage irreversibly disrupts the folding stability of membrane proteins. As a result of the oxidative photocatalysis on intracellular membranes, we propose that photocatalysis-induced oxidation occurs mainly in membrane proteins related to protein quality control (PQC), leading to the accumulation of misfolded proteins. Notably, we initially report that the intracellular membrane oxidation activates non-canonical inflammasome caspases (4 and 5) and triggers subsequent gasdermin-D (GSDMD)-driven pyroptosis, which is an immunogenic and inflammatory cell death process mediated by inflammasomes[13–15]. Considering that pyroptosis of cancer cells can stimulate immune responses against tumours[16–18], identifying chemical stimuli capable of inducing pyroptosis and elucidating their underlying mechanisms hold promise for providing strategies to promote antitumour responses. To this end, these results provide a potential approach to induce the activation of non-canonical inflammasome caspases and consequent pyroptosis through intracellular membrane-focused oxidation.

## Results

### Photocatalytic cycles to generate oxidising radical species

We synthesised BTP by pairing benzothiadiazole and triphenylamine, which are an electron donor and acceptor, respectively (Fig. 1a and Supplementary Figs. 1–9). This donor-acceptor type molecular structure promotes the charge separation ability of its excitons, thereby enhancing its electron transfer activity as a photocatalyst. We measured the photophysical and electrophysical properties of BTP to estimate its redox potentials of BTP (Supplementary Fig. 10a–c). Based on the ground and excited state redox potentials of BTP, we suggest that the photocatalytic cycle of BTP causes membrane oxidation (Fig. 1b).

Under light irradiation, the excited state of BTP (BTP*) exists for a few nanoseconds, generating $E^{*(0/-)}$ and $E^{*(+/0)}$ which are the reduction and oxidation potentials of BTP*, respectively. In aqueous environments, only neighbouring molecules, such as water and oxygen, can allow for the electron transfer from water to BTP* ($E^{*(0/-)}$) and from BTP* to oxygen ($E^{*(+/0)}$). Each redox potential was estimated using cyclic voltammetry and band gap energy (see "Methods"). The $E^{*(0/-)}$ of BTP was 1.47 V (vs. the normal hydrogen electrode), which enables water oxidation via a two-electron pathway yielding $H_2O_2$ ($E^{(H_2O_2/H_2O)} = 1.34$ V at pH 7; Fig. 1b and Supplementary Fig. 11a)[19–21]. To examine the electron transfer between water and BTP*, a fluorescence quenching assay was performed (Fig. 1c). The BTP fluorescence gradually decreased as the water content of acetonitrile solution increased and was mostly quenched at 10% water content owing to the reductive electron transfer from water to BTP*. Furthermore, the excited lifetime of BTP* was diminished as water content increased (Supplementary Fig. 12). These results imply that BTP* was reductively quenched via water oxidation.

Correspondingly, we assayed $H_2O_2$, the expected product of water oxidation, using peroxidase and N,N-diethyl-p-phenylenediamine (DPD)[22] (Fig. 1d). BTP photoexcitation generated $H_2O_2$ in aqueous solution, whereas BTP* could not produce $H_2O_2$ without water (in dimethyl sulfoxide). This result revealed that $H_2O_2$ was generated from water during BTP photocatalysis. Furthermore, we found that hypoxia substantially accelerated photocatalytic $H_2O_2$ production, implying that oxygen functions as quencher of BTP* via oxygen reduction ($E^{*(+/0)} = -1.36$ V, $E^{(O_2/O_2^{\cdot -})} = -0.33$ V; Fig. 1b and Supplementary Fig. 11c)[23]. The oxygen reduction reaction competes with water oxidation by BTP*, indicating that hypoxia could enhance the $H_2O_2$ generation. When BTP* accepts an electron, it transforms into BTP$^{\cdot -}$. BTP$^{\cdot -}$ can easily donate its electron to a nearby $H_2O_2$ propagating to a hydroxyl radical (•OH; $E^{(0/-)} = -0.89$ V, $E^{(H_2O_2/\cdot OH)} = 0.38$ V; Fig. 1b and

Supplementary Fig. 11b)[24]. Therefore, we performed a hydroxyphenyl fluorescein (HPF) assay[25] to confirm •OH generation (Fig. 1e and Supplementary Fig. 13). The HPF assay revealed that BTP photocatalysis generated •OH, which was promoted under hypoxic and $H_2O_2$-supplemented conditions. This result implies that •OH is generated from $H_2O_2$, and this reaction is escalated under hypoxia, considering that oxygen functions as an inhibitor of $H_2O_2$ generation by quenching BTP*. Furthermore, •OH production was impaired in the absence of water (in dimethylformamide), implying that •OH is also produced from water-mediated photocatalysis (Fig. 1e). Electron paramagnetic resonance (EPR) spectroscopy with 5-tert-butoxycarbonyl-5-methyl-1-pyrroline N-oxide (BMPO, spin trap for •OH)[26] followed to clarify •OH generation (Fig. 1f).

When oxygen is reduced by BTP*, superoxide radicals ($O_2^{\cdot -}$) can also be produced, and BTP$^{\cdot +}$ can accept an electron from nearby amino acids, such as Trp, Tyr, and Cys (see Supplementary Information; Fig. 1b and Supplementary Figs. 11d and 14a–c). BTP photocatalysis generates $H_2O_2$ and •OH but does not produce singlet oxygen ($^1O_2$; Supplementary Fig. 15). •OH is a highly oxidising reactive oxygen species (ROS) that is sufficient for various amino acids oxidation[27]. In aqueous conditions, we investigated the oxidation of methionine which is one of the most labile amino acids under oxidative stress. High-resolution mass spectrometry (HRMS) revealed the oxidised products of Met (Supplementary Fig. 16), further exhibiting the potential to inflict severe oxidative stress to proteins.

### Oxidative damage on membranes by BTP photocatalysis

Given the molecular structure of BTP which is composed of several lipophilic aromatic rings and a hydrophilic carboxylic acid, the passive diffusion of BTP across plasma membranes can be limited. To explore the cellular uptake of BTP, we investigated its uptake under physiological conditions at 37 °C, at 4 °C, and in the presence of $NaN_3$. The confocal microscopy results show that the uptake of BTP by HeLa cells dramatically decreased under conditions of 4 °C and $NaN_3$, implying that the penetration mechanism of BTP across plasma membranes relies on energy-consuming processes (Supplementary Fig. 17). We next examined where BTP is located and oxidative stress is produced in cells. Co-localisation experiments revealed that BTP was in Golgi apparatus (GA) and endoplasmic reticulum (ER), but not in mitochondria (Supplementary Figs. 18 and 19). However, BTP photocatalysis changed the localisation pattern from the ER to the mitochondria and plasma membrane (Supplementary Figs. 20–22), with a notable relocation of BTP to the mitochondrial membranes, as confirmed by structured illumination microscopy (SIM) with viable HeLa cells (Supplementary Fig. 21). This change in location is likely because BTP photocatalysis reduces the ER integrity[28], leading to its migration to nearby membranes.

We hypothesised that BTP can be especially proximal to intracellular membranes, including the membranes of ER, GA, and mitochondria, considering the amphiphilic BTP structure. To confirm the proximity of BTP to intracellular membranes, we compared its emission peak in cellular and artificial membrane environments (Supplementary Fig. 23a). In a situation where BTP is dissolved in water, the BTP emission peak occurs at 630 nm in a polar environment, whereas within a non-polar environment, such as membrane, it presents a peak at 580 nm (Supplementary Fig. 23a). Upon introducing artificial lipid bilayers (bicelles) to the BTP aqueous solution, the emission peak shifted from 630 nm to 580 nm (Supplementary Fig. 23b). Subsequently, we measured the emission peak of BTP in cellular environments using Lambda-scan analysis, appearing at 580 nm (Supplementary Fig. 23b, c), indicating that BTP molecules are proximal to intracellular membranes. Therefore, these results imply that BTP photocatalysis inside cells generates reactive radicals near intracellular membranes, leading to intracellular membrane-focused oxidative stress.

Furthermore, we found that BTP photocatalysis generated substantial membrane oxidative stress. The dichlorodihydrofluorescein diacetate (DCFH₂-DA, a ROS indicator) assay showed that BTP photocatalysis increased DCF fluorescence in HeLa cells (Supplementary Figs. 24 and 25). Additionally, we established BTP photocatalysis-induced generation of $O_2^-$ using a dihydroethidium (an $O_2^-$ sensor) assay (Supplementary Fig. 26). These results indicate that BTP photocatalysis induces oxidative stress on intracellular membranes.

Additionally, we investigated cellular lipid oxidation caused by BTP photocatalysis (Supplementary Fig. 27) since lipids are the main component of bio-membranes. Cellular lipids were extracted from HeLa cells with and without BTP photocatalysis, and subsequently analysed using ultra-performance liquid chromatography-mass spectrometry (UPLC-MS). Interestingly, we could not detect any newly generated peaks in the chromatogram following BTP photocatalysis

(Supplementary Fig. 27a). Furthermore, the UPLC-MS results indicated that the oxidation ratio of 15:0–18:1 phosphatidylcholine (PC) was not changed after BTP photocatalysis (Supplementary Fig. 27b). These results suggest that the lipid oxidation caused by BTP photocatalysis might be inefficient in cellular environments. This inefficiency could be due to the limited diffusion of polar hydroxyl radicals into the hydrophobic regions within the lipid bilayer.

### Destabilisation of membrane protein fold by BTP photocatalysis
Membrane proteins are crucial components of intracellular membranes, and their structural damage leads to impaired functions, ultimately affecting cell fate and programmed death signalling[29]. Thus, using an in vitro membrane protein stability assay (Fig. 2a–d), we examined the effects of BTP photocatalysis on membrane protein folds. We adopted *E. coli* rhomboid protease GlpG consisting of six

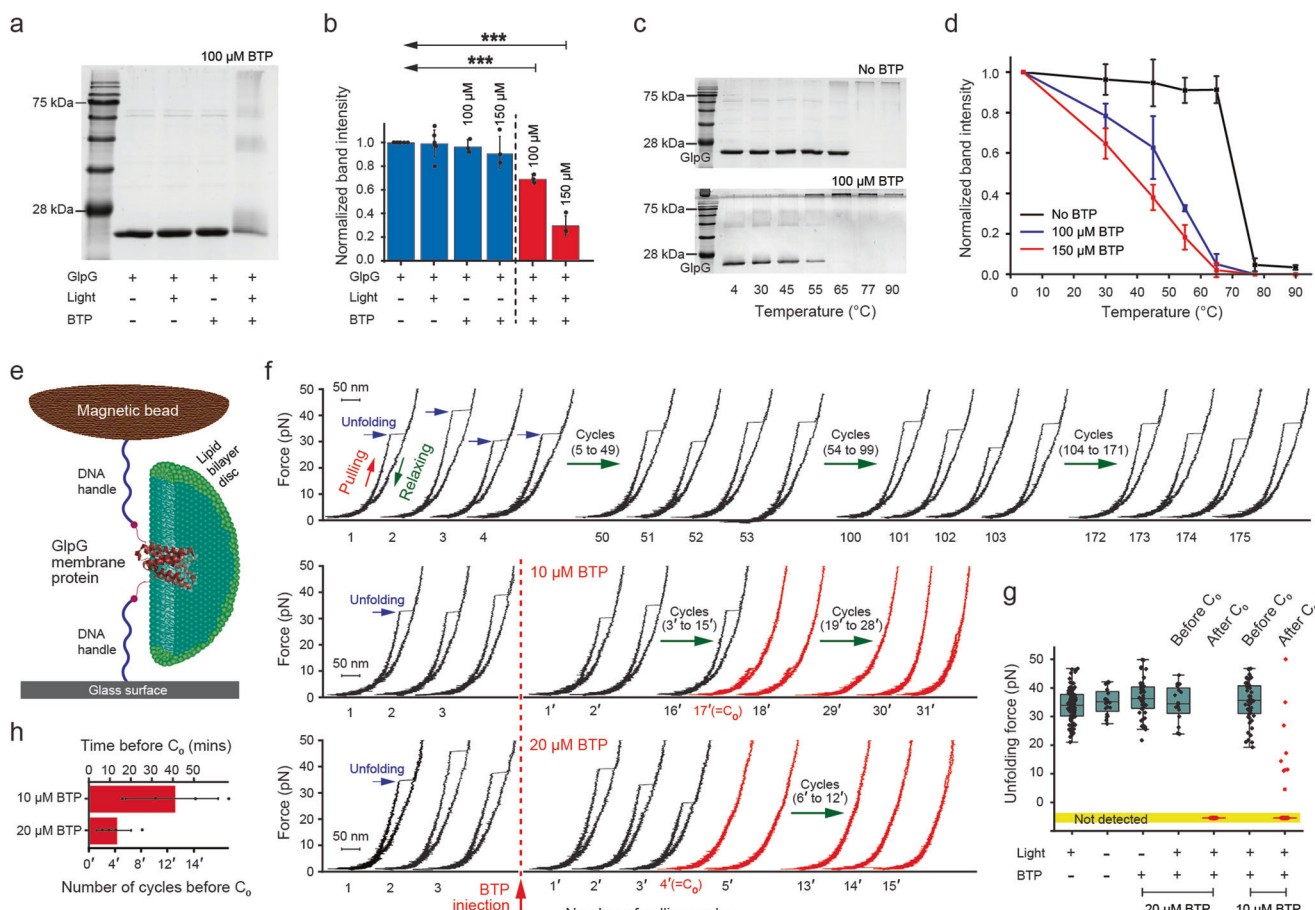

**Fig. 2 | Membrane protein stability assay and single-molecule forced-unfolding assay. a** 15% SDS-PAGE confirming the destabilisation and aggregation of GlpG by BTP photocatalysis. The GlpG sample with 100 μM BTP added was irradiated by blue LED (λ_peak = 450 nm, 30 J·cm⁻²). **b** Quantification of the GlpG destabilisation with normalised band intensity. Data are presented as mean ± s.d. ***$P = 0.001$. One-way ANOVA with post-hoc Turkey HSD test ($n = 6$ independent samples for BTP− conditions, $n = 3$ independent samples for BTP+ conditions). **c** Thermal denaturation assay of GlpGs with or without BTP photocatalysis (λ_peak = 450 nm, 30 J·cm⁻²). Each GlpG sample was incubated at various temperatures for 10 min and then analysed by 15% SDS-PAGE (see Methods for details). **d** Normalised GlpG band intensity of the thermal denaturation assay at each temperature ($n = 6$ independent samples for BTP− conditions, $n = 3$ independent samples for BTP+ conditions). **e** Schematic diagram of single-molecule forced-unfolding assay. The lipid bilayer environment was reconstituted by the bicelle, a lipid bilayer disc composed of lipids and detergents. **f** Representative force-extension curves of GlpG with or without BTP photocatalysis. The repetitive force

scanning of 1–50 pN allows for the observation of repetitive GlpG unfolding. Normal unfolding of GlpG was maintained more than hundred pulling cycles, whereas upon BTP addition, the unfolding forces were drastically reduced. The pulling-cycle number at which the abnormal unfolding ( < 15 pN) appears for the first time is marked as $C_0$. **g** Scatter plot of unfolding forces for various conditions. The light (+) and light (−) indicates the exposure of blue light (λ_peak = 450 nm, 9.16 mW·cm⁻²) and infrared light (λ_peak = 850 nm, 39.51 mW·cm⁻²), respectively. The lower and upper limits of the box indicate the lower quartile (25%) and the upper quartile (75%), respectively. The central line of the box represents the median value, and each whisker extends from the box limits to the furthest data point within 1.5 times the interquartile range (IQR). The number of unfolding cycles for each condition is 118 ($n = 11$ molecules), 20 ($n = 3$ molecules), 48 ($n = 3$ molecules), 16 ($n = 4$ molecules), 36 ($n = 4$ molecules), 52 ($n = 4$ molecules), and 39 ($n = 4$ molecules), respectively. **h** Time span and number of unfolding cycles after BTP addition before the $C_0$ point ($n = 4$ molecules; mean ± s.d.). Source data are provided as a Source Data file.

transmembrane helices, which is a widely studied membrane protein for its folding and stability[30–32]. The helical bundle protein can serve as an appropriate model for investigating the photocatalytic oxidation effects on helical membrane protein, which represents the largest class in a structural perspective[33–35].

Only under the condition of BTP added and blue light exposure (30 J·cm$^{-2}$), we found that the intensity of the gel band for GlpG (~ 22 kDa) was decreased to 31 ± 9.0% from the negative control at 150 μM BTP (Fig. 2a, b), whereas the band intensities for GlpG aggregates ( > 22 kDa) were increased (Fig. 2a). This indicates the destabilisation and aggregation of GlpG due to its oxidative damage by BTP photocatalysis. This result was also supported by thermal denaturation assay (Fig. 2c, d). Indeed, the resistance to the thermal denaturation of GlpG was reduced by the oxidative damage. The transition midpoint temperature of the thermal denaturation was largely decreased from 75 °C at no BTP condition to 35 °C at 150 μM BTP (Fig. 2c, d).

We employed a robust single-molecule tweezer approach to confirm the oxidative destabilisation of GlpG[36], which is likely the original cause for the protein aggregation (Fig. 2e–h). This method entirely excludes the aggregation events[37], allowing us to solely focus on the photocatalytic oxidation effects on the protein stability. In this method, a single GlpG embedded in a lipid bilayer disc (bicelle) was observed to undergo reversible unfolding by mechanical force[30,38]. The repetitive unfolding was reproducible by more than a hundred cycles under the condition of blue light exposure ($\lambda$ = 450 nm, 9.16 mW·cm$^{-2}$) without BTP (Fig. 2f, upper), and the unfolding forces were distributed at 34 ± 5.8 pN (Fig. 2g; Supplementary Fig. 28a for infrared light of $\lambda$ = 850 nm, 39.51 mW·cm$^{-2}$). However, upon the addition of BTP with blue light exposure, the unfolding forces were drastically reduced to less than 15 pN at a critical point in the pulling cycle ($C_0$), even unmeasurable at 20 μM BTP (Fig. 2f, middle and lower). After the $C_0$ point, the unfolding force values did not return to the normal level observed for the GlpG unfolding (Fig. 2f–h). Moreover, the injection of fresh bicelles after $C_0$ was unable to restore the normal unfolding (Supplementary Fig. 28b, c) despite the rapid reconstitution of the intact membrane environment from fresh bicelles, which facilitates the reversible unfolding of GlpG[30]. These results indicate irreversible oxidative damage to the membrane protein, induced by BTP photocatalysis, leading to significant destabilisation of its native protein fold.

## Impact of BTP photocatalysis on protein quality control proposed by proteomics

Since methionine is one of the most labile amino acids under oxidative stress, proteins containing oxidised methionine residues (O-Met) were analysed in HeLa cells using label-free quantitative mass spectrometry for an initial screening of BTP oxidation targets (Fig. 3a)[27,39,40]. The extent of oxidative damage was evaluated for each protein by comparing the average O-Met mass spectra intensities of the experimental groups with those of the control group (Fig. 3a, inset). Proteins that were oxidised more than 2-fold in the experimental group compared to the control group, with p-values lower than 0.05, were considered as oxidised proteins by BTP photocatalysis (p-value < 0.05, Fold Change >2). The identified proteins were categorised based on their GO annotation by cellular location, determining whether they were membrane-localised or not (Fig. 3b and Supplementary Fig. 29a). Proteins annotated as being localised to plasma, organelles, or various other membranes were classified as membrane-specific. Cytosolic proteins, excluding the membrane-cytosol overlying proteins, were then compared with the membrane-specific proteins (Fig. 3c). A greater number and proportion of oxidised proteins were observed in membrane-specific proteins compared to cytosolic proteins: 339 versus 40, accounting for 24.5% and 3.6%, respectively. This result supports the membrane-focused oxidative stress, which matches with the

membrane-localisation property of BTP. Additionally, the proportions of oxidised membrane proteins of the mitochondria, ER, nucleus, and GA were 31.1%, 19.1%, 21.1%, and 7.7%, respectively (the number of oxidised membrane proteins/the detected number of membrane proteins, Fig. 3d). The global membrane oxidation induced by BTP photocatalysis suggests a potential malfunction in biological processes that require the involvement of various organelles.

To further elucidate other oxidation targets of BTP photocatalysis at the membrane, an in-depth secondary search covering 17 amino acids was conducted using mass spectra not identified in the initial search for O-Met analysis (Fig. 3a). This multistage search allowed us to scrutinise extensive protein oxidations by reducing the search space and thus decreasing the number of false positives. Interestingly, membrane-specific proteins were the most oxidised (oxidised protein criteria: p-value < 0.01, Fold Change > 4), followed by membrane-cytosolic proteins, and then cytosolic proteins for all detected amino acid residues (Fig. 3e, f and Supplementary Fig. 29b, c). It is noteworthy that the number of oxidised proteins, as defined by Trp and His oxidation, was prevalent in membrane-specific proteins (Fig. 3f and Supplementary Fig. 29c). Considering their low abundance in a whole cell, it suggests a favourable interaction between BTP and oxidisable-aromatic amino acids. Detailed information on the oxidised proteome for each amino acid is available in Source Data file.

Furthermore, the membrane-specific and membrane-cytosolic proteins were collected to evaluate the total extent of oxidation (Fig. 4a). We selected 250 oxidised proteins as our proteome of interest, applying a conservative threshold (p-value < 0.01, Fold Change > 4; Fig. 4b). These 250 oxidised membrane proteins were predominantly located in the ER, mitochondria, nucleus, and GA, corresponding with the initial O-Met screening (Fig. 4c). Notably, the ER, GA, and mitochondria—three organelles crucial for protein quality control (PQC)[41]—possessed 58.1% of the oxidised membrane proteins resulting from BTP photocatalysis. We therefore focused on PQC-related functions, such as unfolded protein response (UPR) and protein transport. Among 250 oxidised membrane proteins, 97 proteins were categorised into four functional networks, (i) UPR and ER-associated degradation (ERAD), (ii) ER–Golgi transport, (iii) mitochondrial trafficking and transport, and (iv) lipid metabolism (Fig. 4d).

UPR and ERAD are apparently key quality control mechanisms necessary to alleviate stress from the accumulation of misfolded proteins[42]. Additionally, ER-Golgi transport is a process that facilitates ER quality control[43]. Moreover, mitochondrial trafficking and transport are crucial for mitochondrial functions, which mediate mitochondrial UPR[44,45], another axis of the cellular quality control mechanism. Lipid metabolism is also important, as an imbalance in lipid homoeostasis can stimulate UPR[46]. Furthermore, clustering the 250 oxidised proteins by Gene Ontology (GO) biological process and assessing GO enrichment scores demonstrated the oxidative damage imposed by BTP on biological processes related to cellular quality control via the ER and mitochondrial UPR (Fig. 4e). As described in the folding stability experiments (Fig. 2), the significantly oxidised membrane proteins might have lost their folding stability. Therefore, we hypothesised that BTP-induced oxidation and the resulting dysfunction of proteins could deteriorate PQC and escalate UPR stress.

## Cation mobilisation by BTP photocatalysis

BTP photocatalysis causes the destabilisation of the membrane protein structure related to PQC, leading to an irreversible accumulation of misfolded proteins, thereby enhancing stress on the ER, GA, and mitochondria[47,48]. Failure of the PQC and subsequent maladaptive UPR has been reported to trigger Ca$^{2+}$ mobilisation[49], and we confirmed this using Rhod-2 (a Ca$^{2+}$ indicator). Ca$^{2+}$ concentration in the mitochondria increased considerably following light exposure (0.3 mW) for 30 s, indicating mitochondrial Ca$^{2+}$ uptake (Fig. 5a). The line-cut analysis

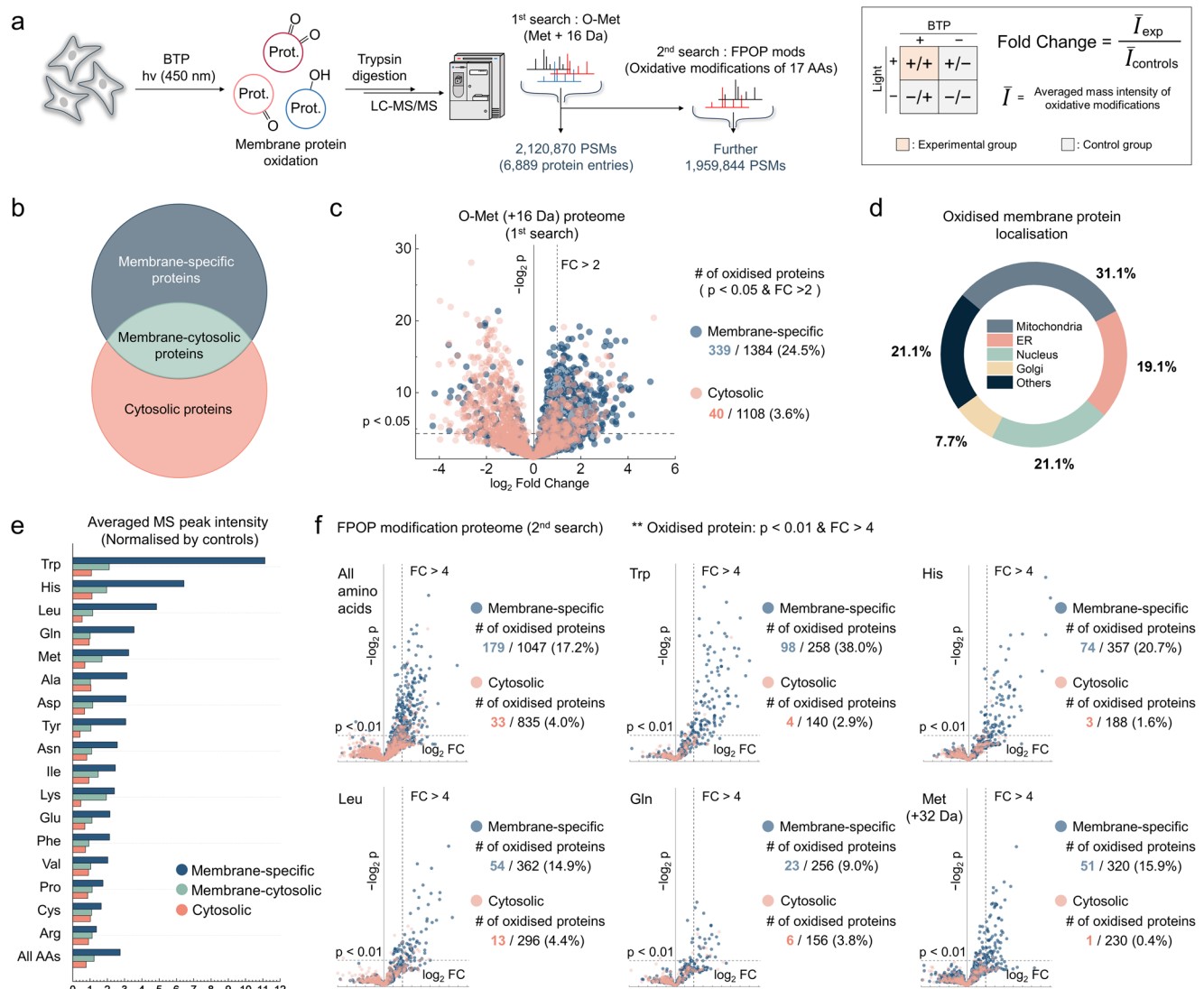

**Fig. 3 | Comprehensive proteomic profiling of membrane protein oxidation induced by BTP photocatalysis. a** Schematic illustration of the proteomic analysis workflow used to investigate the extent of oxidative modifications induced by BTP photocatalysis. The process involves a multistage search strategy for identifying O-Met and FPOP (fast photochemical oxidation of proteins) modifications. In the first search, 2,120,870 peptide-spectrum matches (PSMs) were found, and in the second search, 1,959,844 PSMs were found. Samples subjected to BTP photocatalysis and control samples were analysed and compared based on the fold change in the average precursor intensity of oxidative modifications (inset). **b** Proteins categorised by GO subcellular annotations into 'membrane-specific', located exclusively on membranes, and 'membrane-cytosol', found on both membranes and cytosol. The remaining cytosolic proteins were labelled 'cytosolic' proteins. **c** Volcano plot of O-Met (+16 Da) proteome showcasing oxidation focused on membrane-specific proteins versus cytosolic proteins. Proteins with a *p*-value < 0.05 and Fold Change > 2 were defined as potential oxidation targets of

BTP photocatalysis. *p* values were calculated for Student's one-tailed *t* test. **d** The proportions of oxidised membrane proteins across different organelles based on the O-Met proteome. 'Others' include plasma membranes and unidentified locations. **e** Overview of the 2nd search based on FPOP modifications, showing average oxidation intensities for different amino acids. 'All AAs' represents the aggregated intensities of oxidative modifications of these 17 amino acids. The averaged oxidation intensities of 'membrane-specific', 'membrane-cytosolic', and 'cytosolic' proteins for the corresponding amino acids were presented to compare the degree of oxidation of membrane proteins and soluble proteins for each type of amino acid. The averaged oxidation intensities of three control conditions were normalised to 1. **f** Volcano plots of the proteome other than O-Met, contrasting membrane-specific proteins with cytosolic proteins. Stricter criteria than sole O-Met analysis (*p*-value < 0.01 and Fold Change > 4) were applied for robust identification of oxidised proteins. *p* values were calculated for Student's one-tailed *t* test. Source data are provided as a Source Data file.

supported that the MitoTracker signal was well merged with the increased Rhod-2 signal (Fig. 5b, c) after BTP photocatalysis. This tendency was also observed in flowcytometry with Rhod-2 (Fig. 5d and Supplementary Fig. 30). Accumulation of misfolded proteins and Ca²⁺ leads to osmotic swelling, resulting in mitochondrial dysfunction. We observed BTP photocatalysis-induced mitochondrial swelling accompanied by fission and fusion, known as the mitochondrial PQC process (Fig. 5e)[50]. Accordingly, we conducted a mitochondrial membrane potential assay using TMRE staining. After BTP photocatalysis, the TMRE fluorescence are dramatically diminished, implying that the

mitochondrial membrane potential and functions are damaged after BTP photocatalysis (Fig. 5f). Simultaneously, we found intracellular K⁺ efflux after BTP photocatalysis using flowcytometry with ION K⁺ Green-2 (K⁺ indicator; Fig. 5g). These results show that oxidative photocatalysis induces maladaptive UPR and cation mobilisation in response to oxidative damage of intracellular membrane.

### Non-canonical inflammasome caspases-induced pyroptosis
Given the impact of BTP photocatalysis on membranes, we examined how BTP photocatalysis might impact cell death signalling responses.

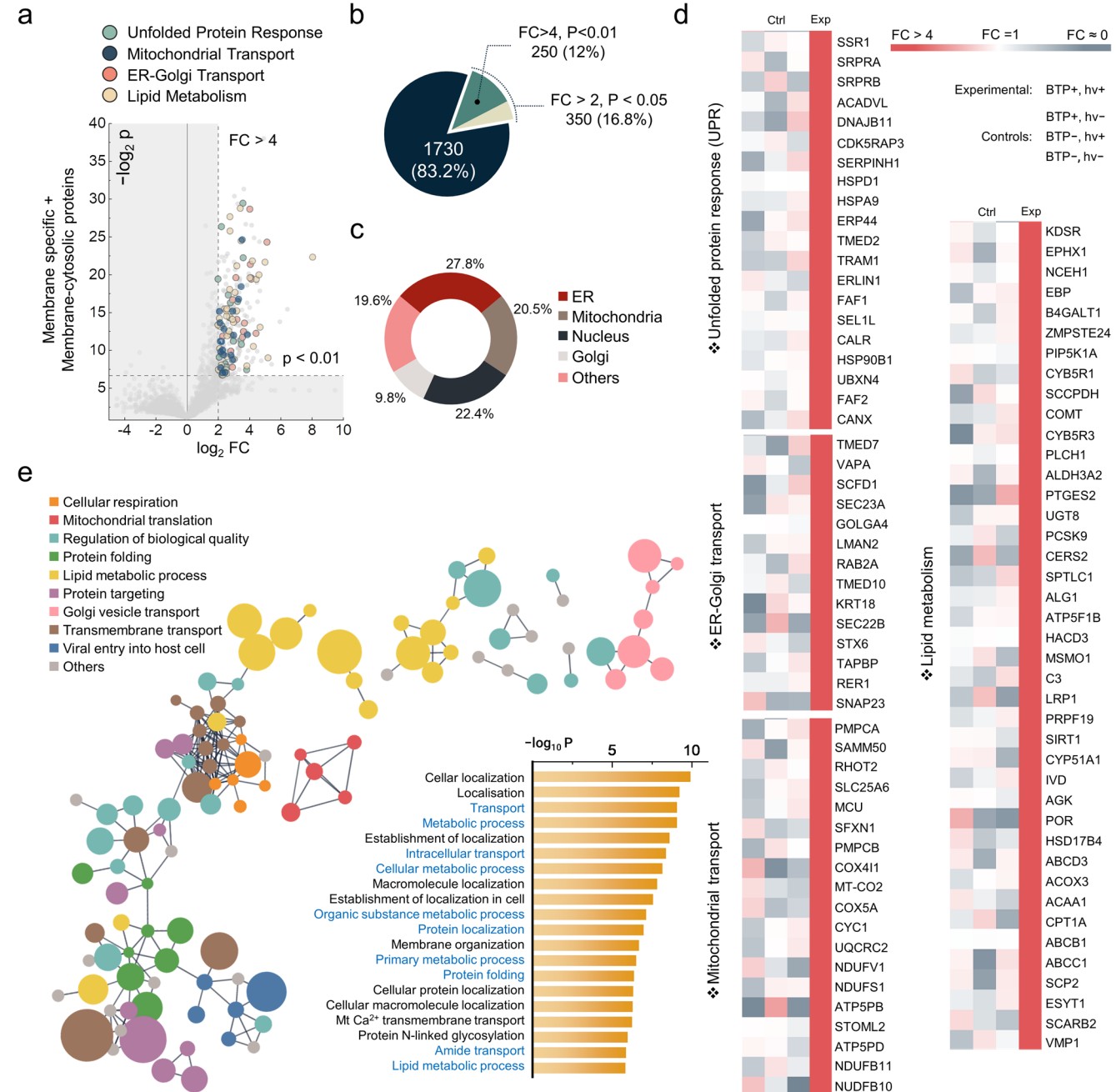

**Fig. 4 | Functional implication of oxidised membrane proteins by BTP photocatalysis. a** Protein quality control (PQC) related proteins was highlighted on the volcano plot of membrane proteins by their functional categories. Oxidised proteins were defined by the strict criteria (*P*-value < 0.01, Fold Change > 4), based on the intensity of oxidative modifications of all amino acids excluding Gly, Ser, Thr, and O-Met (+16 Da). **b** The count and percentage of proteins satisfying the oxidation criteria. **c** Distribution of oxidised membrane proteins across cellular organelles. 'Others' include plasma membranes and unidentified locations.

**d** Heatmap comparison of highlighted protein oxidation between experimental and control conditions. **e** String network of the strictly defined oxidised proteins (*P* < 0.01, FC > 4), filtered for high interaction confidence (0.9) illustrated with GO biological processes and GO enrichment scores. Node size reflects log₂FC values, and disconnected nodes were excluded from the network. Key PQC-related processes were marked in blue. All *P* values were calculated for Student's one-tailed *t* test. Source data are provided as a Source Data file.

HeLa cell viability was tested by examining propidium iodide (PI) or calcein AM uptake, and the MTT assay. The results showed that nearly all HeLa cells died within 24 h following BTP photocatalysis (Fig. 6a, b). Even in hypoxic pancreatic cancer cell lines (Panc-1 and MiaPaca-2), MTT assays indicated severe toxicity caused by escalated •OH generation under hypoxia (Fig. 6c). An examination of the cellular morphology indicated that BTP photocatalysis caused a lytic-type cell death, consistent with a pyroptotic morphology; including plasma membrane swelling and abnormal blebbing (Fig. 7a and

Supplementary Fig. 31)[13,15]. BTP photocatalysis induced lactate dehydrogenase (LDH) release, a measure of plasma membrane rupture, to a greater extent than the pyroptotic stimuli, LPS and nigericin, or the photosensitiser (Ce6) used in photodynamic therapy (Fig. 7b), indicating that cell death induced by BTP photocatalysis is highly lytic. Contrary to apoptosis, PI penetrated the plasma membrane but not the nuclear envelope within an hour after BTP photocatalysis (Supplementary Fig. 32a), implying that the nuclear envelope collapses only after plasma membrane integrity is compromised. Based on these cell

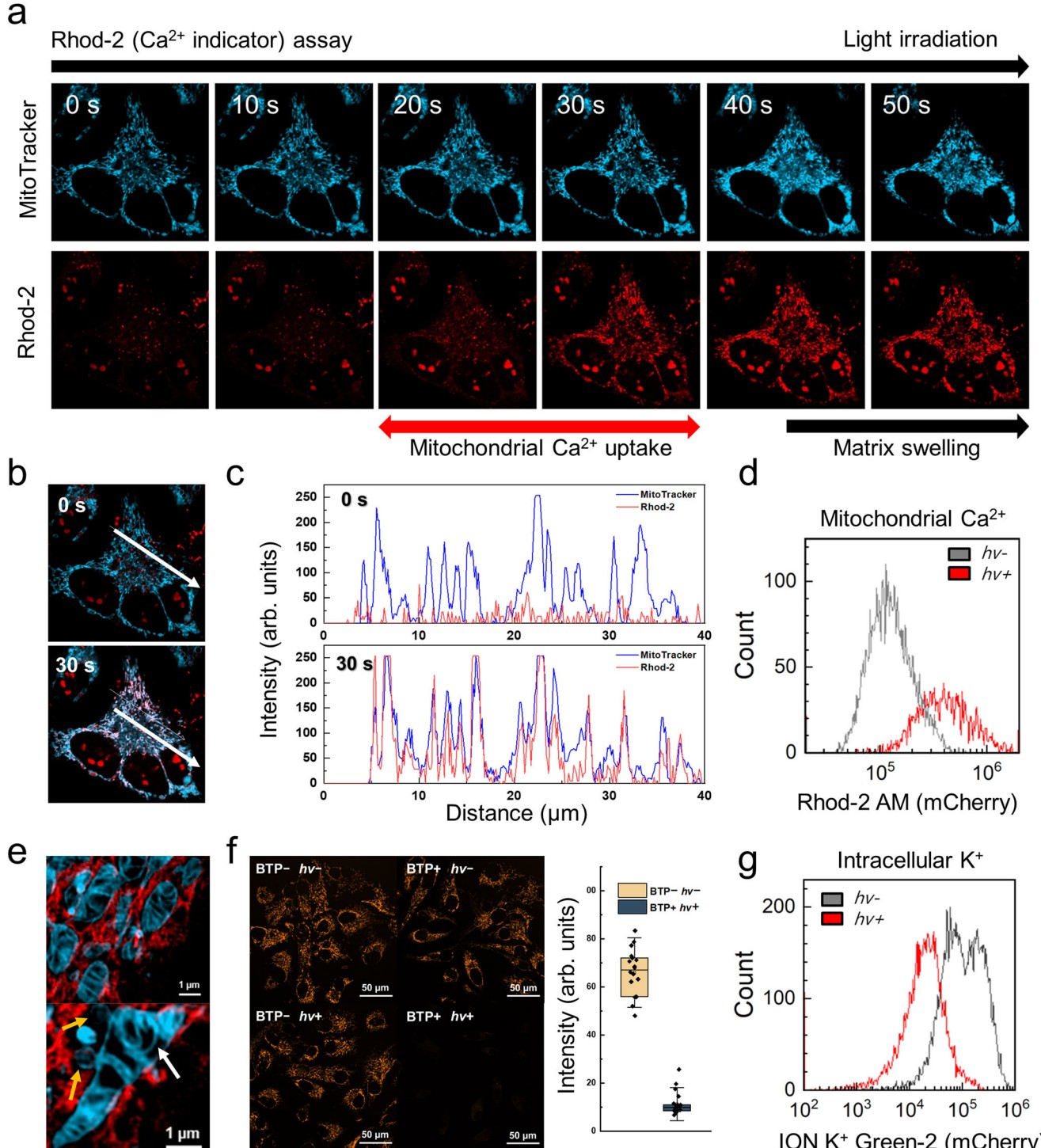

**Fig. 5 | Ca²⁺ and K⁺ mobilisation by BTP photocatalysis. a** Mitochondrial Ca²⁺ assay performed using Rhod-2. HeLa cells were incubated with BTP (5 μM), MitoTracker™ Deep Red FM (0.5 μM), and Rhod-2 (3 μM). The fluorescence of MitoTracker (cyan) and Rhod-2 (red) was measured using time-series confocal microscopy ($t = 0$–50 s, 10 s interval) during light exposure ($\lambda = 445$ nm, 0.3 mW). The fluorescence of Rhod-2 was enhanced dramatically between 20 and 30 s, implying that Ca²⁺ mobilisation occurred at this time. Mitochondrial matrix swelling following Ca²⁺ uptake was also observed after BTP photocatalysis. **b** Merged images of MitoTracker and Rhod-2 signals at $t = 0$ and 30 s. **c** Line-cut analysis of white arrows in (**b**). **d** Flowcytometry for Ca²⁺ mobilisation. HeLa cells were treated with BTP and Rhod-2, and the Rhod-2 fluorescence of each cell was measured before ($h\nu$−) and 2 h after ($h\nu$+) light exposure ($\lambda_{max} = 450$ nm, 10 J·cm⁻²). **e** Live-SIM

images of ER (red) and mitochondria (cyan) after BTP photocatalysis ($\lambda = 488$ nm, 10 mW). Mitochondrial swelling (top), fission, and fusion (bottom) were observed in HeLa cells. Arrows indicate mitochondrial fission (yellow) and fusion (white). **f** Mitochondrial membrane potential assay using tetramethylrhodamine, ethyl ester (TMRE). HeLa cells were incubated with BTP (10 μM) and TMRE (0.5 μM) and irradiated with blue LED light ($\lambda_{max} = 450$ nm, 10 J·cm⁻²). Box plot analysis of TMRE signals from randomly selected cells (BTP+/$h\nu$+ and BTP−/$h\nu$−) ($n = 18$ and 21). The whiskers represent the standard deviations (s.d.), and the box represents to 25% and 75% of the s.d. **g** Flowcytometry of K⁺ efflux in HeLa cells. Intracellular K⁺ was measured with ION K⁺ Green-2, an intracellular K⁺ sensor, before ($h\nu$−) and 2 h after ($h\nu$+) BTP photosensitisation ($\lambda_{max} = 450$ nm, 10 J·cm⁻²). Data are presented as mean ± s.d. Source data are provided as a Source Data file.

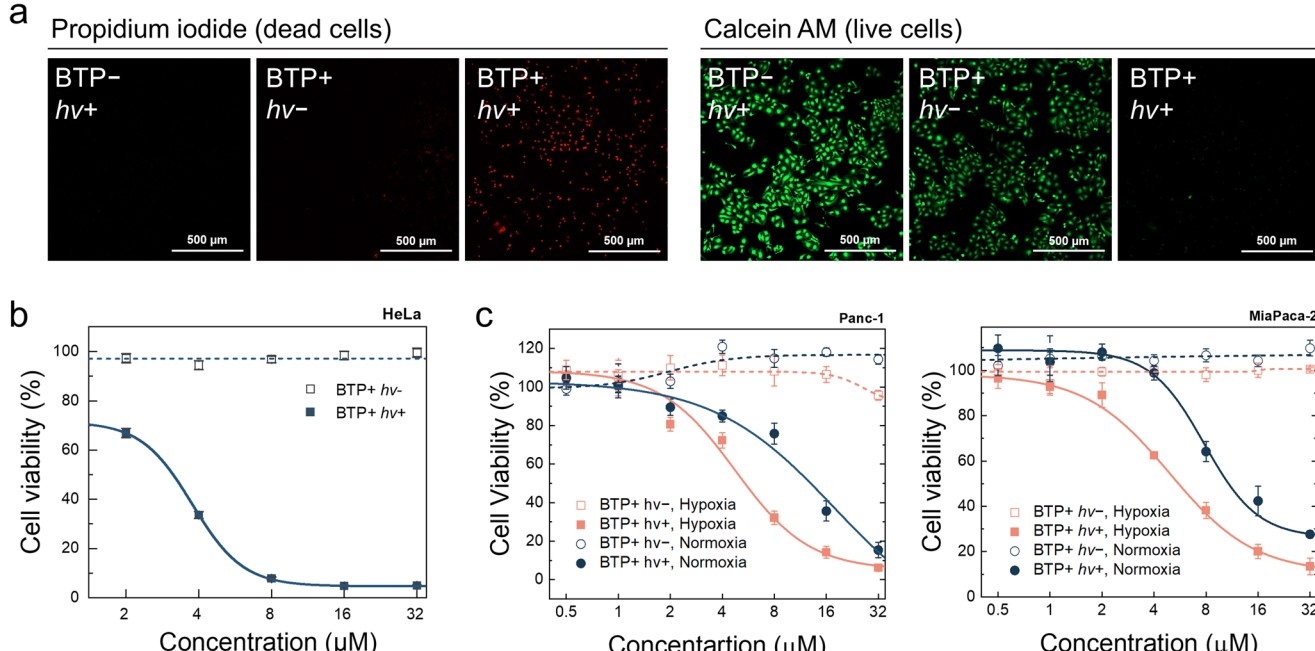

**Fig. 6 | BTP photocatalysis induced cytotoxicity. a** Live or dead assay with Calcein AM (green) and propidium iodide (PI, red). HeLa cells with BTP photocatalysis were stained by Calcein AM and PI 24 h after light exposure ($\lambda_{max}$ = 450 nm, 3.7 J·cm$^{-2}$). The experiment was repeated three times independently, and each experiment showed similar results. **b** MTT assay of HeLa cells with BTP photocatalysis ($\lambda_{max}$ = 450 nm, 10 J·cm$^{-2}$) ($n$ = 4 biologically independent samples). **c** MTT assays for normoxic/hypoxic pancreatic cancer cells (Panc-1 and MiaPaca-2). All data are presented as mean ± s.d ($n$ = 4 biologically independent samples). *$P$ < 0.05. Student's two-tailed $t$ test. Source data are provided as a Source Data file.

death characteristics, we hypothesised that BTP photocatalysis triggers pyroptosis.

In many cases, •OH production and lipid peroxidation are associated with ferroptosis; thus, we first distinguished the characteristics of cell death caused by BTP photocatalysis from ferroptosis. Liproxstatin-1 (a lipid peroxidation and ferroptosis inhibitor) and z-VAD-fmk (a pan-caspase inhibitor) were used to investigate whether caspases or lipid peroxidation are involved in the lytic cell death triggered by BTP photocatalysis. Interestingly, z-VAD-fmk treatment reduced LDH release caused by BTP photocatalysis, while Liproxstatin-1 was less effective (Supplementary Fig. 32b). In addition, substantial levels of membrane blebbing and PI penetration were still observed in Liproxstatin-1-treated cells, but not in z-VAD-fmk-treated cells (Supplementary Fig. 32c). These results indicate that BTP photocatalysis-triggered cell death depends on caspase activation rather than on lipid peroxidation.

GSDMD cleavage by inflammatory caspases (i.e., caspase-1, -4, or -5) releases the N-terminal domain (GSDMD-NT), which then enacts pyroptosis via forming pores in the plasma membrane, resulting in non-selective ionic flux and the release of cellular immunogenic molecules[14]. Notably, BTP photocatalysis resulted in increased detection of GSDMD-NT in the cell lysates and cell media (Fig. 7c). Furthermore, pyroptotic morphology and LDH release caused by BTP were eliminated in GSDMD$^{-/-}$ iBMDMs when compared to wildtype (WT) control macrophages (Fig. 7d, e and Supplementary Fig. 33). Because GSDMD is typically cleaved by caspase-1 that is engaged by canonical inflammasomes[14], or by caspase-4/5, referred to as non-canonical inflammasomes[51], we examined the processing-associated activation of these caspases using western blot. Unexpectedly, these results showed that BTP photocatalysis causes cleavage of caspase-4/5 rather than caspase-1 (Fig. 7f and Supplementary Fig. 34). We further confirmed ATP secretion by caspase-4/5 and GSDMD activation (Fig. 7g), while the efficient secretions of cleaved interleukins

(IL-18 and IL-1β) were not observed in the western blot analysis due to the lack of active caspase-1 (Fig. 7h). Considering that ER stress or cation mobilisation has been implicated in the activation of caspase-4/5[52–55], we surmise that the accumulation of misfolded proteins by BTP photocatalysis may trigger the cleavage of caspase-4/5 and subsequent pyroptosis.

## Discussion

We propose that photocatalytic membrane oxidation triggers non-canonical pyroptosis using the amphiphilic organic photocatalyst, BTP (Supplementary Fig. 35). Via photocatalysis, BTP generates highly oxidising •OH in a spatiotemporally controlled manner even under hypoxic conditions, thereby damaging the structural stability of membrane proteins. The single-molecule tweezer approach verified that BTP photocatalysis disrupts membrane protein folding. Using the oxidised proteome from the label-free quantification, we found that BTP photocatalysis substantially oxidised PQC-related membrane proteins of ER, GA, and mitochondria in cells. Disruption of the folding stability and oxidation of PQC-related proteins seemed to stimulate the accumulation of misfolded proteins, followed by ER stress, maladaptive UPR, and cation mobilisation. These cellular responses consequently triggered the caspase-4/5-induced GSDMD cleavage and subsequent pyroptosis.

Since pyroptosis is known to generate the most robust immune response, recent studies have focused on various triggers for this cell death pathway. Pyroptosis is usually caused by microbial infection or endotoxins such as LPS. However, we suggest that the intracellular membrane-focused oxidative stress can trigger pyroptosis through non-canonical inflammasome activation. This endotoxin-independent mechanism implies an alternative pathway for inducing pyroptosis. Although the full spectrum of biological processes and their causal relationships remain unexplored in this study, we believe that this study can inspire further research into the pathogenesis of immune-related diseases. In additon, light-controlled pyroptosis can be useful

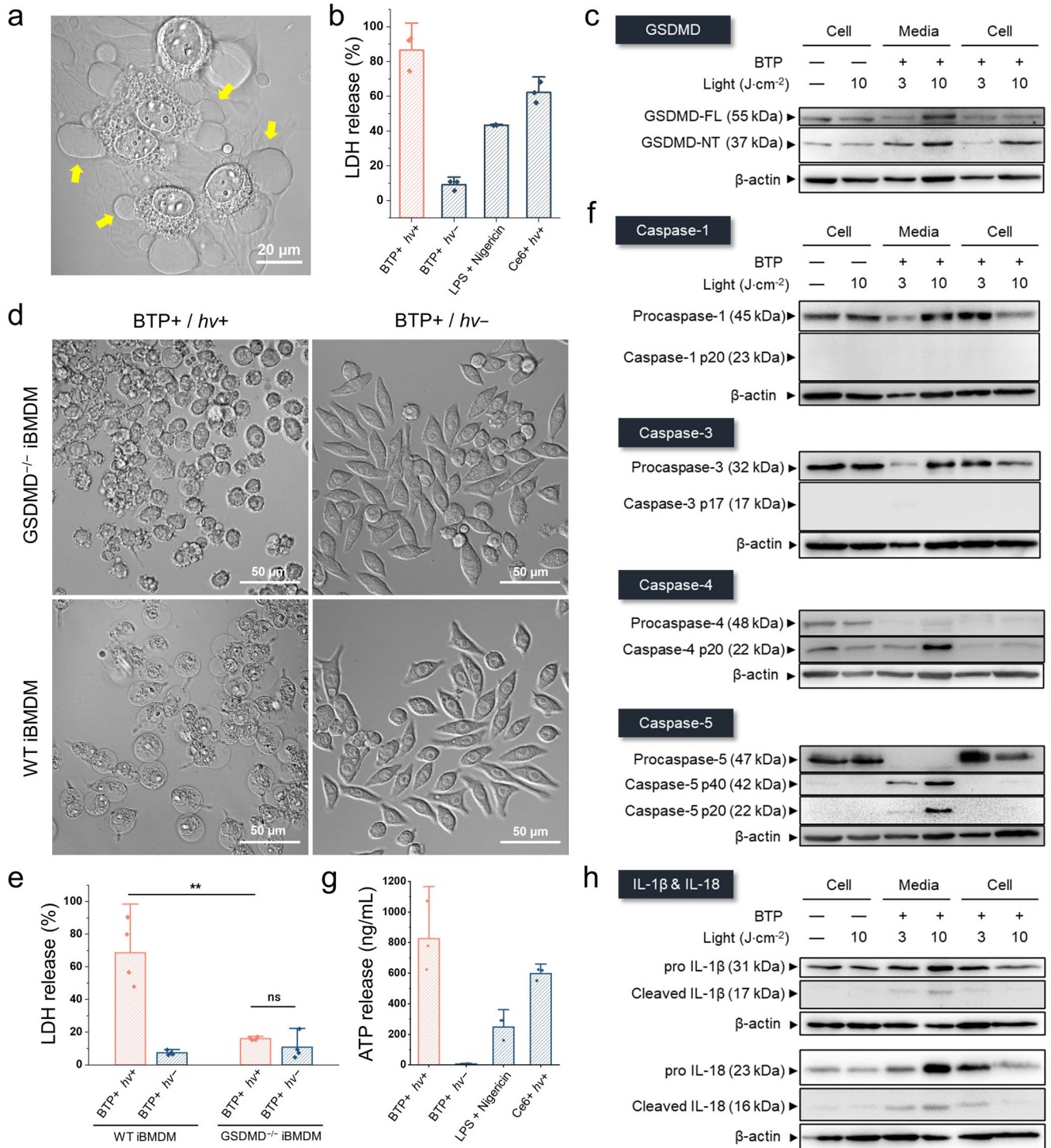

**Fig. 7 | Caspase-4/5-mediated pyroptosis by oxidative photocatalysis on membranes. a** Pyroptotic morphology changes in response to photocatalytic membrane oxidation. Yellow arrows indicate pyroptotic blebbing of dyeing HeLa cells. **b** Lactate dehydrogenase (LDH) release assay. Ce6 (photodynamic therapy agent) and lipopolysaccharide (LPS) + nigericin were used as positive controls for photooxidation-induced cell death and LPS-induced pyroptosis. ($n = 3$ biologically independent samples). **c** Western blots of HeLa cells with BTP photocatalysis for investigating gasdermin D (GSDMD) cleavage. BTP-treated cells were exposed to 3 or 10 J·cm$^{-2}$ of light energy, and the pyroptotic media (Lane: Media) and cell lysate (Lane: Cell) were obtained individually 2 h after BTP photocatalysis. **d** Changes in

the morphology of wild-type immortalised bone marrow-derived macrophages (WT-iBMDMs) and GSDMD knock-out iBMDM (GSDMD$^{-/-}$ iBMDM) in response to BTP photocatalysis. **e** LDH release assay for WT- and GSDMD$^{-/-}$ iBMDM exposed to BTP photocatalysis. ($n = 4$ biologically independent samples). **f** Western blot analysis of HeLa cells with BTP photocatalysis for investigating caspase-1/3/4/5 cleavage. **g** Secretion assay for ATP evaluation. ($n = 3$ biologically independent samples). **h** Western blot analysis of HeLa cells for examination of interleukin cleavages (IL-18 and IL-1β). All data are presented as mean ± s.d. **$P = 0.0018$. Student's two-tailed $t$ test. Source data are provided as a Source Data file.

to induce immune responses spatiotemporally. In particular, BTP photocatalysis induces pyroptosis even in a hypoxic environment, suggesting that this strategy can be therapeutically attractive, considering that most cancers have a hypoxic environment. Consequently, we hope that this method can be widely used to spatiotemporally induce caspase-4/5 activation and pyroptosis in pathogenesis studies and clinical applications.

## Methods

### Synthesis
The methods for organic synthesis of photocatalysts, characterisation, photophysical property analysis (UV-vis, photoluminescence spectroscopy, and time-correlated single photon counting), ROS assays (ABDA, $H_2$DCF-DA, and DHE assays), photocatalytic Met oxidation, laser scanning microscopy (LSM) and structured illumination microscopy (SIM) imaging, Lipid oxidation analysis using UPLC-MS, cation mobilisation assays, cell viability tests, and all experiments excluded from this section are provided in the Supplementary Information.

### Cyclic voltammetry
Cyclic voltammetry (CV) was conducted using a Vertex Potentiostat/ Galvanostat (IVIUM Technologies, Eindhoven, Netherlands). The CV curves were obtained at a scan rate of 10 mV·s$^{-1}$ and a potential step of 2 mV. The electrochemical measurements were performed in a three-electrode system comprising a glassy carbon working electrode, Ag/ AgCl (saturated KCl solution) reference electrode, and a Pt wire counter electrode. BTP was coated on the working electrode using the drop-casting method for CV measurements. $1 \times$ PBS solution (measured pH = 7.24) was used as the supporting aqueous electrolyte. Before measurements, all electrolyte solutions were degassed with argon gas to prevent interference associated with dissolved $O_2$. We primarily measured oxidation and reduction potential (vs. Ag/AgCl) from on-set values, then converted the values to potential versus the normal hydrogen electrode (NHE) by adding $+ 0.197$ V to obtain $E^{(0/-)}$ and $E^{(+/0)}$. The values are $E^{(0/-)}$ ($-0.89$ V vs. NHE) and $E^{(+/0)}$ (1.00 V vs. NHE).

### Excited redox potentials of BTP
The photocatalytic redox potential was calculated using generally used method[56]. $E^{(0/-)}$ ($-0.89$ V vs. NHE) and $E^{(+/0)}$ (1.00 V vs. NHE) values were measured based on the CV results. To estimate the photocatalytic activity of BTP, $E^{*(0/-)}$ (1.47 V vs. NHE) and $E^{*(+/0)}$ ($-1.36$ V vs. NHE) values were also calculated by adding and subtracting the $E^{(0/0)}$ value (2.36 eV), respectively.

### Hydroxyl radical assay (hydroxyphenyl fluorescein, HPF)
Hydroxyl radicals were detected using an HPF assay. HPF (Invitrogen) is an indicator of hydroxyl radicals and peroxynitrite. The hydroxyphenyl group of HPF is eliminated by the hydroxyl radical, and HPF is subsequently converted to the fluorescent form (fluorescein). Thus, we prepared aqueous BTP (5 µM) and HPF solutions (5 µM) and measured fluorescence at 515 nm under various conditions: (1) before and (2) after BTP photoactivation in an Ar-bubbled aqueous solution, (3,4) in a normoxic aqueous solution, (5,6) in DMF solution, and (7,8,9) in 23 mM $H_2O_2$ solution. In this assay, the final concentration of BTP was 5 µM. Furthermore, blue LED ($\lambda_{max} = 450$ nm) (HepatoChem Inc., USA) was used to photoactivate BTP (16.6 mW·cm$^{-2}$ for 2 min = 2 J·cm$^{-2}$), and all conditions except for the normoxic condition (3,4) were maintained in conjugation with Ar bubbling conditions. A microplate reader (SpectraMax M5e, USA) was used to measure fluorescein fluorescence. Results were obtained using three distinct experimental samples.

### Electron paramagnetic resonance (EPR) spectroscopy
EPR spectroscopy with a spin trap was employed to identify the ROS generated by BTP photocatalysis because spin-adducts show various EPR spectra depending on the type of ROS. All EPR spectra were obtained in an aqueous solution at room temperature with 5-tert-butoxycarbonyl-5-methyl-1-pyrroline N-oxide (BMPO). The BTP stock solution (20 mM in DMF) and $H_2O_2$ were added to the BMPO aqueous solution to prepare 1 mM BTP, 10 mM $H_2O_2$, and 10 mM BMPO aqueous solution ($H_2O$:DMF = 95:5, v/v). The solution was irradiated with white room light for 5 min, and then transferred to a capillary EPR tube to measure the EPR spectra. Additionally, a positive control experiment with Fenton reaction was conducted. The stock solutions of $Fe_2SO_4$ and $H_2O_2$ were added to 10 mM BMPO aqueous solution (final concentration: $[Fe_2SO_4] = 1$ mM, $[H_2O_2] = 10$ mM), then the EPR spectrum of this solution was immediately measured to confirm hydroxyl radical generation peaks. EPR measurements were performed at Korea Basic Science Institute (KBSI) in Seoul, Korea. X-band (9.6 GHz) EPR spectra were derived using a Bruker EMX Plus 6/1 spectrometer equipped with a dual-mode cavity (ER 4116DM). The spectra were obtained using the following experimental parameters: microwave frequency, 9.6 GHz; microwave power, 2.9 mW; modulation amplitude, 1 G; time constant, 20.48 ms; 16 scans.

### Hydrogen peroxide assay
The hydrogen peroxide generation by BTP photoactivation was measured by the horseradish peroxidase (Sigma Aldrich, USA) and N,N-diethyl-$p$-phenylenediamine (DPD) (Sigma Aldrich, USA). First, the stock solution of peroxidase (1 mg·mL$^{-1}$ in DI water) and DPD (50 mM in 1 M $H_2SO_4$ aqueous solution) were prepared. Then, the three 50 µM BTP solutions (in normoxic PBS, Ar-bubbled DMSO, and Ar-bubbled PBS) were irradiated by the blue LED ($\lambda_{max} = 450$ nm, 66.7 mW·cm$^{-2}$) (HepatoChem Inc., USA). During the light exposure, assay samples were obtained at 30-minute intervals up to 150 min. The 200 µL of each sample solution was dissolved in the 224 µL DI water, and 80 µL of sodium phosphate buffer (pH 6) was added to each sample solution. Then, 10 µL of DPD and peroxidase stock solution was added. Right after that, the absorbance at 551 nm of each sample solution was measured. The results were obtained from three distinct experimental samples.

### Cell culture
HeLa (CCL-2), PANC-1 (CRL-1469), A549 (CCL-185), and MiaPaca-2 (CRL-1420) cells were purchased from ATCC and grown on the cell culture plates containing 90% Gibco™ DMEM (Thermo Fisher, USA), 10% foetal bovine serum (FBS), 50 units·mL$^{-1}$ of penicillin, and 50 µg·mL$^{-1}$ of streptomycin. The cells were grown at 37 °C in a humidified atmosphere containing 5% $CO_2$. Wild-type and GSDMD$-/-$ immortalised bone-marrow-derived macrophages (iBMDM) were obtained from the James Vince Lab of Walter and Eliza Hall Institute of Medical Research (WEHI). iBMDMs were grown on a cell culture plate with culture media comprising of 90% Gibco™ DMEM (Thermo Fisher, USA), 10% foetal calf serum (FCS), 50 units·mL$^{-1}$ of penicillin, and 50 µg·mL$^{-1}$ of streptomycin. The cells were grown at 37 °C in humidified atmosphere containing 5% $CO_2$.

### Membrane protein expression and purification
The pTrcHisA vector containing the gene for E. coli GlpG membrane protein was transformed into BL21-Gold (DE3) pLysS (Agilent)[30]. A selected colony from the transformed agar plate was used to inoculate 10 ml of Luria-Bertani (LB) medium preculture with ampicillin (100 mg·mL$^{-1}$) and grown overnight at 37 °C. The preculture was added to the 1 L LB medium containing 100 mg × mL$^{-1}$ ampicillin and grown at 37 °C. At an OD600 ≈ 0.7, the cell culture was induced by 0.4 mM Isopropyl β-D-thiogalactoside (IPTG) and further grown at 37 °C for 3 h. Cells were harvested by centrifugation at 5993 × $g$ for 10 mins at 4 °C. The cell pellet was resuspended in 25 mM Tris-HCl (pH 7.4), 150 mM NaCl, 1 mM TCEP, 10% Glycerol, 1 mM PMSF, and then lysed by using Emulsiflex C3 (Avestin) high-pressure homogeniser

( ~ 17000 psi). The cell lysate was mixed with n-dodecyl-β-D-maltoside (DDM) in the final concentration of 1% and kept in slow rotation for 1 h at 4 °C followed by centrifugation at 34811 × g for 30 min at 4 °C. The supernatant was saved, and imidazole was added to the final concentration of 40 mM. For affinity column purification, Ni-IDA resin (Takara Bio) was washed with 1 ml with Tris-HCl (pH 7.4), and 150 mM NaCl was added and incubated for 1 h. The binding solution was loaded to a gravity column and washed with 10 ml of 25 mM Tris-HCl (pH 7.4), 150 mM NaCl, 1 mM TCEP, 0.1% DDM, 10% Glycerol, and 40 mM imidazole three times. The protein sample was then eluted with 25 mM Tris-HCl (pH 7.4), 150 mM NaCl, 1 mM TCEP, 0.1% DDM, 10% glycerol, and 300 mM imidazole. Eluted fractions were mixed, concentrated to ~500 μL with an Amicon 10 K centrifugal filter device (Merck Millipore), and then further purified by size exclusion chromatography (Superdex 200 Increase 10/300 GL, Cytiva). Purified GlpG membrane protein in 25 mM Tris-HCl (pH 7.4), 150 mM NaCl, 1 mM TCEP, 0.1% DDM, and 10% glycerol was stored at −80 °C.

## Membrane protein stability assay

4 μM GlpG was mixed with 100 ~ 150 μM BTP and the mixture was exposed to blue light ($\lambda_{peak}$ = 450 nm, 16.67 mW·cm$^{-2}$) for 30 min. Negative control samples without BTP and/or light were also prepared. 15% SDS-PAGE was used for the analysis, and the reduced gel band intensities for the destabilised protein by irreversible aggregation were quantified and normalised by the negative control with no BTP and no light. The results were obtained from three distinct experimental samples. For thermal denaturation, the sample mixture of GlpG and BTP exposed to the blue light for 30 min was aliquoted in each 15 μL and incubated at various temperatures of 30.0, 45.0, 55.0, 65.0, 77.1, and 89.9 °C for 10 min, followed by cooling to 10 °C in Blue-Ray Biotech Thermal Cycler. The thermal-shocked samples were centrifuged at 17,000 g at 4 °C for 30 min, and then the supernatants were analysed with 15% SDS-PAGE. The gel band intensities for the various temperature conditions were normalised by the one for a normal sample stored at 4 °C. The results were obtained from three distinct experiment samples.

## Single-molecule tweezer assay

The single-molecule forced-unfolding assay was performed on a single-molecule magnetic tweezer apparatus that was custom-built previously described[36]. Sample preparation of DNA-handled GlpG and a robust single-molecule system assembly for the force application was previously reported[57,58]. The lipid bilayer environment for the membrane protein was reconstituted with a bicelle nanostructure, a lipid bilayer disc composed of DMPC lipid and CHAPSO detergent at a 2.5:1 molar ratio. Once singly tethered GlpG is found, repetitive force scanning from 1 pN to 50 pN and then back to 1 pN was applied by moving a pair of magnets toward and back from the sample chamber surface (0.3 mm·s$^{-1}$). Extension change of the molecular construct as a response to force was measured by 3D-tracking the attached magnetic bead. Waiting for 120 secs at 1 pN between every pulling cycle was allowed for the refolding of unfolded GlpG. The negative control experiment was performed in 50 mM Tris (pH 7.5), 150 mM NaCl, and 2.0% bicelle under blue light ($\lambda_{peak}$ = 450 nm, 9.16 mW·cm$^{-2}$). After three to four unfolding/refolding cycles, 10 ~ 20 μM BTP was injected into the sample chamber. A negative control experiment under infrared light ($\lambda_{peak}$ = 850 nm, 39.51 mW·cm$^{-2}$) was also performed. The time durations (BTP injection - permanent damage on protein stability) and unfolding forces were collected and analysed statistically. The results were obtained from distinct experimental samples (n > 15).

## Preparation of tryptic peptides for LC-MS/MS

For LC-MS/MS proteomics, samples (n = 3) were prepared from four groups for comparison. (1) hv − / BTP − : Cells cultured without

light or BTP treatment. (2) hv − /BTP + : Cells were treated with BTP but without light exposure. (3) hv + /BTP − : Cells exposed to 450 nm LED ($\lambda_{max}$ = 450 nm, 16.7 mW·cm$^{-2}$ for 10 min = 10 J·cm$^{-2}$) without BTP treatment. (4) hv + /BTP + : Cells incubated with BTP and exposed to 450 nm LED ($\lambda_{max}$ = 450 nm, 16.7 mW·cm$^{-2}$ for 10 min = 10 J·cm$^{-2}$). HeLa cells were grown in 100 mm cell culture dishes with DMEM supplemented with FBS and antibiotics at 37 °C in a humidified atmosphere containing 5% $CO_2$. For BTP+ conditions, the cultured cells were incubated with 4 μM BTP for 2 h, and the culture medium was exchanged with fresh DMEM before light irradiation. The cells were washed with DPBS and collected using a cell scraper. After a short centrifugation, the cell pellet was lysed using RIPA buffer:protease cocktail inhibitor solution ( = 99:1) (4 °C for 20 min). Cell debris was eliminated from the lysate by centrifugation (16,000 g, 10 min, and 4 °C). On-filter digestion using an S-trapTM mini spin column (PROTIFI, CO2-mini-40) was performed to analyse the whole protein. The protein loading quantity was controlled to 100 μg per sample based on the BCA assay. Low protein-binding microtubes (Eppendorf, Hamburg, Germany) and LC-MS grade solvents were used for all following procedures. The protein suspension (100 μg in 25 μL) was diluted by adding an equal amount of 2 × SDS protein solubilisation buffer (10% SDS, 100 mM triethylammonium bicarbonate, pH 7·8 adjusted with phosphoric acid), followed by three repetitions of 10 s of sonication and 10 s of break cycle. The solution was centrifuged at 13,000 × g for 10 min, and the supernatant was transferred to a new microtube. Reduction and alkylation were performed to prevent the self-crosslinking of cysteine. A total of 12 μL of reduction solution (100 mM dithiothreitol in water) was added to the microtube and heated for 10 min at 95 °C. After cooling for 5 min at RT, 8 μL of 330 mM iodoacetamide was added and incubated for 30 min in the dark. The supernatant was then collected after 13,000 × g of centrifugation for 10 min, followed by sequential addition of 7 μL of 12% phosphoric acid and 479 μL of S-trap binding buffer (90% aqueous methanol containing a final concentration of 100 mM triethylammonium bicarbonate, pH 7.1). The solution was transferred to an S-trap mini spin column and centrifuged at 4000 × g for 30 s. The unbound flow-through was labelled UB. Using the rotator, the S-trap unit was screwed (3 min, 180°), followed by a washing step with 400 μL of S-trap binding buffer and centrifugation at 4000 × g for 30 s. The washing step was repeated three times. The flow-through was labelled as W, and the column unit was transferred to a new microtube. UB and W were maintained at −78 °C in the case of undesired leakage. To digest 100 μg of protein, 5 μg of LC-MS grade trypsin (Promega, #V5280) dissolved in 125 μL of digestion buffer (50 mM Tris) was added to the column. The bottom ejection hole of the column was sealed with parafilm and incubated overnight at 37 °C. After the overnight reaction, parafilm was removed, and the column was moved to a new microtube for the following elution step: 1) centrifugation at 1000 × g for 1 min after adding 80 μL of digestion buffer, 2) centrifugation at 1000 × g for 1 min after adding 80 μL of 0.2% formic acid, and 3) centrifugation at 4000 × g for 1 min after adding 80 μL of 50% acetonitrile/0.2% formic acid solution. The total flow-through was collected and dried using a speed-vac yielding peptide powder. The pH fractionation followed to improve the number of identified proteins.

## High-pH reversed-phase chromatography for peptide fractionation

A Pierce High pH Reversed-Phase Peptide Fractionation Kit (Thermo Scientific, #84868) was used for the chromatography. The spin column was conditioned prior to the elution step. For conditioning, the column was placed in a low-binding microtube and centrifuged at 5000 × g for 2 min after removing the bottom cap to remove the flow-through. The top screw cap was then opened, and the column was filled with 300 μL of acetonitrile. After closing the cap, centrifugation

(3000 × g) was performed for 1 min, and the flow-through was removed. Then, 300 µL of 0.1% trifluoroacetic acid (TFA) solution was added to the column, followed by centrifugation (3000 × g) for 1 min to remove the flow-through. The wash step with 0.1% TFA was repeated. After conditioning the pH fractionation column, the previously prepared peptide powder was dissolved in 300 µL of 0.1% TFA solution and centrifuged in the conditioned column for 1 min at 3000 × g; the flow-through was labelled as FT. The same process was performed for 300 µL of water in a new microtube, which was labelled W. FT and W were maintained at −78 °C in the case of undesired leakage. Further elution steps were performed using different acetonitrile/0.1% triethylamine solutions ranging from 5% to 50% acetonitrile (v/v). Each eluted sample was collected after centrifugation at 3000 × g for 1 min in a new tube. A total of eight fractionated samples were dried using a speed-vac to obtain the peptide powder.

## LC-MS/MS
Dry tryptic peptides were analysed using LC-MS/MS. A Q Exactive Plus Orbitrap mass spectrometer (Thermo Fisher Scientific, MA, USA) incorporated with a nanoelectrospray ion source was used for analyses. A C18 reverse-phase HPLC column (500 mm × 75 µm ID) was used to separate the pure analyte from the crude peptide suspension. An acetonitrile/0.1% formic acid gradient of 2.4%–24% was used as eluent at a flow rate of 300 nL/min. For MS/MS analysis, precursor ion scan MS spectra (m/z 400 −2000) were acquired with an internal lock mass. The 20 most intense ions were isolated via high-energy collision-induced dissociation. The mass spectrometry proteomics data were deposited to the ProteomeXchange Consortium via the PRIDE partner repository with the dataset identifiers PXD038746 and 10.6019/PXD038746.

## LC-MS/MS data processing
All MS/MS samples were analysed using the Sequest Sorcerer platform (Sagen-N Research, San Jose, CA, USA). Sequest was set to search for Homo sapiens (20612 entries, UniProt (http://www.uniprot.org)), which includes frequently observed contaminants assuming the action of digestion enzyme trypsin. Sequest was searched with a fragment ion mass tolerance of 1.00 Da and parent ion tolerance of 10.0 PPM. The carbamidomethyl of cysteine was specified as a fixed modification in Sequest. Oxidation of methionine and acetyl at the N-terminus were specified as variable modifications in Sequest. Scaffold Q+ (version 5.1.0, Proteome Software Inc., Portland, OR) was used to validate MS/MS based peptide and protein identification. A peptide with a probability of higher than 99% for achieving an FDR of lower than 1.0% based on the no Scaffold Local FDR algorithm was accepted as true identification. A protein identification with a probability of higher than 14.0% for achieving an FDR of less than 1.0% and containing two or more identified peptides was accepted. Protein Prophet algorithm[59] was used to calculate the protein probabilities. Proteins that contained similar peptides and could not be differentiated by MS/MS analysis alone were grouped to satisfy the principles of parsimony. The GO annotations for the proteins were retrieved from the NCBI database (downloaded on 11 February 2021). Of the 5173389 spectra in the experiment at the given thresholds, 2120870 (41%) were included in the quantification. The top 3 precursor intensity of peptides aggregated for each protein from the proteomic data was used for label free quantification. The values were $\log_2$-transformed, pruned of those matched to multiple proteins, and non-reproducibly detected values were filled by imputed values representing a normal distribution around the detection limit. A new distribution was created by a Gaussian distribution with a downshift of 1.8 and width of 0.3 standard deviations. All processes were conducted using the Perseus software platform of Max Planck Institute of Biochemistry. As a result, we obtained mass intensities of oxidative modifications for each identified protein. Using these intensities from triplicated control conditions and

an experimental condition, we calculated the P-values and Fold change values. The fold change values for each protein were calculated as 'the average mass intensities of oxidative modifications in the experimental condition divided by the average mass intensities of oxidative modifications in the control conditions' described as:

$$\text{Fold change} = \frac{\bar{I}_{experimental}}{\bar{I}_{controls}}, \tag{1}$$

where $\bar{I}$ is the averaged mass intensities of oxidative modifications.

## Modification search to identify oxidised amino acids
To comprehensively identify peptides including all oxidised amino acids, we employed a multi-stage search strategy where only the spectra unidentified in the first search were searched against the proteins (6889 entries) identified in the first search using a modification search tool, MODplus (v1.02)[60]. The search parameters were as follows: precursor mass tolerance = ± 20 ppm, 13 C errors in precursor mass = −1/0/1/2, fragment mass tolerance = ± 20 ppm, enzyme = trypsin, the number of enzymatic termini = 1, the number of missed cleavages = any, fixed modifications = Carbamidomethyl of cysteine, variable modifications = MS-common modifications provided by MODplus and fast photochemical oxidation of proteins (FPOP)-related modifications (Supplementary Table 1)[61,62], the number of modifications/peptide = any within the modified mass range of −150 to +350 Da, decoy search = 1. All identifications were subsequently rescored by Percolator (v3.06)[63] and validated at an estimated FDR of 1%, resulting in a total of 1,959,844 identifications.

To quantify oxidised amino acids in proteins, we extracted peptides including FPOP-related modifications and aggregated their precursor intensities for each corresponding protein. To evaluate the extent of oxidative modifications excluding methionine oxidation (+ 15.995), the methionine oxidations were excluded from the quantification (methionine di-oxidations were included).

As a result, we obtained the precursor intensities for each protein and for oxidative modification of 17 amino acids (see Supplementary Table 1). Additionally, we represented 'All AAs' by aggregating intensities of oxidative modifications of these 17 amino acids. To compare the degree of oxidation of membrane proteins and soluble proteins for each type of amino acid, we presented the averaged oxidation intensities of 'membrane-specific', 'membrane-cytosolic', and 'cytosolic' proteins for the corresponding amino acids (Fig. 3e). The averaged oxidation intensities of three control conditions were normalised to 1. The fold change values were calculated in the same way above.

## LDH assay and ATP release assay
The LDH-Glo™ Cytotoxicity Assay (Promega J2380, USA), LDH assay kit (ab65393, Abcam), and ATP determination kit (Invitrogen 2409086, USA) were used to detect the release of LDH and ATP. The samples for these assays were prepared in the same manner as that for the ELISA experiment. To set the maximum LDH release control, HeLa cells were lysed with 0.2% Triton X-100 for 15 min, and the medium was used for 100% LDH release. For the LDH assay, HeLa cells were grown with Liproxstatin-1 (10 µM) (Sigma Aldrich SML1414, USA) and z-VAD-fmk (4 µM) (Promega G7231, USA) for 16 h. BTP was then added to the cells (8 µM), followed by incubation for 2 h. After replacing the growth medium with a serum-free medium, the cells treated with the BTP/inhibitor were exposed to blue LED light ($\lambda_{max}$ = 450 nm, 3 J•cm⁻²). Two hours after light exposure, samples were obtained from the media for the LDH assay. For the LDH assay with GSDMD⁻/⁻ iBMDM, WT and GSDMD⁻/⁻ iBMDM were grown on cell culture dishes for one day. iBMDMs were incubated with BTP (8 µM) for 2 h, and then the cells were exposed to blue LED light ($\lambda_{max}$ = 450 nm, 10 J•cm⁻²) after the media was changed. Two hours after irradiation, samples were

obtained from the media for the LDH assay. Results were obtained using three distinct experimental samples.

## Western blot

To investigate the activation of GSDMD and caspases by BTP photo-activation, we conducted western blot analysis with the following antibodies: GSDMD antibody, NBP2-33422 (Novus Biologicals); caspase-1 antibodies, ab207802 (Abcam) and PA5-29342 (Invitrogen); caspase-3 antibody, ab32351 (Abcam); caspase-4 p20 antibody, A94799 (Antibodies); caspase-5 antibody, sc-393346 (Santa Cruz); IL-1β antibody, P420B (Invitrogen); and IL-18 antibody, PA5-79479 (Invitrogen). HeLa, PANC-1, and A549 cells incubated with BTP (10 μM) for 2 h were irradiated with a blue LED ($\lambda_{max}$ = 450 nm, 3 or 10 J·cm$^{-2}$), and the cells were further incubated for 2 h in a serum-free media. After incubation, the media were obtained and concentrated using an Amicon® Ultra-4 Centrifugal Filter, and a HaltTM protease inhibitor cocktail was added. The remaining cells were lysed with RIPA protein extraction solution (50 mM Tris-HCl pH 7.5, 150 mM NaCl, 1% NP-40, 0.5% deoxycholic acid, 0.1% SDS, and 1 mM PMSF) containing HaltTM protease inhibitor cocktail. The lysates and media were stored at –80 °C until further use. For western blotting, the protein concentration of each sample was determined using a BCA Protein Assay (Thermo Fisher 23227, USA). After protein denaturation with SDS-PAGE loading buffer (Biosesang, Republic of Korea) for 7 min at 70 °C, the loaded protein solutions were separated by SDS-PAGE gel electrophoresis, and the separated protein bands were transferred onto a 0.2 μm polyvinylidene difluoride (PVDF) blotting membrane (GE Healthcare, Germany). Proteins on the membrane were blocked with 4% skim milk in 0.1% Tween-20 in Tris-buffered saline (TBST) for an hour. Subsequently, the membrane was incubated with the primary antibodies at a ratio of 1000:1 (500:1 for caspase-5 antibody) in 4% skim milk for 16 h at 4 °C. After washing thrice with TBST (for 10 min each), the membrane was further incubated with the appropriate secondary antibody (3000:1 dilution in TBST), which are listed as follows: anti-mouse HRP (Invitrogen 31430, USA) or anti-rabbit HRP (Abcam ab205718, USA) After washing thrice, western blot chemiluminescence was measured using a ChemiDoc™ MP imaging system (Bio-rad, CA, USA) after development with Clarity reagent (Bio-rad, CA, USA). After the experiment, the antibodies attached to the membrane were eliminated using Restore Western Blot Stripping Buffer (Thermo Fisher, 21059). Western blotting was repeated using an antibody against β-actin (Invitrogen MA5-15739, USA). The experiment was repeated using negative controls (without BTP and/or light exposure).

## Statistics and reproducibility

Statistical data are presented as means ± standard deviation. Origin 2020 and Excel (Microsoft 365 MSO Version 2402) was used to process data. All experimental data from cells were generated from at least three biologically independent experiments.

## Reporting summary

Further information on research design is available in the Nature Portfolio Reporting Summary linked to this article.

## Data availability

The authors declare that the data supporting the findings of this study are available within the article and its Supplementary Information. The protein information utilised in this study is available from the Homo sapiens protein sequence database (20612 entries, UniProt (http://www.uniprot.org)). Raw mass spectrometry dataset used for oxidised proteome analysis have been deposited to the ProteomeXchange Consortium via the PRIDE partner repository under accession code PXD038746 [doi.org/10.6019/PXD038746] and project DOI 10.6019/PXD038746. All other data are provided in the main text, supplementary information, or from the corresponding author upon request. Source data are provided with this paper.

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

## Acknowledgements

This work was supported and funded by the National Research Foundation of Korea (NRF- 2021R1A2C2009504, 2021M3H4A1A03051390, 2021R1F1A1047853, and NRF-2020R1C1C1003937), National Cancer Centre (NCC) (Research Fund HA22C010100), Korea Technology & Information Promotion Agency for SMEs (TIPA) (grant S3198656), New Renewable Energy Core Technology Development Project of the Korea Institute of Energy Technology Evaluation and Planning (KETEP) granted

financial resource from the Ministry of Trade, and Industry & Energy, Republic of Korea (No. 20223030010240). This work also was supported by the Carbon Neutral Institute Research Fund (1.220098.01) of UNIST and Research Fund (1.190147.01) of UNIST. E. H. acknowledges support from Basic Science Research Programme through the NRF funded by the Ministry of Education (2022R1A6A3A13062947–Research Subsidies for Ph.D. Candidates). J. E. V. was funded by a National Health and Medical Research Council (NHMRC) of Australia ideas grant (1183070) and investigator grant (1172929).

## Author contributions

C.L. and T.-H. K. conceived and conceptualised this study. C.L., M. P., and T.-H. K. wrote the manuscript. C.L., M.P., and B. W. contributed equally to this work. C.L. contributed to all aspects of this study in conception, conduction, and analysis of experiments, in particular organic synthesis, spectroscopic analysis, photocatalysis design, ROS assays, confocal and SIM imaging, cell viability assay, and immunoblots. M.P. performed sampling and analysis of oxidised proteome and participated in cytotoxicity tests, hypoxia application, and immunoblots. S. N. analysed oxidised proteome and reviewed proteomics. J.K.S. reviewed and supported all proteomics. B.W. and D.M. conceived and conducted protein stability assays and single-molecule tweezer experiments. C.G.L. performed MTT assays and flowcytometry. E.H. conceived and performed electrochemical analysis for amino acid oxidation. G.Y. and J.K.L. performed organic synthesis and characterisation. C.L. and G.Y. performed ROS assay. J.K.L. conducted TCSPC and electrochemical analysis. D.-H.R. performed cyclic voltammetry. Y.H.K performed MTT assay under hypoxia. J.Y. participated in immunoblots and cell experiments. J.E.V. and S.A.H. generated and provided immortalised WT and GSDMD knock out iBMDMs and reviewed the manuscript. T.-H.K. supervised all aspects of this study.

## Competing interests

The authors declare no competing interests.
