## [Peer Review File · Nature Communications]

Reviewers' Comments:

Reviewer #1:

Remarks to the Author:

Lee et al. study how HeLa cells respond to oxidative damage that originates at intracellular membranes. They develop a photocatalyst, BTP, capable of generating reactive oxygen species upon irradiation with blue light. BTP-generated damage afflicts both lipids and membrane proteins and is localized to ER and mitochondrial membranes. In vitro experiments show that oxidation results in reduced stability of a model membrane protein. Mass spectrometry reveals BTP-induced oxidative damage in membrane proteins (mostly ER, Golgi and mitochondrial membranes) and (to a lesser degree) in cytosolic proteins. A number of protein quality control components showed oxidative damage, which might result in the accumulation of non-functional proteins that were not necessarily direct targets of oxidative damage. Rather, these secondary effects are attributed to maladaptive UPR. Cell culture experiments suggest that BTP causes (non-canonical) pyroptosis, rather than apoptosis, through membrane oxidation.

In the opinion of this reviewer, the development of BTP is a useful advance for studying membrane-associated oxidative stress. The electro-/photo-chemical characterization of BTP look sound, but I leave it to reviewers with expertise in this particular area to scrutinize the data presented here. The magnetic tweezers experiments nicely show that BTP-induced damage lowers the (mechanical) stability of the GlpG model in bicelles irreversibly, suggesting for chemical damage to the protein. The mass spec proteomics experiments are informative and provide a good overview of the effects of the oxidative stress in living cells.

I have some concerns about how convincingly the authors establish that the observed effects are indeed due to membrane-associated damage. Related to that point, I am wondering how solid the interpretation is. Specific concerns are listed below. It would be good if the authors can address them before publication.

Specific points:

I am not convinced that (1) the observed effects can ultimately be attributed to membrane protein damage, and the (2) the (secondary) effects are indeed due to maladaptive UPR, as opposed to non-specific/global damage. While the fraction of membrane proteins show a larger fraction of damaged proteins, the absolute number of damaged cytosolic proteins exceeds that of membrane proteins, if I understand the results correctly. How can the authors rule out that damage to cytosolic proteins is ultimately responsible for cell death?

For the magnetic tweezers experiments, how well does the model protein GlpG in bicelles reflect damage to native proteins in cellular membranes? Do the authors have any indication that this is a good model?

I am wondering how certain the authors can be that it is indeed damage to the protein rather than to the lipids that causes the loss of stability in the single-molecule measurement. A control experiment of injecting fresh bicelles is presented, but it does not seem particularly convincing. What if even a small fraction of damaged lipids reduced stability? And how quickly do lipids exchange between bicelles?

Also, to what degree are the effects seen in the in vitro single-molecule experiments membrane specific? For the stability measurements shown in Fig 2, it would be nice to include a cytosolic protein as a control, in order to address this point.

Can the authors provide any information as to whether the degree and type of damage is uniform? What can be said about the type of damage that occurs, and how pervasive it is?

For the mass spec results, I am surprised that membrane proteins are enriched less than 2-fold in

the oxidized proteome, relative to soluble proteins. Similarly, the enrichment of mitochondrial proteins within the pool of oxidized membrane proteins seems surprising, given that BTP is not localized to mitochondrial membranes initially. Does the damage occur only after the re-distribution of the BTP from the ER to the mitochondrial membrane? What are the time scales?

The experimental procedures should be describe better. It is difficult to find some of the experimental conditions that seem important. For instance, how long were cells irradiated before samples for LC-MS were collected?

Reviewer #2:

Remarks to the Author:

The manuscript titled 'Oxidative photocatalysis on membranes triggers non-canonical pyroptosis' introduces a novel amphiphilic photocatalyst called BTP, designed to be incorporated into cell membranes, where it can initiate oxidation of the membranes and membrane proteins upon exposure to light. The authors extensively characterized this photocatalysis system and employed various approaches to demonstrate that BTP damages intracellular membranes, leading to denaturation of membrane proteins and subsequent lytic cell death. Moreover, they propose that the oxidation of intracellular membrane proteins is responsible for triggering non-canonical pyroptosis. The work presents a comprehensive set of characterization experiments, thoroughly discussed throughout the manuscript. However, there are some concerns regarding the conclusions drawn from the current data, which need to be addressed before the manuscript can be considered suitable for publication in Nature Communications.

Considering my expertise in structural proteomics for membrane proteins, I will provide suggestions for experiment design and data illustration in the sections related to probe design, characterization, and proteomics. However, I may not be the most suitable reviewer for the sections involving cation mobilization and non-canonical inflammasome caspases-induced pyroptosis, given my lack of knowledge in these areas. Hence, my input in those sections will be from the perspective of a broad reader of Nature Communications.

1. Regarding the 'impact of BTP photocatalysis on protein quality control proposed by proteomics' section: The authors justified their focus on analyzing oxidized methionine residues (O-Met) due to methionine's lability under oxidative stress. However, it's essential to consider that within a protein, the oxidation level also depends on the solvent accessible surface area, which may vary for different amino acids. Additionally, Met is naturally present in the form of O-Met in many proteins, and during sample handling, it can easily become oxidized. To strengthen the conclusions, I suggest the following: a. Instead of analyzing only O-Met, the authors should analyze peptides containing all oxidized amino acids. Existing literature, such as works by Rinas et al. and Johnson et al., (Rinas A, Mali SV, Espino AJ, Jones ML, *Anal. Chem.* 2016, 88, 20, 10052–10058. Johnson D, Smith PB, Espino AJ, Gershenson A, Jones ML, *Anal. Chem.* 2020, 92, 2, 1691–1696.) provide suitable methods to identify and quantify all oxidized residues in soluble and membrane proteins in cell, which would be more representative and convincing than focusing solely on oxidized Met. b. Including three controls: (1) 2h lamp irradiation without BTP, (2) no light or BTP, and (3) BTP treatment without light. By subtracting the control data, it would be possible to attribute the observed oxidation specifically to BTP. c. Considering that hydroxyl radicals generated may also diffuse into the cytoplasm, the authors should investigate if the damage occurs predominantly in the membrane or if there are broader effects throughout the cell. d. Regarding the absence of the Met peak in extended Fig.1k, the authors should clarify if this means all the Met residues are oxidized or provide an explanation for the observation.

2. The evidence for oxidative damage on membranes is not sufficiently demonstrated. While the authors mention the oxidation of membrane lipids, they only test phosphatidylcholine (PC) in solution using LC-MS. To verify this concept, I recommend conducting a lipidomic experiment after BTP incubation. Extract the lipids using established protocols and perform HPLC analysis. This simple experiment would provide more direct evidence and could even include a separation of organelle-specific lipids through ultra-centrifugation.

3. The design and significance of the study appear promising, but several conclusions seem to rely on a limited number of targets or observations (e.g., a single protein's unfolding, oxidation of only PC in solution, and focus on O-Met). Given the complexity of cellular systems, it is essential to gather more comprehensive data to establish robust conclusions.

4. Throughout the manuscript, the authors should clearly indicate the effects of 2h 450nm lamp irradiation on the cells and the resulting oxidation.

5. In Fig.1a, the BTP probe is described as having a tiny-COOH head group and a bulky tail with five aromatic rings. It is unclear how this bulky probe could insert itself into cell membranes. I recommend evaluating the effect of BTP on live cells to assess its penetration ability. Assuming BTP can penetrate the membrane and generate reactive oxygen species (ROS) at the membrane interface, the authors should discuss how these polar ROS penetrate the membrane to oxidize unsaturated double bonds, instead of soluble cytoplasmic components.

6. In Extended Fig.1I, to validate the presented oxidation of PC, a MS/MS spectrum should be provided to support the findings.

By addressing these concerns and incorporating the suggested experiments, the manuscript will be better suited for publication in Nature Communications. About the novelty of this work, I feel I am not able to provide insights on it because it is a slightly different field to me.

Reviewer #3:

Remarks to the Author:

The manuscript by Lee et al. reported an amphiphilic organic photocatalyst (BTP) which can be used for photocatalytic membrane oxidation to trigger non-canonical pyroptosis. The synthesized photocatalyst 'BTP' was investigated for the generation of hydroxyl radicals and hydrogen peroxide via water oxidation and the mechanisms behind those processes in detail. Also, non-canonical pyroptosis caused by the oxidation of intracellular membrane proteins were well studied too. Overall, this manuscript is suitable for its publication in Nat. Comm. for the detailed analysis in photocatalyst performance and the convincing non-canonical pyroptosis induction. However, there are still several unclear issues in the manuscript, and minor revision is needed. More detailed comments are listed as follow.

1. As shown in Figure 1f, the results from EPR experiments showed significant differences between BTP + H₂O₂ + hv and another group. However, the EPR spectra did not show the standard spectra shape of hydroxyl radical. Please renew this figure or give an explanation.

2. Authors claim in the manuscript that the BTP can target endomembrane system. However, based merely on observations with the naked eye from both CLSM and SIM images, it is difficult to draw such a conclusion. At least, I cannot draw this conclusion from the data provided by the authors in this manuscript. Although later mentioned by the author that the oxidized proportion of membrane proteins was higher than that of non-membrane proteins, it can only demonstrate that BTP might be prone to enrich in the endomembrane system instead of specific targeting endomembrane system. Unless the authors can provide more compelling evidence, I suggest they consider revising the conclusion they have made in this manuscript.

3. Author mentioned that BTP is able to penetrate the plasma membrane. While from Extended Data Fig 2c, it seems BTP (green fluorescence) can partially localize in the plasma membrane. I think this can be well answered by performing a co-localization experiment between BTP and commercially available plasma membrane dyes.

4. All the co-localization imaging and intracellular ROS generation experiments were investigated by using HeLa cells. Did the GA and ER targeting and following Mito transfer or ROS generation were cell dependent? Please give the explain by experimental results.

5. Similar with Q4, the morphological features of pyroptotic cells and the cleavage of caspase-4/5 in Fig. 5 were investigated by only using HeLa cells. Did the pyroptosis-mediated cell death were cell dependent? Please give the explain by experimental results.

Reviewer #4:

Remarks to the Author:

In this work, Lee and co-workers report an oxidative photocatalyst that can localize cellular membranes and generate hydroxyl radicals upon blue light irradiation under hypoxia. The authors found that light-triggered oxidative stress destabilizes membrane proteins, which consequently induces gasdermin-D-driven pyroptosis. The redox activity of photocatalyst, the mechanism of hydroxyl radical generation, and pyroptotic cell death are well-characterized and very convincing. However, it has not been demonstrated what benefits or new insights can be brought into

biological research by this approach, other than artificially inducing cell death. There is also concern that the organelle membrane selectivity of this photocatalyst seems too low to study local oxidative stress in cells.

Overall, in its current form, I think that the work would not be impressive to many biological readers and does not meet the potential significance and impact required for Nature Communications. It is suggested that the manuscript could be reconsidered to submit other journals after the following points have been addressed:

1. The authors claim that BTP localizes in intracellular membrane (endomembranes). However, this description is not appropriate. Extended Data Fig.2c clearly showed that BTP is also present in the plasma membrane. As the authors mentioned, BTP also translocates to mitochondrial membranes during incubation. Therefore, it is clear that BTP shows virtually no membrane selectivity. The low membrane selectivity is also suggested by proteome analysis.

2. Fig. 3: It is unclear how to calculate fold change for quantitative mass spectrometry. Are the authors quantifying only peptides containing oxidized Met or all peptides to calculate protein abundance? How many peptides per protein are used to calculate abundance? According to Fig.2, the amount of membrane protein should decrease after photooxidation. Why does fold change increase for many proteins?

3. Related to the comment 2, fold change should vary greatly depending on light exposure time. Have the authors performed experiments with different light exposure times? How were the light exposure times determined?

4. The Fig.3 caption states that a total of 314 proteins were detected, but the main text says 313.

5. The authors must demonstrate what kind of new biological insights would be obtained by inducing pyroptosis in a light-dependent manner.

Reviewer #5:

Remarks to the Author:

Lee, Park et al. have devised a clever strategy to cause membrane protein oxidation to trigger non-canonical regulated cell death via caspase 4/5 activation. This could be clinically useful to stop cancer cell proliferation or inflammation driven diseases. The strategy involves photocatalysis via a novel fatty acid like molecule. The strategy is interesting and the fact that it works under hypoxic conditions does make it attractive therapeutically. Could also be an experimental tool to study oxidative stress induced signaling and immune response. The study offers sufficient evidence that BTP photocatalysis works to induce cell death in cell models but offers little in new findings of downstream signaling post increases in oxidative stress. Caspase4/5 activation to GSDMD cleavage is well studied cell death pathway. However, the organic BTP photocatalyst could be a powerful experimental tool and of potential clinical significance and warrants publication. Also, the targeted approach to specifically oxidize membrane proteins is an exciting approach that warrants further study.

Major Points

Figure 4 – Not sure I see evidence of mitochondrial dysfunction(line 214-215 main text). Figure 4a nor 4e is that convincing that mitochondrial dysfunction has occurred. Not sure it really needs to be shown or stated that mitochondrial dysfunction is occurring since BTP photocatalysis induced cell death is evident in other figures of the manuscript. If authors want to prove mitochondrial dysfunction is occurring then the authors should extend the mitotracker experiment (>50s) to show that the calcium overload causes mitochondrial fragmentation or use a membrane potential sensitive dye to show mitochondrial membrane potential dissipates. Could also immunostain for cytochrome c release. At 50s the mitochondrial network in the hela cells looks healthy (4A) even though calcium is being taken up by the mitochondria. Seahorse mitochondrial respiration experiment is also a possibility.

Figure 7c- Authors should consider moving the cancer cell line data (7c) that demonstrates the effectiveness of BTP photocatalysis in a cancer cell line to the main figures. Perhaps end of figure 5 or if the authors have data showing caspase 4/5 induced GSDMD cleavage is the mechanism for decreased cancer cell viability then it could be a new figure.

Minor points

Figure 3- Authors should make more clear in the legend and text what the log2 is comparing (i.e. hv- over hv+ in hela cells)?

Reviewer #1 (Remarks to the Author):

Lee et al. study how HeLa cells respond to oxidative damage that originates at intracellular membranes. They develop a photocatalyst, BTP, capable of generating reactive oxygen species upon irradiation with blue light. BTP-generated damage afflicts both lipids and membrane proteins and is localized to ER and mitochondrial membranes. In vitro experiments show that oxidation results in reduced stability of a model membrane protein. Mass spectrometry reveals BTP-induced oxidative damage in membrane proteins (mostly ER, Golgi and mitochondrial membranes) and (to a lesser degree) in cytosolic proteins. A number of protein quality control components showed oxidative damage, which might result in the accumulation of non-functional proteins that were not necessarily direct targets of oxidative damage. Rather, these secondary effects are attributed to maladaptive UPR. Cell culture experiments suggest that BTP causes (non-canonical) pyroptosis, rather than apoptosis, through membrane oxidation.

In the opinion of this reviewer, the development of BTP is a useful advance for studying membrane-associated oxidative stress. The electro-/photo-chemical characterization of BTP look sound, but I leave it to reviewers with expertise in this particular area to scrutinize the data presented here. The magnetic tweezers experiments nicely show that BTP-induced damage lowers the (mechanical) stability of the GlpG model in bicelles irreversibly, suggesting for chemical damage to the protein. The mass spec proteomics experiments are informative and provide a good overview of the effects of the oxidative stress in living cells.

I have some concerns about how convincingly the authors establish that the observed effects are indeed due to membrane-associated damage. Related to that point, I am wondering how solid the interpretation is. Specific concerns are listed below. It would be good if the authors can address them before publication.

Response:

We sincerely appreciate your thoughtful comments, which have significantly contributed to the improvement of the manuscript. As you noted, this study covers a wide spectrum, ranging from photochemical analysis to biological phenomena such as non-canonical pyroptosis. Therefore, it was challenging to prove all the causal relationships of biological processes strictly. In particular, our statement that 'BTP photocatalysis triggers pyroptosis due to membrane-associated damages' required further validation, considering that O-Met (+16 Da) proteome analysis did not exhibit a dramatic difference between membrane proteins and non-membrane proteins, as you highlighted.

We have revised and supplemented the O-Met analysis by analysing oxidative modifications of 17 amino acids (excluding Gly, Ser, and Thr) based on hydroxyl radical protein footprinting. As a result, we confirmed that the oxidation levels in membrane proteins were significantly higher than those in cytosolic proteins. Furthermore, we have conducted a new investigation into the emission peak shift of BTP, supporting the proximity of BTP molecules to membranes. These results suggest that ROS generation by BTP photocatalysis is focused on intracellular

membranes, leading to specific damage to membrane proteins. Consequently, we now propose that the observed biological processes are induced by the membrane-focused oxidative stress.

Specific points:

Q 1-1.

I am not convinced that (1) the observed effects can ultimately be attributed to membrane protein damage, and the (2) the (secondary) effects are indeed due to maladaptive UPR, as opposed to non-specific/global damage. While the fraction of membrane proteins show a larger fraction of damaged proteins, the absolute number of damaged cytosolic proteins exceeds that of membrane proteins, if I understand the results correctly. How can the authors rule out that damage to cytosolic proteins is ultimately responsible for cell death?

Response 1-1

The O-Met proteome results in the original manuscript showed that BTP photocatalysis oxidised not only membrane proteins but also a considerable amount of non-membrane proteins. We have addressed two major points for a more rigorous analysis of oxidative protein modifications.

First, we have supplemented the criteria for membrane and non-membrane proteins. In the original manuscript, only proteins located at the organelle membrane were classified as ‘membrane-proteins’. Thus, proteins located in the distinctive membranes such as caveolae and annulate lamellae, and ambiguous locations were categorised as ‘non-membrane proteins’. However, some neglected membrane proteins are likely to be proximal to the intracellular membrane, potentially leading to an overestimation of the oxidation levels in ‘non-membrane proteins’. For more accurate interpretation, we have rigorously revised the criteria to (i) membrane-specific proteins, (ii) membrane-cytosol overlapping proteins (*e.g.*, membrane-interacting proteins, peripheral proteins, etc.), and (iii) cytosolic proteins, based on GO annotations from Uniprot (Extended Data Fig. 6).

Second, we have analysed oxidative modifications of 17 amino acids in addition to mono-oxidised methionine (O-Met, +16 Da). Methionine is the most labile amino acid against oxidative stress, implying that proteomics based solely on methionine oxidation could include noise from sample handling. Therefore, other information on protein oxidation was required to identify oxidised proteins more accurately. Accordingly, we have re-analysed oxidised proteins, using hydroxyl radical protein footprinting as the second stage of our modification search (Fig.3).

Using these revised methods, we have obtained much clearer results than those from a single O-Met analysis. The number of identified oxidised membrane proteins was significantly higher than that of oxidised cytosolic proteins, even with stricter criteria for oxidised proteins ($p < 0.01$ and Fold change > 4). Furthermore, the number of oxidised membrane proteins was much

greater than the number of oxidised cytosolic proteins for the 17 amino acids. Based on these results, we have revised the Figures and corresponding paragraphs as follows.

Revised:

Impact of BTP photocatalysis on protein quality control proposed by proteomics

Since methionine is one of the most labile amino acids under oxidative stress, proteins containing oxidised methionine residues (O-Met) were analysed in HeLa cells using label-free quantitative mass spectrometry for an initial screening of BTP oxidation targets (Fig. 3a)^{27,39,40}. The extent of oxidative damage was evaluated for each protein by comparing the average O-Met mass spectra intensities of the experimental groups with those of the control group (Fig. 3a, inset). Proteins that were oxidised more than 2-fold in the experimental group compared to the control group, with p-values lower than 0.05, were considered as 'oxidised proteins' by BTP photocatalysis (p-value <0.05, Fold Change >2). The identified proteins were categorised based on their GO annotation by cellular location, determining whether they were membrane-localised or not (Fig. 3b, Extended Data Fig. 6a). Proteins annotated as being localised to plasma, organelles, or various other membranes were classified as 'membrane-specific'. Cytosolic proteins, excluding the membrane-cytosol overlying proteins, were then compared with the membrane-specific proteins (Fig. 3c). A greater number and proportion of oxidised proteins were observed in membrane-specific proteins compared to cytosolic proteins: 339 versus 40, accounting for 24.5% and 3.6%, respectively. This result supports the membrane-focused oxidative stress, which matches with the membrane-localisation property of BTP. Additionally, the proportions of oxidised membrane proteins of the mitochondria, ER, nucleus, and Golgi apparatus were 31.1%, 19.1%, 21.1%, and 7.7%, respectively (the number of oxidised membrane proteins/the detected number of membrane proteins, Fig. 3d). The global membrane oxidation induced by BTP photocatalysis suggests a potential malfunction in biological processes that require the involvement of various organelles.

To further elucidate other oxidation targets of BTP photocatalysis at the membrane, an in-depth secondary search covering 17 amino acids was conducted using mass spectra not identified in the initial search for O-Met analysis (Fig. 3a). This multistage search allowed us to scrutinise extensive protein oxidations by reducing the search space and thus decreasing the number of false positives. Interestingly, membrane-specific proteins were the most oxidised (oxidised protein criteria: p-value <0.01, Fold Change >4), followed by membrane-cytosolic proteins, and then cytosolic proteins for all detected amino acid residues (Fig 3e, f; Extended Data Fig. 6b, c). It is noteworthy that the number of oxidised proteins, as defined by Trp and His oxidation, was prevalent in membrane-specific proteins (Fig. 3f, Extended Data Fig. 6c). Considering their low abundance in a whole cell, it suggests a favourable interaction between BTP and oxidisable-aromatic amino acids. Detailed information on the oxidised proteome for each amino acid is available in 'Source Data Proteomics' spreadsheet.

Fig. 3 | Comprehensive proteomic profiling of membrane protein oxidation induced by BTP photocatalysis.

a, Schematic illustration of the proteomic analysis workflow used to investigate the extent of oxidative modifications induced by BTP photocatalysis. The process involves a multistage search strategy for identifying O-Met and FPOP (fast photochemical oxidation of proteins) modifications. In the first search, 2,120,870 peptide-spectrum matches (PSMs) were found, and in the second search, 1,959,844 PSMs were found. Samples subjected to BTP photocatalysis and control samples were analysed and compared based on the fold change in the average precursor intensity of oxidative modifications (inset). **b**, Proteins categorized by GO subcellular annotations into 'membrane-specific', located exclusively on membranes, and 'membrane-cytosol', found on both membranes and cytosol. The remaining cytosolic proteins were labelled 'cytosolic' proteins. **c**, Volcano plot of O-Met (+16 Da) proteome showcasing oxidation focused on membrane-specific proteins versus cytosolic proteins. Proteins with a p-value < 0.05 and fold change > 2 were defined as potential oxidation targets of BTP photocatalysis. **d**, The proportions of oxidised membrane proteins across different organelles based on the O-Met proteome. 'Others' include plasma membranes and unidentified locations. **e**, Overview of the 2nd search based on FPOP modifications, showing average oxidation intensities for different amino acids. 'All AAs' represents the aggregated intensities of oxidative modifications of these 17 amino acids. The averaged oxidation intensities of 'membrane-specific', 'membrane-cytosolic', and 'cytosolic' proteins for the corresponding amino acids were presented to compare the degree of oxidation of membrane proteins and soluble proteins for each type of amino acid. The averaged oxidation intensities of three control conditions were normalised to 1. **f**, Volcano plots of the proteome other than O-Met, contrasting membrane-specific proteins with cytosolic proteins. Stricter criteria than sole O-Met analysis (p-value < 0.01 and fold change > 4) were applied for robust identification of oxidised proteins.

Extended Data Fig. 6 | Volcano plots for oxidative modifications of each amino acid. a, Classification of proteins based on Gene Ontology (GO) subcellular annotations: 'Membrane-specific' proteins are exclusively located on membranes, while 'membrane-cytosol' proteins are present on both membranes and in the cytosol. Proteins localized in the cytosol are referred to as 'cytosolic' proteins. **b,** Volcano plots depicting the classification of proteins based on detected MS spectra intensities of oxidative modifications for 17 amino acids. **c,** Comparison of protein oxidation between membrane-specific and cytosolic proteins, focusing on amino acids not highlighted in the main figures.

Revised Method:

Modification search to identify oxidized amino acids.

To comprehensively identify peptides including all oxidized amino acids, we employed a multi-stage search strategy where only the spectra unidentifiable in the first search were searched against the proteins (6,889 entries) identified in the first search using a modification search tool, MODplus (v1.02)⁵⁸. The search parameters were as follows: precursor mass tolerance = ± 20 ppm, ¹³C errors in precursor mass = -1/0/1/2, fragment mass tolerance = ± 20 ppm, enzyme = trypsin, the number of enzymatic termini = 1, the number of missed cleavages = any, fixed modifications = Carbamidomethyl of cysteine, variable modifications = MS-common modifications provided by MODplus and fast photochemical oxidation of proteins (FPOP)-related modifications (Supplementary Table 1)^{59,60}, the number of modifications/peptide = any within the modified mass range of -150 to +350 Da, decoy search = 1. All identifications were subsequently rescored by Percolator (v3.06)⁶¹ and validated at an estimated FDR of 1%, resulting in a total of 1,959,844 identifications.

To quantify oxidized amino acids in proteins, we extracted peptides including FPOP-related modifications and aggregated their precursor intensities for each corresponding protein. To evaluate the extent of oxidative modifications excluding methionine oxidation (+15.995), the methionine oxidations were excluded from the quantification (methionine di-oxidations were included).

As a result, we obtained the precursor intensities for each protein and for oxidative modification of 17 amino acids (see Supplementary Table 1). Additionally, we represented 'All AAs' by aggregating intensities of oxidative modifications of these 17 amino acids. To compare the degree of oxidation of membrane proteins and soluble proteins for each type of amino acid, we presented the averaged oxidation intensities of 'membrane-specific', 'membrane-cytosolic', and 'cytosolic' proteins for the corresponding amino acids (Fig. 3e). The averaged oxidation intensities of three control conditions were normalised to 1. The fold change values were calculated in the same way as described above.

Additionally, we have verified BTP molecular proximity to the membrane by comparing its emission peak in cellular and artificial membrane environments. The emission peak of BTP significantly responds to the hydrophobicity of its surroundings. In a polar environment like water, BTP's emission peak occurs at 630 nm, whereas within a non-polar environment, such as the membrane, it presents a peak at 580 nm, as shown in Extended Data Fig. 3a. Upon introducing artificial lipid bilayers (bicelles) to the BTP aqueous solution, the emission peak shifted from 630 nm to 580 nm (Extended Data Fig. 3b), indicating that BTP is close to hydrophobic lipid bilayers. Subsequently, we measured the emission peak of BTP in cellular environments using Lambda-scan analysis, appearing at 580 nm (Extended Data Fig. 3c). Based on these observations, we infer that BTP resides in close proximity to the intracellular membrane. These results are included in the Extended Data Figure below.

Revised:

We hypothesized that BTP can be especially proximal to intracellular membranes, including the membranes of ER, GA, and mitochondria, considering the amphiphilic BTP structure. To confirm the proximity of BTP to intracellular membranes, we compared its emission peak in cellular and artificial membrane environments (Extended Data Fig. 3a). In a situation where BTP is dissolved in water, the BTP emission peak occurs at 630 nm in a polar environment, whereas within a non-polar environment, such as membrane, it presents a peak at 580 nm (Extended Data Fig. 3a). Upon introducing artificial lipid bilayers (bicelles) to the BTP aqueous solution, the emission peak shifted from 630 nm to 580 nm (Extended Data Fig. 3b). Subsequently, we measured the emission peak of BTP in cellular environments using Lambda-scan analysis, appearing at 580 nm (Extended Data Fig. 3b,c), indicating that BTP molecules are proximal to intracellular membranes. Therefore, these results imply that BTP photocatalysis inside cells generates reactive radicals near intracellular membranes, leading to intracellular membrane-focused oxidative stress.

Extended Data Fig. 3 | Intracellular membrane localisation of BTP and intracellular ROS generation. **a**, A schematic image of BTP fluorescence depending on surrounding environments. **b**, Normalised BTP fluorescence in different conditions. Blue: BTP in 50 mM Tris buffer (pH 7.5, [BTP] = 20 μ M), orange: BTP in 50 mM Tris buffer (pH 7.5, [BTP] = 20 μ M) + bicelles, red: intracellular environment (HeLa cells). The fluorescence of BTP inside cells were obtained by the Lambda-scan mode of confocal microscopy. **c**, Confocal images of BTP were taken at each wavelength, and merged was produced. **d**, H₂DCF-DA assay for intracellular ROS generation. HeLa cells were incubated with BTP (5 μ M) and H₂DCF-DA (20 μ M) and irradiated with blue LED light ($\lambda_{\text{max}} = 450 \text{ nm}$, 5 J·cm⁻²). The green signal corresponds to the DCF fluorescence. **e**, Dot plot analysis of DCF signals from randomly selected 20 cells (BTP+/hv+ and BTP+/hv-) (n = 20). **f**, O₂⁻ assay using dihydroethidium (DHE). HeLa cells were treated with DHE (5 μ M) and exposed to BTP photocatalysis ($\lambda_{\text{max}} = 450 \text{ nm}$, 10 J·cm⁻²). Red signals correspond to DNA-intercalated 2-hydroxyethidium. Live HeLa cells were used for all ROS generation assays. Data are presented as mean \pm s.d. Source data are provided as a Source Data file.

Comprehensively, these results suggest that the observed biological process can mainly be attributed to membrane-focused oxidative damage.

Furthermore, the secondary effects of BTP photocatalysis, including pyroptosis and other biological phenomena, can arise from both maladaptive UPR and global oxidative damage, because the maladaptive UPR and global damages are closely related. Considering that BTP is located in various types of intracellular membranes (ER, Golgi, mitochondria, etc), the oxidative stress caused by BTP can induce global damage although their oxidative stress is specifically localised to the intracellular membrane. However, the global oxidative damages possibly cannot be recovered because of the maladaptive UPR. That is, we believe that the observed biological processes can be induced by both the global accumulation of oxidatively damaged proteins and the initiation of maladaptive UPR.

Q 1-2.

For the magnetic tweezers experiments, how well does the model protein GlpG in bicelles reflect damage to native proteins in cellular membranes? Do the authors have any indication that this is a good model?

Response 1-2:

The helical bundle protein with six transmembrane (TM) helices, GlpG, can serve a good model membrane protein for investigating the oxidative damage effects due to the following reasons. First, helical membrane proteins are the largest class of membrane proteins (Corin, K. & Bowie, J. U. *EMBO Rep.* **2022**, 23, e53025; Shimizu et al, *BBA-Biomembranes* **2018**, 1860, 1077-1091; Vinothkumar et al, *Q. Rev. Biophys.* **2010**, 43, 65-158). Second, GlpG has been widely adopted as a model protein in studies on the folding and stability of helical membrane proteins (Hong et al, *Cur. Opin. Struct. Biol* **2022**, 72, 237; Choi et al. *Science* **2019**, 366, 6469, 1150; Paslawski et al, *Proc. Natl. Acad. Sci. USA* **2015**, 26, 112, 7978; Min et al, *Nat. Chem. Biol.* **2015**, 11, 981). Third, GlpG is monomer and shows reversible unfolding (i.e., unfolding and then refolding back), enabling the application of the single-molecule tweezer approach and the quantitative analyses (Choi et al. *Science* **2019**, 366, 6469, 1150; Min et al, *Nat. Chem. Biol.* **2015**, 11, 981). Overall, GlpG can be regarded as an appropriate model protein for our purpose, in terms of both the structural and methodological perspectives.

At this moment, the bicelles are the best, working membrane mimetics for our in vitro single-molecule assay. The detergent micelles are heterogeneous in stabilizing membrane proteins, while the vesicles and nano-discs can impose limitations on observing a large number of repetitive unfolding events from a single membrane protein (WCB Wijesinghe et al, *J. Mol. Biol.* **2023**, 435, 167975). In contrast, the bicelles, lipid bilayer discs wrapped by detergents, were successfully verified for stabilizing membrane proteins and driving numerous (un) folding events from a single membrane protein (Choi et al. *Science* **2019**, 366, 6469, 1150; Min et al, *Nat. Chem. Biol.* **2015**, 11, 981).

As we know, in vitro experiments, such as in biochemistry, structural biology, and single-molecule biophysics, do not perfectly mimic the cellular environments. The point here is whether we can gain some insights that cannot be obtained from cellular methods. The extent of oxidation effects on individual membrane proteins in cells should vary due to the crowdedness of cellular environments and other complex factors. The single-molecule assay cannot profile this feature, but can identify its potential structural effects at a fine resolution. Based on the results described in the main text, we believe our single-molecule assay effectively serves our purpose, i.e., understanding how the photocatalytic oxidation affects membrane protein stability. We edited the relevant part of the main text, as shown below.

Revised:

Thus, using an *in vitro* membrane protein stability assay (Fig. 2a–d), we examined the effects of BTP photocatalysis on membrane protein folds. We adopted *E. coli* rhomboid protease GlpG consisting of six transmembrane helices, which is a widely studied membrane protein for its folding and stability^{30–32}. The helical bundle protein can serve as an appropriate model for investigating the photocatalytic oxidation effects on helical membrane protein, which represent the largest class in a structural perspective^{33–35}.

Q 1-3.

I am wondering how certain the authors can be that it is indeed damage to the protein rather than to the lipids that causes the loss of stability in the single-molecule measurement. A control experiment of injecting fresh bicelles is presented, but it does not seem particularly convincing. What if even a small fraction of damaged lipids reduced stability? And how quickly do lipids exchange between bicelles?

Also, to what degree are the effects seen in the *in vitro* single-molecule experiments membrane specific? For the stability measurements shown in Fig 2, it would be nice to include a cytosolic protein as a control, in order to address this point.

Response 1-3:

A previous report (Min et al, *Nat. Chem. Biol.* **2015**, 11, 981) demonstrates that GlpG stability is rapidly restored upon injecting fresh bicelles, while its folding reversibility is lost in their absence. The time scale of this bicelle exchange only takes ~1 min. Thus, it is reasonable to expect that upon the bicelle exchange, the intact membrane environment is quickly reconstituted for the membrane protein, leading to the normal (un)folding detection. However, the control experiment shows that once damaged, the protein stability did not return to the normal level, even upon the addition of fresh bicelles (see the statistics added in Extended Data Fig. 5c). This result indicates that irreversible oxidative damage occurred to the membrane protein by the BTP photocatalysis.

The extent of its oxidation before the bicelle exchange is difficult to detect *in situ* in the single-molecule experiments, although DMPC lipids used for bicelle reconstitution are expected to have a relatively higher resistance to oxidation due to the absence of double bonds. However, we would like to emphasize that the purpose of our single-molecule assay was to probe the potential structural effects due to photocatalytic oxidation at a single-molecule resolution. As described in the previous paragraph, the single-molecule assay fulfils its purpose, although it cannot specifically detect possible lipid oxidation. We edited the relevant part of the main text, as shown below.

Revised:

Moreover, the injection of fresh bicelles after C_0 was unable to restore the normal unfolding (Extended Data Fig. 5b) despite the rapid reconstitution of the intact membrane environment from fresh bicelles, which facilitates the reversible unfolding of GlpG³⁰. These results indicate irreversible oxidative damage to the membrane protein, induced by BTP photocatalysis, leading to significant destabilisation of its native protein fold.

The single-molecule assay is not suitable for identifying the membrane specificity either, though BTP shows specificity towards bicelles and intracellular membranes (Extended Data Fig. 3). In cell environments, the main mechanism for the oxidation of membrane protein is associated with its hydrophobic interaction with BTP photosensitiser. Additionally, the single-molecule condition is totally different from the crowded cellular environments containing numerous other components vulnerable to oxidation. Thus, in the single-molecule condition, the ROS stress primarily targets proteins, allowing us to specifically examine their possible oxidation effects at a fine single-molecule resolution, whether the proteins are soluble or not. Moreover, the extent of oxidative damage varies among different proteins; therefore, a simple comparison between a selected membrane protein and a selected soluble protein is limited to draw a conclusion for the membrane specificity. For the above reasons, it is challenging to expect that in the single-molecule assay, soluble proteins are always less likely affected by the BTP photocatalysis. Indeed, the experiments with maltose-binding protein (MBP), a widely adopted soluble protein for folding studies, show the similar oxidative damage and protein destabilisation (Fig. R1). This is probably because the hydrophobic BTP molecules can weakly interact with a hydrophobic region of MBP. Therefore, we should employ another approach for identifying the membrane specificity, as shown in Extended Data Fig. 3 and Fig. 3. The revised proteomic analysis also demonstrates that the oxidation effect is predominant at membranes in cells where BTP is localised.

Fig. R1 | Single-molecule forced unfolding assay with maltose binding protein (MBP). **a**, Representative force-extension curves of MBP under blue light and BTP injection. MBP is a water-soluble model protein that has been widely used in protein folding studies. The blue and red dots point to the unfolding of the external α -helices at the C-terminus at lower forces and the remaining core structure at higher forces, respectively. These signals constitute the characteristic pattern of normal MBP unfolding. **b**, Representative force-extension curves of MBP under blue light and BTP injection. The C_0 and C_1 indicate the pulling cycle number, at which an abnormal unfolding pattern or no detectable unfolding force appears for the first time, respectively. **c**, Unfolding forces observed in each condition ($n = 87, 9,$ and 37). Source data are provided as a Source Data file.

Q 1-4.

Can the authors provide any information as to whether the degree and type of damage is uniform? What can be said about the type of damage that occurs, and how pervasive it is?

Response 1-4:

Reactive radicals such as hydroxyl and superoxide radicals oxidise various biomolecules, producing a variety of oxidised forms. Therefore, there might be different forms of oxidative damage caused by BTP photocatalysis. In particular, hydroxyl radicals are known as highly reactive species that can oxidise most amino acid residues. In the revised manuscript, we have investigated oxidative modifications of 17 amino acids, as we responded above (Fig. 3 and Extended Data Fig. 6). Detailed information is described below (Table R1).

FPOP-related modifications		
Name	Sites	Mass
Oxidation	MFHILVWYADEKNPQR	15.994915
Dioxidation	CFMWY	31.989829
Trioxidation	CFWY	47.984744
Carbonylation	EIKLPQRV	13.979265
Arg deguanidation	R	-43.053433
H to D	H	-22.031969
H to N	H	-23.015984
H - 10	H	-10.0320
H + 5	H	4.9735
Decarboxylation	DE	-30.010565
CO loss	DE	-27.994915
CO2 loss	DE	-43.989829

Table. R1 | Modifications considered during the second search to identify oxidized amino acids. Modifications caused by fast photochemical oxidation of protein (FPOP). The overlapping ones between a and b were only considered once (as FPOP-related modifications) during the search.

These chemical modifications of amino acid residues cause structural destabilisation of proteins, leading to protein misfolding and dysfunctions. However, such damage occurs only in localised spaces where BTP molecules exist, because radical species have very short lifetimes and diffusion lengths. Nonetheless, these damaged proteins can be diffused and accumulated in the ER, accelerating various biological processes, including maladaptive UPR.

Q 1-5.

For the mass spec results, I am surprised that membrane proteins are enriched less than 2-fold in the oxidized proteome, relative to soluble proteins. Similarly, the enrichment of mitochondrial proteins within the pool of oxidized membrane proteins seems surprising, given that BTP is not localized to mitochondrial membranes initially. Does the damage occur only

after the re-distribution of the BTP from the ER to the mitochondrial membrane? What are the time scales?

Response 1-5:

We apologise for the misleading information from proteomic analysis. In this study, ‘fold change’ does not refer to an enrichment of a protein, but rather to the increase in oxidative modifications. However, as you noted, the membrane-focused oxidation was less evident due to the innate noise of label-free quantification and O-Met. Through multi-modifications search of 17 amino acids, we have improved the proteomic evidence with less variance and a stricter threshold than in the original analysis, as we responded above (Fig. 3 and Extended Data Fig. 6).

We are interested in the observation that BTP localisation is changed after photocatalysis. To measure the time scales of the re-distribution, we have conducted time-lapse colocalization imaging (Fig. R2). The results are described as below.

Fig. R2 | Time-lapse colocalization imaging with MitoTracker (red), CellLight™ Golgi RFP BacMam2.0 (blue), and BTP (green). Colocalization patterns are monitored during light irradiation ($\lambda = 450$ nm) using HeLa cells.

The time-lapse images demonstrate that BTP signals (green) are distinct from mitochondrial signals (red) before photocatalysis (0 s). However, one minute after photocatalysis, the BTP signals and mitochondrial signals become completely colocalized. Considering that we irradiated the cells with blue light for 10 minutes in other experiments, the re-distribution appears to occur at an early stage of photocatalysis. Therefore, it seems reasonable that many mitochondrial proteins are oxidised by BTP photocatalysis.

Q 1-6.

The experimental procedures should be describe better. It is difficult to find some of the experimental conditions that seem important. For instance, how long were cells irradiated before samples for LC-MS were collected?

Response 1-6:

We apologise for this issue. We revised and complemented the method section of proteomics including sampling and data analysis. For the LC-MS/MS sampling for proteomics, we irradiated blue light (450 nm) for 10 minutes (10 J·cm²).

Reviewer #2 (Remarks to the Author):

The manuscript titled 'Oxidative photocatalysis on membranes triggers non-canonical pyroptosis' introduces a novel amphiphilic photocatalyst called BTP, designed to be incorporated into cell membranes, where it can initiate oxidation of the membranes and membrane proteins upon exposure to light. The authors extensively characterized this photocatalysis system and employed various approaches to demonstrate that BTP damages intracellular membranes, leading to denaturation of membrane proteins and subsequent lytic cell death. Moreover, they propose that the oxidation of intracellular membrane proteins is responsible for triggering non-canonical pyroptosis. The work presents a comprehensive set of characterization experiments, thoroughly discussed throughout the manuscript. However, there are some concerns regarding the conclusions drawn from the current data, which need to be addressed before the manuscript can be considered suitable for publication in Nature Communications.

Considering my expertise in structural proteomics for membrane proteins, I will provide suggestions for experiment design and data illustration in the sections related to probe design, characterization, and proteomics. However, I may not be the most suitable reviewer for the sections involving cation mobilization and non-canonical inflammasome caspases-induced pyroptosis, given my lack of knowledge in these areas. Hence, my input in those sections will be from the perspective of a broad reader of Nature Communications.

Q 2-1.

Regarding the 'impact of BTP photocatalysis on protein quality control proposed by proteomics' section: The authors justified their focus on analyzing oxidized methionine residues (O-Met) due to methionine's lability under oxidative stress. However, it's essential to consider that within a protein, the oxidation level also depends on the solvent accessible surface area, which may vary for different amino acids. Additionally, Met is naturally present in the form of O-Met in many proteins, and during sample handling, it can easily become oxidized. To strengthen the conclusions, I suggest the following:

a. Instead of analyzing only O-Met, the authors should analyze peptides containing all oxidized amino acids. Existing literature, such as works by Rinas et al. and Johnson et al., (Rinas A, Mali SV, Espino AJ, Jones ML, *Anal. Chem.* 2016, 88, 20, 10052–10058. Johnson D, Smith PB, Espino AJ, Gershenson A, Jones ML, *Anal. Chem.* 2020, 92, 2, 1691–1696.) provide suitable methods to identify and quantify all oxidized residues in soluble and membrane proteins in cell, which would be more representative and convincing than focusing solely on oxidized Met.

b. Including three controls: (1) 2h lamp irradiation without BTP, (2) no light or BTP, and (3) BTP treatment without light. By subtracting the control data, it would be possible to attribute the observed oxidation specifically to BTP.

c. Considering that hydroxyl radicals generated may also diffuse into the cytoplasm, the authors should investigate if the damage occurs predominantly in the membrane or if there are broader effects throughout the cell.

d. Regarding the absence of the Met peak in extended Fig.1k, the authors should clarify if this means all the Met residues are oxidized or provide an explanation for the observation.

Response 2-1:

We express our gratitude for your kind and detailed review of our manuscript. All your constructive comments have improved our manuscript. We have performed additional experiments and analyses for all the points you raised.

(A): As you highlighted, methionine can be naturally oxidised in cellular environments and also become oxidised during sample preparation. This naturally or preparation-induced oxidised methionine enhances the noise, leading to potential over- or underestimation and large variance. Therefore, we completely agree with your concerns about the necessity of analysing oxidative modifications of all amino acids.

We have re-analysed LC-MS/MS raw data for 17 amino acids, excluding Gly, Ser, and Thr, as referenced by fast photochemical oxidation of protein (FPOP) footprinting. We have applied a multi-stage search strategy using the modification search tool MODplus (Na et al. *Anal. Chem.* **2019**, 91, 17, 11324). Only the spectra unidentified in the first search were re-examined against the identified protein list from the first search (6,889 entries). This method allowed us to exclude non-specific signals derived from labile O-Met (+16 Da). To further attenuate undesired O-Met noise, mono-oxidation of methionine newly searched by MODplus was also excluded. The FPOP-related modifications selected for our second search are listed in Supplementary Table 1.

a MS-common modifications			b FPOP-related modifications		
Name	Sites	Mass	Name	Sites	Mass
Oxidation	MWPF	15.994915	Oxidation	MFHILWYADEKNPQR	15.994915
Dioxidation	MWC	31.989829	Dioxidation	CFMWY	31.989829
Dethiomethyl	M	-48.003371	Trioxidation	CFWY	47.984744
Trp->Kynurenin	W	3.994915	Carbonylation	EIKLPQRV	13.979265
Deamidated	NQ	0.984016	Arg deguanidation	R	-43.053433
Carbamyl	N-term, K	43.005814	H to D	H	-22.031969
Carbamidomethyl	N-term, KH	57.021464	H to N	H	-23.015984
Phospho	STY	79.966331	H - 10	H	-10.0320
Dehydrated	STD	-18.010565	H + 5	H	4.9735
Ammonia-loss	N	-17.026549	Decarboxylation	DE	-30.010565
Glu->pyro-Glu	N-term E	-18.010565	CO loss	DE	-27.994915
Gln->pyro-Glu	N-term Q	-17.026549	CO2 loss	DE	-43.989829
Formyl	N-term, ST	27.994915			
Acetyl	Prot-N-term, K	42.010565			
Methyl	DEKHR	14.01565			
Dimethyl	KR	28.0313			
Cation:Na	DE	21.981943			
Cation:K	DE	37.955882			
Cation:Fe[II]	DE	53.919289			
Iodo	Y	125.896648			
AEBS	KY	183.035399			
Trioxidation	C	47.984744			
Cys->Dha	C	-33.987721			
Pyro-carbamidomethyl	N-term C	39.994915			
CarbamidomethylDTT	C	209.018035			

Supplementary Table 1 | Modifications considered during the second search to identify oxidized amino acids. **a.** MS-common modifications provided by MODplus **b.** Modifications caused by cell FPOP. The overlapping ones between **a** and **b** were only considered once (as FPOP-related modifications) during the search.

Using these revised methods, we obtained more obvious results indicating that BTP photocatalysis triggers intracellular membrane-specific oxidative stress. For most amino acids, the oxidation level of membrane proteins was significantly higher than that of cytosolic proteins, even when applying more conservative criteria for oxidised proteins ($p < 0.05$, Fold Change $> 2 \rightarrow p < 0.01$, Fold Change > 4). Based on these results, we have revised the Figures and corresponding paragraphs as follows.

Revised:

Impact of BTP photocatalysis on protein quality control proposed by proteomics

Since methionine is one of the most labile amino acids under oxidative stress, proteins containing oxidised methionine residues (O-Met) were analysed in HeLa cells using label-free quantitative mass spectrometry for an initial screening of BTP oxidation targets (Fig. 3a)^{27,39,40}. The extent of oxidative damage was evaluated for each protein by comparing the average O-Met mass spectra intensities of the experimental groups with those of the control group (Fig. 3a, inset). Proteins that were oxidised more than 2-fold in the experimental group compared to the control group, with p-values lower than 0.05, were considered as 'oxidised proteins' by BTP photocatalysis (p -value < 0.05 , Fold Change > 2). The identified proteins were categorised based on their GO annotation by cellular location, determining whether they were membrane-localised or not (Fig. 3b, Extended Data Fig. 6a). Proteins annotated as being localised to plasma, organelles, or various other membranes were classified as 'membrane-specific'. Cytosolic proteins, excluding the membrane-cytosol overlying proteins, were then compared with the membrane-specific proteins (Fig. 3c). A greater number and proportion of oxidised proteins were observed in membrane-specific proteins compared to cytosolic proteins: 339 versus 40, accounting for 24.5%

and 3.6%, respectively. This result supports the membrane-focused oxidative stress, which matches with the membrane-localisation property of BTP. Additionally, the proportions of oxidised membrane proteins of the mitochondria, ER, nucleus, and Golgi apparatus were 31.1%, 19.1%, 21.1%, and 7.7%, respectively (the number of oxidised membrane proteins/the detected number of membrane proteins, Fig. 3d). The global membrane oxidation induced by BTP photocatalysis suggests a potential malfunction in biological processes that require the involvement of various organelles.

To further elucidate other oxidation targets of BTP photocatalysis at the membrane, an in-depth secondary search covering 17 amino acids was conducted using mass spectra not identified in the initial search for O-Met analysis (Fig. 3a). This multistage search allowed us to scrutinise extensive protein oxidations by reducing the search space and thus decreasing the number of false positives. Interestingly, membrane-specific proteins were the most oxidised (oxidised protein criteria: p-value <0.01, Fold Change >4), followed by membrane-cytosolic proteins, and then cytosolic proteins for all detected amino acid residues (Fig 3e, f; Extended Data Fig. 6b, c). It is noteworthy that the number of oxidised proteins, as defined by Trp and His oxidation, was prevalent in membrane-specific proteins (Fig. 3f, Extended Data Fig. 6c). Considering their low abundance in a whole cell, it suggests a favourable interaction between BTP and oxidisable-aromatic amino acids. Detailed information on the oxidised proteome for each amino acid is available in ‘Source Data Proteomics’ spreadsheet.

Fig. 3 | Comprehensive proteomic profiling of membrane protein oxidation induced by BTP photocatalysis. **a**, Schematic illustration of the proteomic analysis workflow used to investigate the extent of oxidative modifications induced by BTP photocatalysis. The process involves a multistage search strategy for identifying O-Met and FPOP (fast photochemical oxidation of proteins) modifications. In the first search, 2,120,870 peptide-spectrum matches (PSMs) were found, and in the second search, 1,959,844 PSMs were found. Samples subjected to BTP photocatalysis and control samples were analysed and compared based on the fold change in the average precursor intensity of oxidative modifications (inset). **b**, Proteins categorized by GO subcellular annotations into

'membrane-specific', located exclusively on membranes, and 'membrane-cytosol', found on both membranes and cytosol. The remaining cytosolic proteins were labelled 'cytosolic' proteins. **c**, Volcano plot of O-Met (+16 Da) proteome showcasing oxidation focused on membrane-specific proteins versus cytosolic proteins. Proteins with a p-value <0.05 and fold change >2 were defined as potential oxidation targets of BTP photocatalysis. **d**, The proportions of oxidised membrane proteins across different organelles based on the O-Met proteome. 'Others' include plasma membranes and unidentified locations. **e**, Overview of the 2nd search based on FPOP modifications, showing average oxidation intensities for different amino acids. 'All AAs' represents the aggregated intensities of oxidative modifications of these 17 amino acids. The averaged oxidation intensities of 'membrane-specific', 'membrane-cytosolic', and 'cytosolic' proteins for the corresponding amino acids were presented to compare the degree of oxidation of membrane proteins and soluble proteins for each type of amino acid. The averaged oxidation intensities of three control conditions were normalised to 1. **f**, Volcano plots of the proteome other than O-Met, contrasting membrane-specific proteins with cytosolic proteins. Stricter criteria than sole O-Met analysis (p-value < 0.01 and fold change > 4) were applied for robust identification of oxidised proteins.

Extended Data Fig. 6 | Volcano plots for oxidative modifications of each amino acid. a, Classification of proteins based on Gene Ontology (GO) subcellular annotations: 'Membrane-specific' proteins are exclusively located on membranes, while 'membrane-cytosol' proteins are present on both membranes and in the cytosol. Proteins localized in the cytosol are referred to as 'cytosolic' proteins. **b**, Volcano plots depicting the classification of proteins based on detected MS spectra intensities of oxidative modifications for 17 amino acids. **c**, Comparison of protein oxidation between membrane-specific and cytosolic proteins, focusing on amino acids not highlighted in the main figures.

Revised Method:

Modification search to identify oxidized amino acids.

To comprehensively identify peptides including all oxidized amino acids, we employed a multi-stage search strategy where only the spectra unidentified in the first search were searched against the proteins (6,889 entries) identified in the first search using a modification search tool, MODplus (v1.02)⁵⁸. The search parameters were as follows: precursor mass tolerance = ± 20 ppm, ^{13}C errors in precursor mass = $-1/0/1/2$, fragment mass tolerance = ± 20 ppm, enzyme = trypsin, the number of enzymatic termini = 1, the number of missed cleavages = any, fixed modifications = Carbamidomethyl of cysteine, variable modifications = MS-common modifications provided by MODplus and fast photochemical oxidation of proteins (FPOP)-related modifications (Supplementary Table 1)^{59,60}, the number of modifications/peptide = any within the modified mass range of -150 to $+350$ Da, decoy search = 1. All identifications were subsequently rescored by Percolator (v3.06)⁶¹ and validated at an estimated FDR of 1%, resulting in a total of 1,959,844 identifications.

To quantify oxidized amino acids in proteins, we extracted peptides including FPOP-related modifications and aggregated their precursor intensities for each corresponding protein. To evaluate the extent of oxidative modifications excluding methionine oxidation ($+15.995$), the methionine oxidations were excluded from the quantification (methionine di-oxidations were included).

As a result, we obtained the precursor intensities for each protein and for oxidative modification of 17 amino acids (see Supplementary Table 1). Additionally, we represented 'All AAs' by aggregating intensities of oxidative modifications of these 17 amino acids. To compare the degree of oxidation of membrane proteins and soluble proteins for each type of amino acid, we presented the averaged oxidation intensities of 'membrane-specific', 'membrane-cytosolic', and 'cytosolic' proteins for the corresponding amino acids (Fig. 3e). The averaged oxidation intensities of three control conditions were normalised to 1. The fold change values were calculated in the same way as described above.

Furthermore, the functions of the oxidised membrane proteins determined by the new method were closely related to ER-/mitochondria-mediated protein quality control. Consequently, the newly adopted method strongly supported our claim that BTP photocatalysis triggers intracellular membrane-focused oxidative stress. We have revised the Figure and corresponding paragraphs as follows.

Revised:

Furthermore, the 'membrane-specific' and 'membrane-cytosolic' proteins were collected to evaluate the total extent of oxidation (Fig. 4a). We selected 250 oxidised proteins as our proteome of interest, applying a conservative threshold (p-value <0.01 , Fold Change >4 ; Fig. 4b). These 250 oxidised membrane proteins were predominantly located in the ER, mitochondria, nucleus, and GA, corresponding with the initial O-Met screening (Fig. 4c). Notably, the ER, GA, and mitochondria—three organelles crucial for protein quality control (PQC)⁴¹—possessed 58.1% of the oxidised membrane proteins resulting from BTP photocatalysis. We therefore focused on PQC-related functions, such as unfolded protein response (UPR) and protein transport. Among 250 oxidised membrane proteins, 97 proteins were categorised into four functional networks, i) UPR and ER-associated degradation (ERAD), ii) ER–Golgi transport, iii) mitochondrial trafficking and transport, and iv) lipid metabolism (Fig. 4d).

UPR and ERAD are apparently key quality control mechanisms necessary to alleviate stress from the accumulation of misfolded proteins⁴². Additionally, ER–Golgi transport is a process that facilitates ER quality control⁴³. Moreover, mitochondrial trafficking and transport are crucial for mitochondrial functions, which mediate mitochondrial UPR^{44,45}, another axis of the cellular quality control mechanism. Lipid metabolism is also important, as an imbalance in lipid homeostasis can stimulate UPR⁴⁶. Furthermore, clustering the 250 oxidised proteins by Gene Ontology (GO) biological process and assessing GO enrichment scores demonstrated the oxidative damage imposed by BTP on biological processes related to cellular quality control via the ER and mitochondrial unfolded protein response (Fig. 4e). As described in the folding stability experiments (Fig. 2), the

significantly oxidised membrane proteins might have lost their folding stability. Therefore, we hypothesised that BTP-induced oxidation and the resulting dysfunction of proteins could deteriorate PQC and escalate UPR stress.

Fig. 4 | Functional implication of oxidised membrane proteins by BTP photocatalysis. **a**, Protein quality control (PQC) related proteins was highlighted on the volcano plot of membrane proteins by their functional categories. Oxidised proteins were defined by the strict criteria (p -value < 0.01 , fold change > 4), based on the intensity of oxidative modifications of all amino acids excluding Gly, Ser, Thr, and O-Met (+16 Da). **b**, The count and percentage of proteins satisfying the oxidation criteria. **c**, Distribution of oxidised membrane proteins across cellular organelles. ‘Others’ include plasma membranes and unidentified locations. **d**, Heatmap comparison of highlighted protein oxidation between experimental and control conditions. **e**, String network of the strictly defined oxidised proteins ($p < 0.01$, $FC > 4$), filtered for high interaction confidence (0.9) illustrated with GO biological processes and GO enrichment scores. Node size reflects $\log_2 FC$ values, and disconnected nodes were excluded from the network. Key PQC-related processes were marked in blue.

(B): We apologise for the confusion regarding the experimental and control conditions. In the supplementary information of the original manuscript, we noted that there were three control

conditions (BTP+hv⁻, BTP-hv⁺, and BTP-hv⁻) for proteome analysis. As shown in Fig. R3, we averaged the intensities of the collected oxidised peptides identified in the three control conditions (BTP+hv⁻, BTP-hv⁺, and BTP-hv⁻). Fold Change values were calculated as ‘the intensity of oxidised peptide identified in experimental condition over the averaged intensity of oxidised peptide identified in control conditions’. Using this method, we estimated O-Met level compared with their basal oxidation. Additionally, the multi-modifications search used in the revised analysis enabled improved assignments of our oxidation targets. In the revised manuscript, we have moved the detailed methods from Supplementary information to Method section. Furthermore, we have added visualised information about how the Fold Change values are calculated to Fig. 3.

Fig. R3 | Conditions of samples prepared for proteomic analysis. Fold change of oxidative modifications was evaluated based on average intensity of collected oxidised peptides per protein with or without BTP photocatalysis.

Revised:

Preparation of tryptic peptides for LC-MS/MS.

For LC-MS/MS proteomics, samples were prepared from four groups for comparison. (1) hv⁻/BTP⁻: Cells cultured without light or BTP treatment. (2) hv⁻/BTP⁺: Cells were treated with BTP but without light exposure. (3) hv⁺/BTP⁻: Cells exposed to 450 nm LED ($\lambda_{max} = 450$ nm, 16.7 mW·cm⁻² for 10 minutes = 10 J·cm⁻²) without BTP treatment. (4) hv⁺/BTP⁺: Cells incubated with BTP and exposed to 450 nm LED ($\lambda_{max} = 450$ nm, 16.7 mW·cm⁻² for 10 minutes = 10 J·cm⁻²). HeLa cells were grown in 100 mm cell culture dishes with DMEM supplemented with FBS and antibiotics at 37°C in a humidified atmosphere containing 5% CO₂. For BTP⁺ conditions, the cultured cells were incubated with 4 μM BTP for 2 hours, and the culture medium was exchanged with fresh DMEM before light irradiation. The cells were washed with DPBS and collected using a cell scraper. After a short centrifugation, the cell pellet was lysed using RIPA buffer:protease cocktail inhibitor solution (=99:1) (4°C for 20 min). Cell debris was eliminated from the lysate by centrifugation (16,000 g, 10 min, and 4°C). On-filter digestion using an S-trapTM mini spin column (PROTIFI, CO2-mini-40) was performed to analyse the whole protein. The protein loading quantity was controlled to 100 μg per sample based on the BCA assay. Low protein-binding microtubes (Eppendorf, Hamburg, Germany) and LC-MS grade solvents were used for all following procedures. The protein suspension (100 μg in 25 μL) was diluted by adding an equal amount of 2× SDS protein solubilization buffer (10% SDS, 100 mM triethylammonium bicarbonate, pH 7-8 adjusted with phosphoric acid), followed by three repetitions of 10 s of sonication and 10 s of break cycle. The solution was centrifuged at 13,000×g for 10 min, and the supernatant was transferred to a new microtube. Reduction and alkylation were performed to prevent the self-crosslinking of cysteine. A total of 12 μL of reduction solution (100 mM dithiothreitol in water) was added to the microtube and heated for 10 min at 95°C. After cooling for 5 min at RT, 8 μL of 330 mM iodoacetamide was added and incubated for 30 min in the dark. The supernatant was then collected after 13,000×g of centrifugation for 10 min, followed by sequential addition of 7 μL of 12% phosphoric acid and 479 μL of S-trap binding buffer (90% aqueous methanol containing a final concentration of 100 mM triethylammonium bicarbonate, pH 7.1). The solution was transferred to an S-trap mini spin column and

centrifuged at 4000×g for 30 s. The unbound flow-through was labelled UB. Using the rotator, the S-trap unit was screwed (3 min, 180°), followed by a washing step with 400 µL of S-trap binding buffer and centrifugation at 4000×g for 30 s. The washing step was repeated three times. The flow-through was labelled as W, and the column unit was transferred to a new microtube. UB and W were maintained at -78°C in the case of undesired leakage. To digest 100 µg of protein, 5 µg of LC-MS grade trypsin (Promega, #V5280) dissolved in 125 µL of digestion buffer (50 mM Tris) was added to the column. The bottom ejection hole of the column was sealed with parafilm and incubated overnight at 37°C. After the overnight reaction, parafilm was removed, and the column was moved to a new microtube for the following elution step: 1) centrifugation at 1000×g for 1 min after adding 80 µL of digestion buffer, 2) centrifugation at 1000×g for 1 min after adding 80 µL of 0.2% formic acid, and 3) centrifugation at 4000×g for 1 min after adding 80 µL of 50% acetonitrile/0.2% formic acid solution. The total flow-through was collected and dried using a speed-vac yielding peptide powder. The pH fractionation followed to improve the number of identified proteins.

LC-MS/MS data processing.

All MS/MS samples were analysed using Sequest Sorcerer platform (Sagen-N Research, San Jose, CA, USA). Sequest was set to search for Homo sapiens (20612 entries, UniProt (<http://www.uniprot.org>)), which includes frequently observed contaminants assuming the action of digestion enzyme trypsin. Sequest was searched with a fragment ion mass tolerance of 1.00 Da and parent ion tolerance of 10.0 PPM. The carbamidomethyl of cysteine was specified as a fixed modification in Sequest. Oxidation of methionine and acetyl at the N-terminus were specified as variable modifications in Sequest. Scaffold Q+ (version 5.1.0, Proteome Software Inc., Portland, OR) was used to validate MS/MS based peptide and protein identification. A peptide with a probability of higher than 99% for achieving an FDR of lower than 1.0% based on the no Scaffold Local FDR algorithm was accepted as true identification. A protein identification with a probability of higher than 14.0% for achieving an FDR of less than 1.0% and containing two or more identified peptides was accepted. Protein Prophet algorithm⁵⁷ was used to calculate the protein probabilities. Proteins that contained similar peptides and could not be differentiated by MS/MS analysis alone were grouped to satisfy the principles of parsimony. The GO annotations for the proteins were retrieved from the NCBI database (downloaded on 11 February 2021). Of the 5173389 spectra in the experiment at the given thresholds, 2120870 (41%) were included in the quantification. The top 3 precursor intensity of peptides aggregated for each protein from the proteomic data was used for label free quantification. The values were log₂-transformed, pruned of those matched to multiple proteins, and non-reproducibly detected values were filled by imputed values representing a normal distribution around the detection limit. A new distribution was created by a Gaussian distribution with a downshift of 1.8 and width of 0.3 standard deviations. All processes were conducted using the Perseus software platform of Max Planck Institute of Biochemistry. As a result, we obtained mass intensities of oxidative modifications for each identified protein. Using these intensities from triplicated control conditions and an experimental condition, we calculated the p-values and Fold change values. The fold change values for each protein were calculated as 'the average mass intensities of oxidative modifications in the experimental condition / the average mass intensities of oxidative modifications in the control conditions' as follows.

$Fold\ change = \frac{I_{experimental}}{I_{controls}}$, where I = Averaged mass intensities of oxidative modifications.

(C): As you noted, reactive oxygen species (ROS) are diffusive. However, hydroxyl radicals have extremely short diffusion lengths and lifetimes in cell environments (*BMC Veterinary Research* **2021**, 17, 226; *Am. J. Med.* **1991**, 91, 31). Therefore, we expected that oxidative damage occurs only in localised spaces where BTP molecules exist. In the original manuscript, the O-Met proteome analysis showed that BTP photocatalysis mainly oxidised membrane proteins. However, a considerable amount of non-membrane proteins was also detected. By investigating FPOP modifications, the revised results suggest that BTP photocatalysis predominantly induces intracellular membrane-focused protein oxidation, as described above (Fig. 3 and Extended Data Fig. 6).

Furthermore, we have verified BTP molecular proximity to the membrane by comparing its emission peak in cellular and artificial membrane environments. The emission peak of BTP significantly responds to the hydrophobicity of its surroundings. In a polar environment like water, BTP's emission peak occurs at 630 nm, whereas within a non-polar environment, such as the membrane, it presents a peak at 580 nm, as shown in Extended Data Fig. 3a. Upon introducing artificial lipid bilayers (bicelles) to the BTP aqueous solution, the emission peak shifted from 630 nm to 580 nm (Extended Data Fig. 3b), indicating that BTP is close to hydrophobic lipid bilayers. Subsequently, we measured the emission peak of BTP in cellular environments using Lambda-scan analysis, appearing at 580 nm (Extended Data Fig. 3c). Based on these observations, we infer that BTP resides in close proximity to the intracellular membrane. These results are included in the Extended Data Figure below.

Revised:

We hypothesized that BTP can be especially proximal to intracellular membranes, including the membranes of ER, GA, and mitochondria, considering the amphiphilic BTP structure. To confirm the proximity of BTP to intracellular membranes, we compared its emission peak in cellular and artificial membrane environments (Extended Data Fig. 3a). In a situation where BTP is dissolved in water, the BTP emission peak occurs at 630 nm in a polar environment, whereas within a non-polar environment, such as membrane, it presents a peak at 580 nm (Extended Data Fig. 3a). Upon introducing artificial lipid bilayers (bicelles) to the BTP aqueous solution, the emission peak shifted from 630 nm to 580 nm (Extended Data Fig. 3b). Subsequently, we measured the emission peak of BTP in cellular environments using Lambda-scan analysis, appearing at 580 nm (Extended Data Fig. 3b,c), indicating that BTP molecules are proximal to intracellular membranes. Therefore, these results imply that BTP photocatalysis inside cells generates reactive radicals near intracellular membranes, leading to intracellular membrane-focused oxidative stress.

Extended Data Fig. 3 | Intracellular membrane localisation of BTP and intracellular ROS generation. **a**, A schematic image of BTP fluorescence depending on surrounding environments. **b**, Normalised BTP fluorescence in different conditions. Blue: BTP in 50 mM Tris buffer (pH 7.5, [BTP] = 20 μ M), orange: BTP in BTP in 50 mM Tris buffer (pH 7.5, [BTP] = 20 μ M) + bicelles, red: intracellular environment (HeLa cells). The fluorescence of BTP inside cells were obtained by the Lambda-scan mode of confocal microscopy. **c**, Confocal images of BTP were taken at each wavelength, and merged was produced. **d**, H₂DCF-DA assay for intracellular ROS generation. HeLa cells were incubated with BTP (5 μ M) and H₂DCF-DA (20 μ M) and irradiated with blue LED light ($\lambda_{\text{max}} = 450$ nm, 5 J·cm⁻²). The green signal corresponds to the DCF fluorescence. **e**, Dot plot analysis of DCF signals from randomly selected 20 cells (BTP+/hv+ and BTP+/hv-) (n = 20). **f**, O₂⁻ assay using dihydroethidium (DHE). HeLa cells were treated with DHE (5 μ M) and exposed to BTP photocatalysis ($\lambda_{\text{max}} = 450$ nm, 10 J·cm⁻²). Red signals correspond to DNA-intercalated 2-hydroxyethidium. Live HeLa cells were used for all ROS generation assays. Data are presented as mean \pm s.d. Source data are provided as a Source Data file.

(D): There was a mistake in presenting our results. In the experiments, BTP could not oxidise all methionine, thereby we should show the complete data, including the natural methionine peaks. We have reconstructed the figure by adding the missing peaks, and found that only 10% of the total methionine was oxidised under those conditions. The figure has been revised as follows.

Revised:

Extended Data Fig. 1 | Photophysical properties of BTP and photocatalytic oxidation reactions. **a**, Absorption and fluorescence spectra of the aqueous BTP solution. **b**, Cyclic voltammetry curves of BTP and blank

at a scan rate of 10 mV/s in a three-electrode system. **c**, Ground and excited redox potential of BTP. **d**, Proposed scheme of electron transfer for reductive and oxidative quenching cycles. **e**, Time-correlated single-photon counting (TCSPC) spectra of BTP in acetonitrile containing various amounts of H₂O (0, 5, and 10% H₂O). **f**, ·OH generation assay using HPF. The results indicated a change in ·OH generation with irradiation energy (blue LED, $\lambda_{\text{max}} = 450 \text{ nm}$, $16.6 \text{ mW}\cdot\text{cm}^{-2}$) ($n = 3$). **g**, A three-electrode system for measuring the photocurrent by photocatalytic amino acid oxidation. (See supplementary information) **h**, Transient photocurrent responses depending on amino acids. Light exposure was turned on and off every 10 s and repeated. **i**, Relative photocurrent for each amino acid. **j**, ¹O₂ generation assay using ABDA. The absorbance of ABDA ($\lambda = 400 \text{ nm}$) was normalised to that observed under non-irradiated condition, and the decrease in the absorbance represents ¹O₂ generation. Methylene blue was used as a positive control. ($n = 3$) **k**, Methionine oxidation by BTP photocatalysis. Oxidised methionine (O-Met) was detected by HRMS. The inserted image exhibits isotopic distribution of [O-Met + H]. [BTP] = 100 μM , [Met] = [PC] = 1 mM. A blue LED lamp was used to excite BTP ($\lambda_{\text{max}} = 450 \text{ nm}$, $16.6 \text{ mW}\cdot\text{cm}^{-2}$ for 2 h). Data are presented as mean \pm s.d. ($n = 3$). Source data are provided as a Source Data file.

Q 2-2.

The evidence for oxidative damage on membranes is not sufficiently demonstrated. While the authors mention the oxidation of membrane lipids, they only test phosphatidylcholine (PC) in solution using LC-MS. To verify this concept, I recommend conducting a lipidomic experiment after BTP incubation. Extract the lipids using established protocols and perform HPLC analysis. This simple experiment would provide more direct evidence and could even include a separation of organelle-specific lipids through ultra-centrifugation.

Response 2-2:

We agree with your concern that more direct evidence should be provided to verify lipid oxidation induced by BTP photocatalysis. As you suggested, we have performed ultra-performance liquid chromatography-mass spectrometry (UPLC-MS) analysis by extracting total lipids from HeLa cells with or without BTP photocatalysis. The UPLC results imply that the chromatogram of experimental groups (BTP+, hv+) is almost the same as that under control conditions (BTP-, hv-). Furthermore, 15:0-18:1 PC was identified using an internal standard (SPLASH® LIPIDOMIX® Mass Spec Standard), and its oxidation forms were searched. However, the elevated oxidation of 15:0-18:1 PC was not detected after BTP photocatalysis.

These results suggest that lipid oxidation induced by BTP photocatalysis is possibly inefficient in cell environment. As you noted in Q2-5, the penetration of polar radical species into the hydrophobic region of membranes could be limited. The unsaturated double bonds of lipids are hidden in the hydrophobic region, while the oxidisable part of membrane proteins is present on the membrane interface. Therefore, ROS generated on membrane interface would preferentially oxidise membrane proteins rather than unsaturated lipids. Additionally, other experiments in this study indicated that BTP photocatalysis does not induce ferroptosis, and Liproxatin-1 (a lipid peroxidation inhibitor) cannot inhibit BTP photocatalysis-induced cell death. This result is reasonable if lipid oxidation caused by BTP photocatalysis is relatively inefficient compared to membrane protein oxidation. Therefore, we have deleted the indirect evidence for intracellular lipid oxidation (*in vitro* PC oxidation experiment and *in cell* fluorometric lipid peroxidation assays) from our manuscript. We have revised the Figure and corresponding paragraphs as follows.

Revised:

Furthermore, we found that BTP photocatalysis generated substantial membrane oxidative stress. The dichlorodihydrofluorescein diacetate (DCFH₂-DA, a ROS indicator) assay showed that BTP photocatalysis increased DCF fluorescence in HeLa cells (Extended Data Fig. 3d, e; Supplementary Fig. 10). Additionally, we established BTP photocatalysis-induced generation of O₂⁻ using a dihydroethidium (an O₂⁻ sensor) assay (Extended Data Fig. 3f). These results indicate that BTP photocatalysis induces oxidative stress on intracellular membranes.

Additionally, we investigated cellular lipid oxidation caused by BTP photocatalysis (Extended Data Fig. 4), since lipids are the main component of bio-membranes. Cellular lipids were extracted from HeLa cells with and without BTP photocatalysis, and subsequently analysed using ultra-performance liquid chromatography-mass spectrometry (UPLC-MS). Interestingly, we could not detect any newly generated peaks in the chromatogram following BTP photocatalysis (Extended Data Fig. 4a). Furthermore, the UPLC-MS results indicated that the oxidation ratio of 15:0-18:1 phosphatidylcholine (PC) was not changed after BTP photocatalysis (Extended Data Fig. 4b). These results suggest that the lipid oxidation caused by BTP photocatalysis might be inefficient in cellular environments. This inefficiency could be due to the limited diffusion of polar hydroxyl radicals into the hydrophobic regions within the lipid bilayer.

Extended Data Fig. 4 | Lipid oxidation analysis using ultra-performance liquid chromatography-MS (UPLC-MS). **a**, Base peak chromatograms of lipid extracts from HeLa cells (BTP+/hv+ and BTP-/hv-) for ESI positive and negative mods. The HeLa cells were incubated with BTP (5 μ M) and irradiated with blue LED light ($\lambda_{\text{max}} = 450 \text{ nm}$, $10 \text{ J}\cdot\text{cm}^{-2}$), then total lipids of the cells were extracted using Folch's method. The lipid extracts were analysed by UPLC-MS. To identify 15:0-18:1 phosphatidylcholine (PC), SPLASH LipidoMIXTM Internal Standard was added to the lipid extracts. **b**, The relative abundance of oxidised 15:0-18:1 PC and the molecular

structure of 15:0-18:1 PC. Relative abundances are calculated as $A_{O-PC}/(A_{O-PC}+A_{PC})$, where A_{O-PC} represents the retention time area of oxidised 15:0-18:1 PC and A_{PC} represents the retention time area of native 15:0-18:1 PC ($n = 2$). Data are presented as mean \pm s.d. Source data are provided as a Source Data file.

Q 2-3.

The design and significance of the study appear promising, but several conclusions seem to rely on a limited number of targets or observations (e.g., a single protein's unfolding, oxidation of only PC in solution, and focus on O-Met). Given the complexity of cellular systems, it is essential to gather more comprehensive data to establish robust conclusions.

Response 2-3:

We completely understand your point that our conclusion should be established by more comprehensive data. In particular, the conclusion that BTP photocatalysis triggers membrane-specific oxidative stress was based on a limited number of experimental results. But for this part, we believe that the evidence for this has been strengthened with the analysis of oxidative modifications of 17 amino acids (Fig. 3 and Extended Data Fig. 6) and the BTP emission shift analysis (Extended Data Fig. 3).

However, we still understand that a significant number of biological processes and their causal relationships remain unexplored in the mechanism by which membrane-focused oxidative stress triggers pyroptosis. Nevertheless, we believe that researchers can be inspired by the observations we report here, and these insights can be applied in many fields. Therefore, we have added the limitations of our studies in the conclusion part. Our further research aims to reveal more definitive mechanisms. The revised conclusion is described below.

Revised:

We propose that photocatalytic membrane oxidation triggers non-canonical pyroptosis using the amphiphilic organic photocatalyst, BTP (Extended Data Fig. 9). Via photocatalysis, BTP generates highly oxidising $\bullet OH$ in a spatiotemporally controlled manner even under hypoxic conditions, thereby damaging the structural stability of membrane proteins. The single-molecule tweezer approach verified that BTP photocatalysis disrupts membrane protein folding. Using the oxidised proteome from the label-free quantification, we found that BTP photocatalysis substantially oxidised PQC-related membrane proteins of ER, GA, and mitochondria in cells. Disruption of the folding stability and oxidation of PQC-related proteins seemed to stimulate the accumulation of misfolded proteins, followed by ER stress, maladaptive UPR, and cation mobilisation. These cellular responses consequently triggered the caspase-4/5-induced GSDMD cleavage and subsequent pyroptosis.

Since pyroptosis is known to generate the most robust immune response, recent studies have focused on various triggers for this cell death pathway. Pyroptosis is usually caused by microbial infection or endotoxins such as LPS. However, we suggest that the intracellular membrane-focused oxidative stress can trigger pyroptosis through non-canonical inflammasome activation. This endotoxin-independent mechanism implies an alternative pathway for inducing pyroptosis. Although the full spectrum of biological processes and their causal relationships remain unexplored in this study, we believe that this study can inspire further research into the pathogenesis of immune-related diseases. Additionally, light-controlled pyroptosis can be useful to induce immune responses spatiotemporally. In particular, BTP photocatalysis induces pyroptosis even in a hypoxic environment, suggesting that this strategy can be therapeutically attractive, considering that most cancers have a hypoxic environment. Consequently, we hope that this method can be widely used to spatiotemporally induce caspase-4/5 activation and pyroptosis in pathogenesis studies and clinical applications.

Q 2-4:

Throughout the manuscript, the authors should clearly indicate the effects of 2h 450nm lamp irradiation on the cells and the resulting oxidation.

Response 2-4:

We apologise for the confusion regarding the experimental conditions and believe there may be some misleading about this issue. Cells were irradiated with blue light for no more than 10 minutes (450 nm LED), while only *in vitro* samples (Met and PC) were irradiated for 2 hours. The results of the Intracellular ROS assay confirmed that irradiation for 10 minutes had no significant effect on cells. We have revised the irradiation condition of 'Method' section. ($10 \text{ J}\cdot\text{cm}^{-2} \rightarrow 16.7 \text{ mW}\cdot\text{cm}^{-2}$ for 10 minutes) as described above (Method section: Preparation of tryptic peptides for LC-MS/MS).

Q 2-5.

In Fig.1a, the BTP probe is described as having a tiny-COOH head group and a bulky tail with five aromatic rings. It is unclear how this bulky probe could insert itself into cell membranes. I recommend evaluating the effect of BTP on live cells to assess its penetration ability. Assuming BTP can penetrate the membrane and generate reactive oxygen species (ROS) at the membrane interface, the authors should discuss how these polar ROS penetrate the membrane to oxidize unsaturated double bonds, instead of soluble cytoplasmic components.

Response 2-5:

To determine how BTP molecules penetrate plasma membranes, we investigated BTP uptake under various physiological conditions: at 37 °C, at 4 °C, and in the presence of NaN₃. The experimental results suggest that the BTP does not penetrate the plasma membrane efficiently under conditions of 4 °C and in the presence of NaN₃. This implies that the penetration mechanism of BTP into cells depends on energy-consuming processes (*Nat. Chem.* **2022** 14, 274-283). These experimental results have been added to the revised manuscript.

Revised:

Given the molecular structure of BTP which is composed of several lipophilic aromatic rings and a hydrophilic carboxylic acid, the passive diffusion of BTP across plasma membranes can be limited. To explore the cellular uptake of BTP, we investigated its uptake under physiological conditions at 37 °C, at 4 °C, and in the presence of NaN₃. The confocal microscopy results show that the uptake of BTP by HeLa cells dramatically decreased under conditions of 4 °C and NaN₃, implying that the penetration mechanism of BTP across plasma membranes relies on energy-consuming processes (Extended Data Fig. 2a).

Extended Data Fig. 2 | Co-localisation imaging. **a**, BTP uptake under physiological conditions at 37 °C, at 4 °C, and in the presence of NaN_3 . HeLa cells were pre-incubated at the conditions for 30 minutes, then further incubated with BTP (10 μM) for 2 hours. **b**, Co-localisation of BTP (5 μM). HeLa cells were transfected with CellLight™ Golgi RFP BacMam2.0 and Sec61b-mGFP constructs to stain the Golgi apparatus and endoplasmic reticulum (ER), respectively. MitoTracker™ Deep Red FM was used to stain mitochondria. Fluorescence signals of GolgiRFP, Sec61b-mGFP, and MitoTracker are represented in red. **c**, Changes in location of BTP after photocatalysis. HeLa cells were stained with BTP and MitoTracker and imaged before and after light exposure (confocal laser, $\lambda = 445 \text{ nm}$). Enlarged images of white boxes show that BTP moves to the mitochondrial membrane during photocatalysis. **d**, Live-structured illumination microscopy (live-SIM) images before and after BTP photocatalysis ($\lambda = 445 \text{ nm}$). The green and red signals correspond to BTP and MitoTracker, respectively. **e**, Co-localisation of BTP with a plasma membrane staining dye. HeLa cells were stained with BTP (10 μM for 2 hours) and CellMask™ Deep Red Plasma Membrane Stain (5 $\mu\text{g}/\text{mL}$ for 10 minutes). The HeLa cells were imaged before and after light exposure (confocal laser, $\lambda = 445 \text{ nm}$). Live HeLa cells were used for all co-localisation imaging. Data are presented as mean \pm s.d. Source data are provided as a Source Data file.

The unsaturated double bonds of lipids are hidden in the hydrophobic region, while the oxidisable part of membrane proteins is present on the membrane interface. Therefore, ROS generated on membrane interface would preferentially oxidise membrane proteins rather than unsaturated lipids. Additionally, we believe that membrane proteins are predominantly oxidised by BTP photocatalysis compared to cytosolic proteins due to the short lifetimes of hydroxyl radicals. Specifically, the UPLC-MS results imply that lipid oxidation induced by

BTP photocatalysis is possibly inefficient in cell environment, which aligns with your hypothesis.

Q 2-6.

In Extended Fig.1I, to validate the presented oxidation of PC, a MS/MS spectrum should be provided to support the findings.

Response 2-6:

In the original manuscript, we conducted the PC oxidation experiments in a vial. However, PC evenly distributed in solution is much more oxidisable than in the lipid bilayer structure because the unsaturated double bonds are freely exposed. Therefore, we believe that Extended Fig. 1I cannot directly support the intracellular lipid oxidation after BTP photocatalysis. Furthermore, the UPLC-MS results suggest that intracellular lipid oxidation might be inefficient in the cell environment. Consequently, we have removed the Extended Fig. 1I to avoid misleading readers (The revised Extended Fig. 1 is shown above).

By addressing these concerns and incorporating the suggested experiments, the manuscript will be better suited for publication in Nature Communications. About the novelty of this work, I feel I am not able to provide insights on it because it is a slightly different field to me.

Reviewer #3 (Remarks to the Author):

The manuscript by Lee et al. reported an amphiphilic organic photocatalyst (BTP) which can be used for photocatalytic membrane oxidation to trigger non-canonical pyroptosis. The synthesized photocatalyst 'BTP' was investigated for the generation of hydroxyl radicals and hydrogen peroxide via water oxidation and the mechanisms behind those processes in detail. Also, non-canonical pyroptosis caused by the oxidation of intracellular membrane proteins were well studied too. Overall, this manuscript is suitable for its publication in Nat. Comm. for the detailed analysis in photocatalyst performance and the convincing non-canonical pyroptosis induction. However, there are still several unclear issues in the manuscript, and minor revision is needed. More detailed comments are listed as follow.

Q 3-1.

As shown in Figure 1f, the results from EPR experiments showed significant differences between BTP + H₂O₂ + hv and another group. However, the EPR spectra did not show the standard spectra shape of hydroxyl radical. Please renew this figure or give an explanation.

Response 3-1:

We sincerely appreciate your constructive comments. Your insights have significantly contributed to the improvement of our manuscript.

We have added the EPR spectrum for the Fenton reaction to Figure 1f as a positive control. Hydroxyl radical generation is well known, and the EPR peaks were consistent with the experimental conditions of BTP (Fig. 1f). Therefore, we have concluded that the peaks we measured corresponded to BMPO-hydroxyl radicals, which align with the interpretations of references (*J. Phys. Chem. A* **2016**, 120, 18, 2815; *Eur. J. Med. Chem.* **2011**, 46, 4, 1348; *Environ. Sci. Technol.* **2020**, 54, 10, 6415). The additional control experiments are shown in the ‘source data’ file. We have added these results to the Fig. 1.

Revised:

Fig. 1 | Photocatalytic cycles of BTP for inducing pyroptosis via membrane oxidation. **a**, Amphiphilic molecular structure of BTP. **b**, Photocatalytic cycles of BTP. The right circle represents the reductive quenching cycle that produces H_2O_2 and $\cdot\text{OH}$, and the left circle represents oxidative quenching cycle that induces $\text{O}_2^{\cdot-}$ generation and amino acid oxidation. This oxidative photocatalysis triggers non-canonical pyroptosis. The inserted cell image shows the pyroptotic morphology of HeLa cells with BTP ($5 \mu\text{M}$) photocatalysis. **c**, Quenching of the fluorescence of BTP by photoinduced electron transfer from H_2O . The spectra represent the variation in BTP fluorescence with % H_2O (0–12%) in acetonitrile. **d**, H_2O_2 generation assay with DPD and horseradish peroxidase. BTP ($50 \mu\text{M}$) in normoxic PBS, Ar-bubbled PBS (hypoxic PBS), and DMSO were irradiated by the

blue LED ($\lambda_{\max} = 450 \text{ nm}$, $66.7 \text{ mW}\cdot\text{cm}^{-2}$) for 150 minutes. At 30 min intervals, the change in absorbance at 551 nm was measured to indicate H_2O_2 generation. (n = 2). e, $\cdot\text{OH}$ generation assay with HPF. The results represent HPF fluorescence measured under various conditions (See method section) with/without light exposure (blue LED, $\lambda_{\max} = 450 \text{ nm}$, $2 \text{ J}\cdot\text{cm}^{-2}$) (n = 3). f, Electron paramagnetic resonance (EPR) spectroscopy with 10 mM BMPO. A spectrum of $\cdot\text{OH}$ spin adduct, BMPO-OH, was observed after BTP photocatalysis with H_2O_2 ([BTP] = 1 mM, $[\text{H}_2\text{O}_2] = 10 \text{ mM}$) and Fenton reaction as positive control ($[\text{Fe}_2\text{SO}_4] = 1 \text{ mM}$, $[\text{H}_2\text{O}_2] = 10 \text{ mM}$). Data are presented as mean \pm s.d. (n = 2-3). * $P < 0.0005$; ** $P < 0.05$. Student's two-tailed t -test. Source data are provided as a Source Data file.

Q 3-2.

Authors claim in the manuscript that the BTP can target endomembrane system. However, based merely on observations with the naked eye from both CLSM and SIM images, it is difficult to draw such a conclusion. At least, I cannot draw this conclusion from the data provided by the authors in this manuscript. Although later mentioned by the author that the oxidised proportion of membrane proteins was higher than that of non-membrane proteins, it can only demonstrate that BTP might be prone to enrich in the endomembrane system instead of specific targeting endomembrane system. Unless the authors can provide more compelling evidence, I suggest they consider revising the conclusion they have made in this manuscript.

Response 3-2:

We completely agree with your concern that our conclusion should be established by more compelling data. In particular, the conclusion that BTP photocatalysis triggers membrane-targeted oxidation was based on a limited number of experimental results. Given that the ROS is diffusive in cell environment, we have revised the word 'membrane-targeted' to 'membrane-focused'. Furthermore, we have performed additional experiments and analysis to support that BTP photocatalysis induces membrane-focused oxidative stress and membrane protein-specific oxidation.

First, we have verified BTP molecular proximity to the membrane by comparing its emission peak in cellular and artificial membrane environments. The emission peak of BTP significantly responds to the hydrophobicity of its surroundings. In a polar environment like water, BTP's emission peak occurs at 630 nm, whereas within a non-polar environment, such as the membrane, it presents a peak at 580 nm, as shown in Extended Data Fig. 3a. Upon introducing artificial lipid bilayers (bicelles) to the BTP aqueous solution, the emission peak shifted from 630 nm to 580 nm (Extended Data Fig. 3b), indicating that BTP is close to hydrophobic lipid bilayers. Subsequently, we measured the emission peak of BTP in cellular environments using Lambda-scan analysis, appearing at 580 nm (Extended Data Fig. 3c). Based on these observations, we infer that BTP resides in close proximity to the intracellular membrane. These results are included in the Extended Data Figure below.

Revised:

We hypothesized that BTP can be especially proximal to intracellular membranes, including the membranes of ER, GA, and mitochondria, considering the amphiphilic BTP structure. To confirm the proximity of BTP to

intracellular membranes, we compared its emission peak in cellular and artificial membrane environments (Extended Data Fig. 3a). In a situation where BTP is dissolved in water, the BTP emission peak occurs at 630 nm in a polar environment, whereas within a non-polar environment, such as membrane, it presents a peak at 580 nm (Extended Data Fig. 3a). Upon introducing artificial lipid bilayers (bicelles) to the BTP aqueous solution, the emission peak shifted from 630 nm to 580 nm (Extended Data Fig. 3b). Subsequently, we measured the emission peak of BTP in cellular environments using Lambda-scan analysis, appearing at 580 nm (Extended Data Fig. 3b,c), indicating that BTP molecules are proximal to intracellular membranes. Therefore, these results imply that BTP photocatalysis inside cells generates reactive radicals near intracellular membranes, leading to intracellular membrane-focused oxidative stress.

Extended Data Fig. 3 | Intracellular membrane localisation of BTP and intracellular ROS generation. **a**, A schematic image of BTP fluorescence depending on surrounding environments. **b**, Normalised BTP fluorescence in different conditions. Blue: BTP in 50 mM Tris buffer (pH 7.5, [BTP] = 20 μ M), orange: BTP in BTP in 50 mM Tris buffer (pH 7.5, [BTP] = 20 μ M) + bicelles, red: intracellular environment (HeLa cells). The fluorescence of BTP inside cells were obtained by the Lambda-scan mode of confocal microscopy. **c**, Confocal images of BTP were taken at each wavelength, and merged was produced. **d**, H₂DCF-DA assay for intracellular ROS generation. HeLa cells were incubated with BTP (5 μ M) and H₂DCF-DA (20 μ M) and irradiated with blue LED light ($\lambda_{\text{max}} = 450$ nm, 5 J·cm⁻²). The green signal corresponds to the DCF fluorescence. **e**, Dot plot analysis of DCF signals from randomly selected 20 cells (BTP+/hv+ and BTP+/hv-) (n = 20). **f**, O₂⁻ assay using dihydroethidium (DHE). HeLa cells were treated with DHE (5 μ M) and exposed to BTP photocatalysis ($\lambda_{\text{max}} = 450$ nm, 10 J·cm⁻²). Red signals correspond to DNA-intercalated 2-hydroxyethidium. Live HeLa cells were used for all ROS generation assays. Data are presented as mean \pm s.d. Source data are provided as a Source Data file.

Second, we have analysed oxidative modifications of 17 amino acids in addition to mono-oxidised methionine (O-Met, +16 Da). Methionine is the most labile amino acid against oxidative stress, implying that proteomics based solely on methionine oxidation could include

noise from sample handling. Therefore, other information on protein oxidation was required to identify oxidised proteins more accurately. Accordingly, we have re-analysed oxidised proteins, using hydroxyl radical protein footprinting as the second stage of our modification search.

Using these revised methods, we have obtained much clearer results than those from a single O-Met analysis. The number of identified oxidised membrane proteins was significantly higher than that of oxidised cytosolic proteins, even with stricter criteria for oxidised proteins ($p < 0.01$ and Fold change > 4). Furthermore, the number of oxidised membrane proteins was much greater than the number of oxidised cytosolic proteins for the 17 amino acids. Based on these results, we have revised the Figures and corresponding paragraphs as follows. Comprehensively, these results indicate that BTP photocatalysis triggers membrane-focused oxidative damages.

Revised:

Impact of BTP photocatalysis on protein quality control proposed by proteomics

Since methionine is one of the most labile amino acids under oxidative stress, proteins containing oxidised methionine residues (O-Met) were analysed in HeLa cells using label-free quantitative mass spectrometry for an initial screening of BTP oxidation targets (Fig. 3a)^{27,39,40}. The extent of oxidative damage was evaluated for each protein by comparing the average O-Met mass spectra intensities of the experimental groups with those of the control group (Fig. 3a, inset). Proteins that were oxidised more than 2-fold in the experimental group compared to the control group, with p-values lower than 0.05, were considered as 'oxidised proteins' by BTP photocatalysis (p -value < 0.05 , Fold Change > 2). The identified proteins were categorised based on their GO annotation by cellular location, determining whether they were membrane-localised or not (Fig. 3b, Extended Data Fig. 6a). Proteins annotated as being localised to plasma, organelles, or various other membranes were classified as 'membrane-specific'. Cytosolic proteins, excluding the membrane-cytosol overlying proteins, were then compared with the membrane-specific proteins (Fig. 3c). A greater number and proportion of oxidised proteins were observed in membrane-specific proteins compared to cytosolic proteins: 339 versus 40, accounting for 24.5% and 3.6%, respectively. This result supports the membrane-focused oxidative stress, which matches with the membrane-localisation property of BTP. Additionally, the proportions of oxidised membrane proteins of the mitochondria, ER, nucleus, and Golgi apparatus were 31.1%, 19.1%, 21.1%, and 7.7%, respectively (the number of oxidised membrane proteins/the detected number of membrane proteins, Fig. 3d). The global membrane oxidation induced by BTP photocatalysis suggests a potential malfunction in biological processes that require the involvement of various organelles.

To further elucidate other oxidation targets of BTP photocatalysis at the membrane, an in-depth secondary search covering 17 amino acids was conducted using mass spectra not identified in the initial search for O-Met analysis (Fig. 3a). This multistage search allowed us to scrutinise extensive protein oxidations by reducing the search space and thus decreasing the number of false positives. Interestingly, membrane-specific proteins were the most oxidised (oxidised protein criteria: p -value < 0.01 , Fold Change > 4), followed by membrane-cytosolic proteins, and then cytosolic proteins for all detected amino acid residues (Fig 3e, f; Extended Data Fig. 6b, c). It is noteworthy that the number of oxidised proteins, as defined by Trp and His oxidation, was prevalent in membrane-specific proteins (Fig. 3f, Extended Data Fig. 6c). Considering their low abundance in a whole cell, it suggests a favourable interaction between BTP and oxidisable-aromatic amino acids. Detailed information on the oxidised proteome for each amino acid is available in 'Source Data Proteomics' spreadsheet.

Fig. 3 | Comprehensive proteomic profiling of membrane protein oxidation induced by BTP photocatalysis. **a**, Schematic illustration of the proteomic analysis workflow used to investigate the extent of oxidative modifications induced by BTP photocatalysis. The process involves a multistage search strategy for identifying O-Met and FPOP (fast photochemical oxidation of proteins) modifications. In the first search, 2,120,870 peptide-spectrum matches (PSMs) were found, and in the second search, 1,959,844 PSMs were found. Samples subjected to BTP photocatalysis and control samples were analysed and compared based on the fold change in the average precursor intensity of oxidative modifications (inset). **b**, Proteins categorized by GO subcellular annotations into 'membrane-specific', located exclusively on membranes, and 'membrane-cytosol', found on both membranes and cytosol. The remaining cytosolic proteins were labelled 'cytosolic' proteins. **c**, Volcano plot of O-Met (+16 Da) proteome showcasing oxidation focused on membrane-specific proteins versus cytosolic proteins. Proteins with a p-value < 0.05 and fold change > 2 were defined as potential oxidation targets of BTP photocatalysis. **d**, The proportions of oxidised membrane proteins across different organelles based on the O-Met proteome. 'Others' include plasma membranes and unidentified locations. **e**, Overview of the 2nd search based on FPOP modifications, showing average oxidation intensities for different amino acids. 'All AAs' represents the aggregated intensities of oxidative modifications of these 17 amino acids. The averaged oxidation intensities of 'membrane-specific', 'membrane-cytosolic', and 'cytosolic' proteins for the corresponding amino acids were presented to compare the degree of oxidation of membrane proteins and soluble proteins for each type of amino acid. The averaged oxidation intensities of three control conditions were normalised to 1. **f**, Volcano plots of the proteome other than O-Met, contrasting membrane-specific proteins with cytosolic proteins. Stricter criteria than sole O-Met analysis (p-value < 0.01 and fold change > 4) were applied for robust identification of oxidised proteins.

Extended Data Fig. 6 | Volcano plots for oxidative modifications of each amino acid. a, Classification of proteins based on Gene Ontology (GO) subcellular annotations: 'Membrane-specific' proteins are exclusively located on membranes, while 'membrane-cytosol' proteins are present on both membranes and in the cytosol. Proteins localized in the cytosol are referred to as 'cytosolic' proteins. **b**, Volcano plots depicting the classification of proteins based on detected MS spectra intensities of oxidative modifications for 17 amino acids. **c**, Comparison of protein oxidation between membrane-specific and cytosolic proteins, focusing on amino acids not highlighted in the main figures.

Q 3-3.

Author mentioned that BTP is able to penetrate the plasma membrane. While from Extended Data Fig 2c, it seems BTP (green fluorescence) can partially localize in the plasma membrane. I think this can be well answered by performing a co-localization experiment between BTP and commercially available plasma membrane dyes.

Response 3-3:

We appreciate your thorough comment. We have conducted an additional experiment to determine how BTP molecules penetrate plasma membranes. We have examined BTP uptake under cellular conditions at 37 °C, at 4 °C, and in the presence of NaN₃. The results suggest that the BTP does not penetrate the plasma membrane efficiently under conditions of 4 °C and

in the presence of NaN₃. That means that the penetration mechanism of BTP into cells depends on energy-consuming processes rather than passive diffusion (*Nat. Chem.* **2022** 14, 274-283). These experimental results have been added to the revised manuscript (Extended Data Fig. 2a).

Revised:

Given the molecular structure of BTP which is composed of several lipophilic aromatic rings and a hydrophilic carboxylic acid, the passive diffusion of BTP across plasma membranes can be limited. To explore the cellular uptake of BTP, we investigated its uptake under physiological conditions at 37 °C, at 4 °C, and in the presence of NaN₃. The confocal microscopy results show that the uptake of BTP by HeLa cells dramatically decreased under conditions of 4 °C and NaN₃, implying that the penetration mechanism of BTP across plasma membranes relies on energy-consuming processes (Extended Data Fig. 2a).

As you suggested, we have investigated the colocalization of BTP with a membrane localised dye (CellMask™ Deep Red Plasma Membrane Stain). Interestingly, BTP was colocalised with the plasma membrane partially after photocatalysis, similar to its redistribution to mitochondria. Given these results, BTP appears to be located in the plasma membrane in the SIM images, because the strong SIM laser likely induced photoactivation of BTP. We have added these results to the Extended Data Fig. 2.

Revised:

We next examined where BTP is located and oxidative stress is produced in cells. Co-localisation experiments revealed that BTP was in Golgi apparatus (GA) and endoplasmic reticulum (ER), but not in mitochondria (Extended Data Fig. 2b, Supplementary Fig. 9). However, BTP photocatalysis changed the localisation pattern from the ER to the mitochondria and plasma membrane (Extended Data Fig. 2c–e), with a notable relocation of BTP to the mitochondrial membranes, as confirmed by structured illumination microscopy (SIM) with viable HeLa cells (Extended Data Fig. 2d). This change in location is likely because BTP photocatalysis reduces the ER integrity²⁸, leading to its migration to nearby membranes.

Extended Data Fig. 2 | Co-localisation imaging. **a**, BTP uptake under physiological conditions at 37 °C, at 4 °C, and in the presence of NaN₃. HeLa cells were pre-incubated at the conditions for 30 minutes, then further incubated with BTP (10 μM) for 2 hours. **b**, Co-localisation of BTP (5 μM). HeLa cells were transfected with CellLight™ Golgi RFP BacMam2.0 and Sec61b-mGFP constructs to stain the Golgi apparatus and endoplasmic reticulum (ER), respectively. MitoTracker™ Deep Red FM was used to stain mitochondria. Fluorescence signals of GolgiRFP, Sec61b-mGFP, and MitoTracker are represented in red. **c**, Changes in location of BTP after photocatalysis. HeLa cells were stained with BTP and MitoTracker and imaged before and after light exposure (confocal laser, λ = 445 nm). Enlarged images of white boxes show that BTP moves to the mitochondrial membrane during photocatalysis. **d**, Live-structured illumination microscopy (live-SIM) images before and after BTP photocatalysis (λ = 445 nm). The green and red signals correspond to BTP and MitoTracker, respectively. **e**, Co-localisation of BTP with a plasma membrane staining dye. HeLa cells were stained with BTP (10 μM for 2 hours) and CellMask™ Deep Red Plasma Membrane Stain (5 μg/mL for 10 minutes). The HeLa cells were imaged before and after light exposure (confocal laser, λ = 445 nm). Live HeLa cells were used for all co-localisation imaging. Data are presented as mean ± s.d. Source data are provided as a Source Data file.

Q 3-4.

All the co-localization imaging and intracellular ROS generation experiments were investigated by using HeLa cells. Did the GA and ER targeting and following Mito transfer or ROS generation were cell dependent? Please give the explain by experimental results.

Response 3-4:

We appreciate the suggestion and have added the colocalisation images and intracellular ROS generation in other cell lines (A549 and PANC-1). We have confirmed that the localisation of BTP to the GA and ER, its redistribution to mitochondria, and ROS generation were similarly investigated in other cell lines. We have added these results to the Supplementary Fig. 9 and 10.

Revised:

Supplementary Fig. 9. Co-localisation of BTP (5 μM). A549 and PANC-1 cells were transfected with CellLight™ Golgi RFP BacMam2.0 and Sec61b-mGFP constructs to stain the Golgi apparatus and endoplasmic reticulum (ER), respectively. Additionally, MitoTracker™ Deep Red FM was used to stain mitochondria. The fluorescence signals of Golgi, ER, and MitoTracker are represented in red. Furthermore, re-localisation of BTP to mitochondria after photocatalysis was also observed. light exposure: confocal laser, $\lambda = 445$ nm.

Supplementary Fig. 10. H₂DCF-DA assay for intracellular ROS generation. A549 and PANC-1 cells were incubated with BTP (5 μM) and H₂DCF-DA (20 μM), followed by irradiation with blue LED light ($\lambda_{\text{max}} = 450$ nm, 5 J·cm⁻²). The green signal corresponds to DCF fluorescence.

Q 3-5.

Similar with Q4, the morphological features of pyroptotic cells and the cleavage of capase-4/5 in Fig. 5 were investigated by only using HeLa cells. Did the pyroptosis-mediated cell death were cell dependent? Please give the explain by experimental results.

Response 3-5:

We have performed the same experiment with A549 and PANC-1 cell line, and we have added the results to the revised manuscript. Given the experimental results from iBMDM and HeLa cell lines, we now expect that BTP photocatalysis triggers non-canonical pyroptosis in various cell lines in a similar manner. We have added these results to the Supplementary Figure 12 and 14.

Revised:

Supplementary Fig. 12. Pyroptotic morphology changes of A549 and PANC-1 cells in response to photocatalytic membrane oxidation. The cells were incubated with BTP (10 μM) for 2 hours. light exposure: confocal laser, $\lambda = 445 \text{ nm}$.

Supplementary Fig. 14. Western blot analysis of A549 and PANC-1 cells with BTP photocatalysis for investigating caspase-4/5 cleavage.

Reviewer #4 (Remarks to the Author):

In this work, Lee and co-workers report an oxidative photocatalyst that can localize cellular membranes and generate hydroxyl radicals upon blue light irradiation under hypoxia. The authors found that light-triggered oxidative stress destabilizes membrane proteins, which consequently induces gasdermin-D-driven pyroptosis. The redox activity of photocatalyst, the mechanism of hydroxyl radical generation, and pyroptotic cell death are well-characterized and very convincing. However, it has not been demonstrated what benefits or new insights can be brought into biological research by this approach, other than artificially inducing cell death. There is also concern that the organelle membrane selectivity of this photocatalyst seems too low to study local oxidative stress in cells.

Overall, in its current form, I think that the work would not be impressive to many biological readers and does not meet the potential significance and impact required for Nature Communications. It is suggested that the manuscript could be reconsidered to submit other journals after the following points have been addressed:

Q 4-1.

The authors claim that BTP localizes in intracellular membrane (endomembranes). However, this description is not appropriate. Extended Data Fig.2c clearly showed that BTP is also present in the plasma membrane. As the authors mentioned, BTP also translocates to mitochondrial membranes during incubation. Therefore, it is clear that BTP shows virtually no membrane selectivity. The low membrane selectivity is also suggested by proteome analysis.

Response 4-1:

We appreciate your thorough and balanced review, and we apologize for not providing an explanation more carefully. First, the re-distribution of BTP to the mitochondrial membrane does not occur during incubation, but rather after photo-irradiation. For the plasma membrane, we have investigated BTP localisation. Similar to the mitochondria, we have confirmed that BTP was not colocalised with the plasma membrane before photoirradiation (Extended Data Fig. 2e). In the original manuscript, the observed plasma membrane localisation of BTP in the SIM images was likely caused by the strong SIM laser. Furthermore, previously published papers have suggested that Ca^{2+} release from triggers the formation of ER-plasma membrane contact site (van Vilet et al., *Molecular Cell* **2017**, 65, 885-899; Strzyz, *Nat. Rev. Mol. Cell Biol.* **2017**, 18, 213; Zaman et al., *Front. Cell Dev. Biol.*, **2020**, 8, 675). Thus, the BTP redistribution to plasma membrane after photoirradiation is likely due to the ER-plasma membrane contact considering that our experiments suggested a rapid Ca^{2+} mobilisation (Extended Data Fig. 7).

Nevertheless, we agree with your concern and understand that the O-Met analysis and other experiments did not effectively prove membrane-focused oxidative stress. Therefore, we have performed additional experiments and analyses to support our conclusion that BTP

photocatalysis induces membrane-focused oxidative stress and membrane protein-specific oxidation.

First, we have verified BTP molecular proximity to the membrane by comparing its emission peak in cellular and artificial membrane environments. The emission peak of BTP significantly responds to the hydrophobicity of its surroundings. In a polar environment like water, BTP's emission peak occurs at 630 nm, whereas within a non-polar environment, such as the membrane, it presents a peak at 580 nm, as shown in Extended Data Fig. 3a. Upon introducing artificial lipid bilayers (bicelles) to the BTP aqueous solution, the emission peak shifted from 630 nm to 580 nm (Extended Data Fig. 3b), indicating that BTP is close to hydrophobic lipid bilayers. Subsequently, we measured the emission peak of BTP in cellular environments using Lambda-scan analysis, appearing at 580 nm (Extended Data Fig. 3c). Based on these observations, we infer that BTP resides in close proximity to the intracellular membrane. These results are included in the Extended Data Figure below.

Revised:

We hypothesized that BTP can be especially proximal to intracellular membranes, including the membranes of ER, GA, and mitochondria, considering the amphiphilic BTP structure. To confirm the proximity of BTP to intracellular membranes, we investigated the emission peak shift of membrane-localised BTP (Extended Data Fig. 3a). In a situation where BTP is dissolved in water, the polar environment around the BTP exciton can stabilise the excited electron, reducing the fluorescence energy. However, the nonpolar environment around bio-membranes destabilises the excited electron of BTP, enhancing the fluorescence energy (Extended Data Fig. 3a). As a results, we confirmed that the wavelength of the emission maximum peak shifts from 630 nm to 580 nm when bicelles, which are artificial lipid bilayers, are added to the BTP aqueous solution (Extended Data Fig. 3b). Notably, Lambda-scan analysis in cell environments exhibited that intracellular BTP showed a similar tendency ($\lambda_{em} = 580$ nm; Extended Data Fig. 3b, c) to the environment with bicelles, indicating that BTP molecules are proximal to intracellular membranes. Therefore, these results imply that BTP photocatalysis inside cells generates reactive radicals nearby intracellular membranes, leading to intracellular membrane-focused oxidative stress.

Extended Data Fig. 3 | Intracellular membrane localisation of BTP and intracellular ROS generation. **a**, A schematic image of BTP fluorescence depending on surrounding environments. **b**, Normalised BTP fluorescence in different conditions. Blue: BTP in 50 mM Tris buffer (pH 7.5, [BTP] = 20 μ M), orange: BTP in 50 mM Tris buffer (pH 7.5, [BTP] = 20 μ M) + bicelles, red: intracellular environment (HeLa cells). The fluorescence of BTP inside cells were obtained by the Lambda-scan mode of confocal microscopy. **c**, Confocal images of BTP were taken at each wavelength, and merged was produced. **d**, $\text{H}_2\text{DCF-DA}$ assay for intracellular ROS generation. HeLa cells were incubated with BTP (5 μ M) and $\text{H}_2\text{DCF-DA}$ (20 μ M) and irradiated with blue LED light ($\lambda_{\text{max}} = 450 \text{ nm}$, 5 $\text{J}\cdot\text{cm}^{-2}$). The green signal corresponds to the DCF fluorescence. **e**, Dot plot analysis of DCF signals from randomly selected 20 cells (BTP+/hv+ and BTP+/hv-) ($n = 20$). **f**, $\text{O}_2^{\cdot -}$ assay using dihydroethidium (DHE). HeLa cells were treated with DHE (5 μ M) and exposed to BTP photocatalysis ($\lambda_{\text{max}} = 450 \text{ nm}$, 10 $\text{J}\cdot\text{cm}^{-2}$). Red signals correspond to DNA-intercalated 2-hydroxyethidium. Live HeLa cells were used for all ROS generation assays. Data are presented as mean \pm s.d. Source data are provided as a Source Data file.

Second, we have re-analysed LC-MS/MS raw data for 17 amino acids, excluding Gly, Ser, and Thr, as referenced by fast photochemical oxidation of protein (FPOP) footprinting. Methionine is the most labile amino acid under oxidative stress, which implies that proteomics based solely on methionine oxidation could contain a significant amount of noise. Therefore, additional information on protein oxidation is required for more accurate identification of oxidised proteins. Accordingly, we have applied a multi-stage search strategy using the modification search tool MODplus (Na et al. *Anal. Chem.* **2019**, 91, 17, 11324). Only the spectra unidentified in the first search were re-examined against the identified protein list from the first search (6,889 entries). This method allowed us to exclude non-specific signals derived from labile O-Met (+16 Da). To further attenuate undesired O-Met noise, mono-oxidation of methionine newly searched by MODplus was also excluded.

Using these changed methods, we obtained results that the protein oxidations were membrane-specific with far less variance than O-Met only analysis. The number of identified oxidised membrane proteins were much more than that of oxidised cytosolic proteins, even through more conservative threshold of the oxidised protein ($p < 0.01$ and Fold change > 4). Moreover, the degree of oxidation of membrane proteins was much higher for most amino acids. Based on these results, we have revised the Figures and corresponding paragraphs as follows.

Comprehensively, these results indicate that BTP photocatalysis triggers intracellular membrane-specific oxidative stress.

Revised:

Impact of BTP photocatalysis on protein quality control proposed by proteomics

Since methionine is one of the most labile amino acids under oxidative stress, proteins containing oxidised methionine residues (O-Met) were analysed in HeLa cells using label-free quantitative mass spectrometry for an initial screening of BTP oxidation targets (Fig. 3a)^{27,39,40}. The extent of oxidative damage was evaluated for each protein by comparing the average O-Met mass spectra intensities of the experimental groups with those of the control group (Fig. 3a, inset). Proteins that were oxidised more than 2-fold in the experimental group compared to the control group, with p-values lower than 0.05, were considered as 'oxidised proteins' by BTP photocatalysis (p-value < 0.05 , Fold Change > 2). The identified proteins were categorised based on their GO annotation by cellular location, determining whether they were membrane-localised or not (Fig. 3b, Extended Data Fig. 6a). Proteins annotated as being localised to plasma, organelles, or various other membranes were classified as 'membrane-specific'. Cytosolic proteins, excluding the membrane-cytosol overlying proteins, were then compared with the membrane-specific proteins (Fig. 3c). A greater number and proportion of oxidised proteins were observed in membrane-specific proteins compared to cytosolic proteins: 339 versus 40, accounting for 24.5% and 3.6%, respectively. This result supports the membrane-focused oxidative stress, which matches with the membrane-localisation property of BTP. Additionally, the proportions of oxidised membrane proteins of the mitochondria, ER, nucleus, and Golgi apparatus were 31.1%, 19.1%, 21.1%, and 7.7%, respectively (the number of oxidised membrane proteins/the detected number of membrane proteins, Fig. 3d). The global membrane oxidation induced by BTP photocatalysis suggests a potential malfunction in biological processes that require the involvement of various organelles.

To further elucidate other oxidation targets of BTP photocatalysis at the membrane, an in-depth secondary search covering 17 amino acids was conducted using mass spectra not identified in the initial search for O-Met analysis (Fig. 3a). This multistage search allowed us to scrutinise extensive protein oxidations by reducing the search space and thus decreasing the number of false positives. Interestingly, membrane-specific proteins were the most oxidised (oxidised protein criteria: p-value < 0.01 , Fold Change > 4), followed by membrane-cytosolic proteins, and then cytosolic proteins for all detected amino acid residues (Fig 3e, f; Extended Data Fig. 6b, c). It is noteworthy that the number of oxidised proteins, as defined by Trp and His oxidation, was prevalent in membrane-specific proteins (Fig. 3f, Extended Data Fig. 6c). Considering their low abundance in a whole cell, it suggests a favourable interaction between BTP and oxidisable-aromatic amino acids. Detailed information on the oxidised proteome for each amino acid is available in 'Source Data Proteomics' spreadsheet.

Fig. 3 | Comprehensive proteomic profiling of membrane protein oxidation induced by BTP photocatalysis.

a, Schematic illustration of the proteomic analysis workflow used to investigate the extent of oxidative modifications induced by BTP photocatalysis. The process involves a multistage search strategy for identifying O-Met and FPOP (fast photochemical oxidation of proteins) modifications. In the first search, 2,120,870 peptide-spectrum matches (PSMs) were found, and in the second search, 1,959,844 PSMs were found. Samples subjected to BTP photocatalysis and control samples were analysed and compared based on the fold change in the average precursor intensity of oxidative modifications (inset). **b**, Proteins categorized by GO subcellular annotations into 'membrane-specific', located exclusively on membranes, and 'membrane-cytosol', found on both membranes and cytosol. The remaining cytosolic proteins were labelled 'cytosolic' proteins. **c**, Volcano plot of O-Met (+16 Da) proteome showcasing oxidation focused on membrane-specific proteins versus cytosolic proteins. Proteins with a p-value < 0.05 and fold change > 2 were defined as potential oxidation targets of BTP photocatalysis. **d**, The proportions of oxidised membrane proteins across different organelles based on the O-Met proteome. 'Others' include plasma membranes and unidentified locations. **e**, Overview of the 2nd search based on FPOP modifications, showing average oxidation intensities for different amino acids. 'All AAs' represents the aggregated intensities of oxidative modifications of these 17 amino acids. The averaged oxidation intensities of 'membrane-specific', 'membrane-cytosolic', and 'cytosolic' proteins for the corresponding amino acids were presented to compare the degree of oxidation of membrane proteins and soluble proteins for each type of amino acid. The averaged oxidation intensities of three control conditions were normalised to 1. **f**, Volcano plots of the proteome other than O-Met, contrasting membrane-specific proteins with cytosolic proteins. Stricter criteria than sole O-Met analysis (p-value < 0.01 and fold change > 4) were applied for robust identification of oxidised proteins.

Extended Data Fig. 6 | Volcano plots for oxidative modifications of each amino acid. a, Classification of proteins based on Gene Ontology (GO) subcellular annotations: 'Membrane-specific' proteins are exclusively located on membranes, while 'membrane-cytosol' proteins are present on both membranes and in the cytosol. Proteins localized in the cytosol are referred to as 'cytosolic' proteins. **b,** Volcano plots depicting the classification of proteins based on detected MS spectra intensities of oxidative modifications for 17 amino acids. **c,** Comparison of protein oxidation between membrane-specific and cytosolic proteins, focusing on amino acids not highlighted in the main figures.

Revised Method:

Modification search to identify oxidized amino acids.

To comprehensively identify peptides including all oxidized amino acids, we employed a multi-stage search strategy where only the spectra unidentifiable in the first search were searched against the proteins (6,889 entries) identified in the first search using a modification search tool, MODplus (v1.02)⁵⁸. The search parameters were as follows: precursor mass tolerance = ± 20 ppm, ¹³C errors in precursor mass = -1/0/1/2, fragment mass tolerance = ± 20 ppm, enzyme = trypsin, the number of enzymatic termini = 1, the number of missed cleavages = any, fixed modifications = Carbamidomethyl of cysteine, variable modifications = MS-common modifications provided by MODplus and fast photochemical oxidation of proteins (FPOP)-related modifications (Supplementary Table 1)^{59,60}, the number of modifications/peptide = any within the modified mass range of -150 to +350 Da, decoy search = 1. All identifications were subsequently rescored by Percolator (v3.06)⁶¹ and validated at an estimated FDR of 1%, resulting in a total of 1,959,844 identifications.

To quantify oxidized amino acids in proteins, we extracted peptides including FPOP-related modifications and aggregated their precursor intensities for each corresponding protein. To evaluate the extent of oxidative modifications excluding methionine oxidation (+15.995), the methionine oxidations were excluded from the quantification (methionine di-oxidations were included).

As a result, we obtained the precursor intensities for each protein and for oxidative modification of 17 amino acids (see Supplementary Table 1). Additionally, we represented 'All AAs' by aggregating intensities of oxidative modifications of these 17 amino acids. To compare the degree of oxidation of membrane proteins and soluble proteins for each type of amino acid, we presented the averaged oxidation intensities of 'membrane-specific', 'membrane-cytosolic', and 'cytosolic' proteins for the corresponding amino acids (Fig. 3e). The averaged oxidation intensities of three control conditions were normalised to 1. The fold change values were calculated in the same way as described above.

Q 4-2.

Fig. 3: It is unclear how to calculate fold change for quantitative mass spectrometry. Are the authors quantifying only peptides containing oxidized Met or all peptides to calculate protein abundance? How many peptides per protein are used to calculate abundance? According to Fig.2, the amount of membrane protein should decrease after photooxidation. Why does fold change increase for many proteins?

Response 4-2:

We apologize for the misleading description regarding the quantification and evaluation of oxidised proteins. In the Fig. 3 legend of the original manuscript, we stated 'Fold Change, oxidised spectrum count change after BTP photocatalysis', which was incorrect. We calculated the Fold change values based on 'spectrum intensity', not on 'spectrum count'. Therefore, we have revised the figures (Fig. 3 and Extended Data Fig. 6) and methods in the text.

Revised method:

As a result, we obtained mass intensities of oxidative modifications for each identified protein. Using these intensities from triplicated control conditions and an experimental condition, we calculated the p-values and Fold change values. The fold change values for each protein were calculated as 'the average mass intensities of oxidative modifications in the experimental condition / the average mass intensities of oxidative modifications in the control conditions' as follows.

$$\text{Fold change} = \frac{I_{\text{experimental}}}{I_{\text{controls}}}, \text{ where } I = \text{Averaged mass intensities of oxidative modifications.}$$

For this research, only peptides that contained oxidative modifications were calculated: O-Met for the 1st search and multiple FPOP-related modifications for the 2nd search, respectively. As you implied, to define 'the amount of proteins after oxidation' or 'the percentile of protein oxidation versus unoxidized proteins', it would be ideal to consider all peptides and compare them on a peptide-by-peptide basis. However, we used gathered intensity (spectrum intensities of peptides) because the aim of our proteomic analysis was to identify which oxidized protein were abundant and, therefore, more likely to perturb biological processes. Detected intensities of peptides (top 3 precursor intensity, exported from Scaffold software) were aggregated for

each protein. Subsequently, these aggregated intensities were compared across four distinct conditions: with or without BTP and photoexcitation (revised Fig. 3a). The average intensity from the triplicate experiments was used for each of the four conditions. It is important to note that strong oxidation in our experiments occurs exclusively through BTP photocatalysis. Therefore, we designated the condition with both BTP and photoexcitation as the experimental group, while the other three conditions, which do not facilitate photocatalysis, served as control groups. Finally, as described above, the fold change for each protein was calculated using the experimental group as the numerator and the control groups as the denominator.

This calculation reflects the fold change in the extent of oxidative modifications of the protein induced by BTP photocatalysis, not the total existing number of proteins. This explains why the fold change of many membrane proteins increased after photocatalysis. The amount of oxidized proteins (*i.e.*, the intensity of oxidatively modified peptides) increased due to the strong oxidative stress induced by BTP, resulting in an increased fold change. Fig. 2 demonstrates the oxidative structural destabilization of a protein, which refers to the decline of 'intact' proteins. However, the extent of oxidative modifications increases. We have revised Fig. 3 and Method section about proteomic data processing as follows.

Revised:

All MS/MS samples were analysed using Sequest Sorcerer platform (Sagen-N Research, San Jose, CA, USA). Sequest was set to search for Homo sapiens (20612 entries, UniProt (<http://www.uniprot.org>)), which includes frequently observed contaminants assuming the action of digestion enzyme trypsin. Sequest was searched with a fragment ion mass tolerance of 1.00 Da and parent ion tolerance of 10.0 PPM. The carbamidomethyl of cysteine was specified as a fixed modification in Sequest. Oxidation of methionine and acetyl at the N-terminus were specified as variable modifications in Sequest. Scaffold Q+ (version 5.1.0, Proteome Software Inc., Portland, OR) was used to validate MS/MS based peptide and protein identification. A peptide with a probability of higher than 99% for achieving an FDR of lower than 1.0% based on the no Scaffold Local FDR algorithm was accepted as true identification. A protein identification with a probability of higher than 14.0% for achieving an FDR of less than 1.0% and containing two or more identified peptides was accepted. Protein Prophet algorithm⁵⁷ was used to calculate the protein probabilities. Proteins that contained similar peptides and could not be differentiated by MS/MS analysis alone were grouped to satisfy the principles of parsimony. The GO annotations for the proteins were retrieved from the NCBI database (downloaded on 11 February 2021). Of the 5173389 spectra in the experiment at the given thresholds, 2120870 (41%) were included in the quantification. The top 3 precursor intensity of peptides aggregated for each protein from the proteomic data was used for label free quantification. The values were log₂-transformed, pruned of those matched to multiple proteins, and non-reproducibly detected values were filled by imputed values representing a normal distribution around the detection limit. A new distribution was created by a Gaussian distribution with a downshift of 1.8 and width of 0.3 standard deviations. All processes were conducted using the Perseus software platform of Max Planck Institute of Biochemistry. As a result, we obtained mass intensities of oxidative modifications for each identified protein. Using these intensities from triplicated control conditions and an experimental condition, we calculated the p-values and Fold change values. The fold change values for each protein were calculated as 'the average mass intensities of oxidative modifications in the experimental condition / the average mass intensities of oxidative modifications in the control conditions' as follows.

$$\text{Fold change} = \frac{I_{\text{experimental}}}{I_{\text{controls}}}, \text{ where } I = \text{Averaged mass intensities of oxidative modifications.}$$

Modification search to identify oxidized amino acids.

To comprehensively identify peptides including all oxidized amino acids, we employed a multi-stage search strategy where only the spectra unidentified in the first search were searched against the proteins (6,889 entries)

identified in the first search using a modification search tool, MODplus (v1.02)⁵⁸. The search parameters were as follows: precursor mass tolerance = ± 20 ppm, ^{13}C errors in precursor mass = $-1/0/1/2$, fragment mass tolerance = ± 20 ppm, enzyme = trypsin, the number of enzymatic termini = 1, the number of missed cleavages = any, fixed modifications = Carbamidomethyl of cysteine, variable modifications = MS-common modifications provided by MODplus and fast photochemical oxidation of proteins (FPOP)-related modifications (Supplementary Table 1)^{59,60}, the number of modifications/peptide = any within the modified mass range of -150 to $+350$ Da, decoy search = 1. All identifications were subsequently rescored by Percolator (v3.06)⁶¹ and validated at an estimated FDR of 1%, resulting in a total of 1,959,844 identifications.

To quantify oxidized amino acids in proteins, we extracted peptides including FPOP-related modifications and aggregated their precursor intensities for each corresponding protein. To evaluate the extent of oxidative modifications excluding methionine oxidation ($+15.995$), the methionine oxidations were excluded from the quantification (methionine di-oxidations were included).

As a result, we obtained the precursor intensities for each protein and for oxidative modification of 17 amino acids (see Supplementary Table 1). Additionally, we represented 'All AAs' by aggregating intensities of oxidative modifications of these 17 amino acids. To compare the degree of oxidation of membrane proteins and soluble proteins for each type of amino acid, we presented the averaged oxidation intensities of 'membrane-specific', 'membrane-cytosolic', and 'cytosolic' proteins for the corresponding amino acids (Fig. 3e). The averaged oxidation intensities of three control conditions were normalised to 1. The fold change values were calculated in the same way as described above.

Q 4-3.

Related to the comment 2, fold change should vary greatly depending on light exposure time. Have the authors performed experiments with different light exposure times? How were the light exposure times determined?

Response 4-3:

We have conducted a methionine sulfoxide (O-Met) immunoblot using a methionine sulfoxide immunoblotting kit (Cayman, Item No. 600160) to generally assess the extent of oxidation over different irradiation times. Whole HeLa cell lysates were collected to measure the intensity of the O-Met antibody band after 5, 10, and 20 min of irradiation with 450 nm LED ($16.7 \text{ mW}\cdot\text{cm}^{-2}$), corresponding to 5, 10, and 20 $\text{J}\cdot\text{cm}^{-2}$ respectively. The SDS gel was run for a short duration (50V, 1h 40min), as in proteomic sampling. Fig. R4 shows that the O-Met intensity reached saturation at 10 min of photoirradiation. Although 20 min of irradiation may induce a greater extent of oxidation beyond O-Met, 10 min was chosen to minimise undesired phototoxicity and protein loss, which can be caused by rapid pyroptotic membrane rupture (10+ min; *Cell Death Differ.* **2019**, 26, 146)

Fig. R4 | Western blot analysis using a methionine sulfoxide immunoblotting assay at different photoirradiation time points. HeLa cells were incubated with 4 μ M BTP and then irradiated with a 450 nm LED, followed by media exchange with fresh DPBS. 20 μ g of the cell lysate was loaded into each well.

Q 4-4.

The Fig.3 caption states that a total of 314 proteins were detected, but the main text says 313.

Response 4-4:

We sincerely apologise for the inappropriate caption. We have revised our proteomic analysis with comprehensive updates, including analysis for multiple modifications of 17 amino acids.

Q 4-5.

The authors must demonstrate what kind of new biological insights would be obtained by inducing pyroptosis in a light-dependent manner.

Response 4-5:

We apologise for the lack of demonstration regarding new biological insights and the significance of this study. The goal of this study is to suggest that ‘oxidative stress on membranes can be a trigger for pyroptosis’. Recent studies have focused on various triggers for pyroptosis, as it is a cell death pathway recognised for generating robust immune responses (*Immunol. Rev.* **2015**, 265, 130-142; *Nature* **2020**, 579, 421-426; *Cell. Death Discov.* **2022**, 8, 338; *Nature* **2017**, 547, 99-103). In most cases, non-canonical pyroptosis is known to be induced by intracellular lipopolysaccharide (LPS). However, our findings suggest that membrane oxidation can trigger non-canonical pyroptosis through an LPS-independent mechanism. This is a new pathway that can induce proptosis without pathogenic molecules. Given that the oxidation level in aged cells is much higher than in young cells, we believe that this study can contribute to the investigation of the pathogenesis of age-related diseases.

Furthermore, inducing pyroptosis in a light-dependent manner can be useful for the spatiotemporal control of cancer cell pyroptosis. Additionally, as noted by reviewer #5, BTP photocatalysis generates oxidative stress even in hypoxic environments, implying that this

photocatalysis could be therapeutically attractive, considering that most cancers have a hypoxic environment.

We have included this demonstration in the conclusion section as follows.

Revised:

We propose that photocatalytic membrane oxidation triggers non-canonical pyroptosis using the amphiphilic organic photocatalyst, BTP (Extended Data Fig. 9). Via photocatalysis, BTP generates highly oxidising $\bullet\text{OH}$ in a spatiotemporally controlled manner even under hypoxic conditions, thereby damaging the structural stability of membrane proteins. The single-molecule tweezer approach verified that BTP photocatalysis disrupts membrane protein folding. Using the oxidised proteome from the label-free quantification, we found that BTP photocatalysis substantially oxidised PQC-related membrane proteins of ER, GA, and mitochondria in cells. Disruption of the folding stability and oxidation of PQC-related proteins seemed to stimulate the accumulation of misfolded proteins, followed by ER stress, maladaptive UPR, and cation mobilisation. These cellular responses consequently triggered the caspase-4/5-induced GSDMD cleavage and subsequent pyroptosis.

Since pyroptosis is known to generate the most robust immune response, recent studies have focused on various triggers for this cell death pathway. Pyroptosis is usually caused by microbial infection or endotoxins such as LPS. However, we suggest that the intracellular membrane-focused oxidative stress can trigger pyroptosis through non-canonical inflammasome activation. This endotoxin-independent mechanism implies an alternative pathway for inducing pyroptosis. Although the full spectrum of biological processes and their causal relationships remain unexplored in this study, we believe that this study can inspire further research into the pathogenesis of immune-related diseases. Additionally, light-controlled pyroptosis can be useful to induce immune responses spatiotemporally. In particular, BTP photocatalysis induces pyroptosis even in a hypoxic environment, suggesting that this strategy can be therapeutically attractive, considering that most cancers have a hypoxic environment. Consequently, we hope that this method can be widely used to spatiotemporally induce caspase-4/5 activation and pyroptosis in pathogenesis studies and clinical applications.

Reviewer #5 (Remarks to the Author):

Lee, Park et al. have devised a clever strategy to cause membrane protein oxidation to trigger non-canonical regulated cell death via caspase 4/5 activation. This could be clinically useful to stop cancer cell proliferation or inflammation driven diseases. The strategy involves photocatalysis via a novel fatty acid like molecule. The strategy is interesting and the fact that it works under hypoxic conditions does make it attractive therapeutically. Could also be an experimental tool to study oxidative stress induced signaling and immune response. The study offers sufficient evidence that BTP photocatalysis works to induce cell death in cell models but offers little in new findings of downstream signaling post increases in oxidative stress. Caspase4/5 activation to GSDMD cleavage is well studied cell death pathway. However, the organic BTP photocatalyst could be a powerful experimental tool and of potential clinical significance and warrants publication. Also, the targeted approach to specifically oxidize membrane proteins is an exciting approach that warrants further study.

Major Points

Q 5-1.

Figure 4 – Not sure I see evidence of mitochondrial dysfunction (line 214-215 main text). Figure 4a nor 4e is that convincing that mitochondrial dysfunction has occurred. Not sure it really needs to be shown or stated that mitochondrial dysfunction is occurring since BTP photocatalysis induced cell death is evident in other figures of the manuscript. If authors want to prove mitochondrial dysfunction is occurring then the authors should extend the mitotracker experiment (>50s) to show that the calcium overload causes mitochondrial fragmentation or use a membrane potential sensitive dye to show mitochondrial membrane potential dissipates. Could also immunostain for cytochrome c release. At 50s the mitochondrial network in the HeLa cells looks healthy (4A) even though calcium is being taken up by the mitochondria. Seahorse mitochondrial respiration experiment is also a possibility.

Response 5-1:

We are sincerely grateful for your constructive and insightful comments. As you suggested, we have investigated mitochondrial dysfunction using a membrane potential-sensitive dye (TMRE). As a result, we have confirmed that TMRE fluorescence was completely eliminated after BTP photocatalysis, implying that the mitochondrial membranes were depolarised. Therefore, we now support the conclusion that BTP photocatalysis triggers mitochondrial damage. We have added the TMRE assay results to the revised manuscript as follows (Extended Data Fig. 7f).

Revised:

Accordingly, we conducted mitochondrial membrane potential assay using TMRE staining. After BTP photocatalysis, the TMRE fluorescence is dramatically diminished, implying that the mitochondrial membrane potential and functions are damaged after BTP photocatalysis (Extended Data Fig. 7f).

Extended Data Fig. 7 | Ca^{2+} and K^{+} mobilisation by BTP photocatalysis. **a**, Mitochondrial Ca^{2+} assay performed using Rhod-2. HeLa cells were incubated with BTP (5 μM), MitoTracker™ Deep Red FM (0.5 μM), and Rhod-2 (3 μM). The fluorescence of MitoTracker (cyan) and Rhod-2 (red) was measured using time-series confocal microscopy ($t = 0\text{-}50$ s, 10 s interval) during light exposure ($\lambda = 445$ nm, 0.3 mW). The fluorescence of Rhod-2 was enhanced dramatically between 20 and 30 s, implying that Ca^{2+} mobilisation occurred at this time. Mitochondrial matrix swelling following Ca^{2+} uptake was also observed after BTP photocatalysis. **b**, Merged images of MitoTracker and Rhod-2 signals at $t = 0$ and 30 s. **c**, Line-cut analysis of white arrows in **b**. **d**, Flow cytometry for Ca^{2+} mobilisation. HeLa cells were treated with BTP and Rhod-2, and the Rhod-2 fluorescence of each cell was measured before ($h\nu^{-}$) and 2 h after ($h\nu^{+}$) light exposure ($\lambda_{\text{max}} = 450$ nm, $10 \text{ J}\cdot\text{cm}^{-2}$). **e**, Live-SIM images of ER (red) and mitochondria (cyan) after BTP photocatalysis ($\lambda = 488$ nm, 10 mW). Mitochondrial swelling (top), fission, and fusion (bottom) were observed in HeLa cells. Arrows indicate mitochondrial fission (yellow) and fusion (white). **f**, Mitochondrial membrane potential assay using tetramethylrhodamine, ethyl ester (TMRE). HeLa cells were incubated with BTP (10 μM) and TMRE (0.5 μM) and irradiated with blue LED light ($\lambda_{\text{max}} = 450$ nm, $10 \text{ J}\cdot\text{cm}^{-2}$). Dot plot analysis of TMRE signals from randomly selected cells (BTP+/ $h\nu^{+}$ and BTP-/ $h\nu^{-}$) ($n = 18$ and 21). **g**, Flow cytometry of K^{+} efflux in HeLa cells. Intracellular K^{+} was measured with ION K^{+} Green-2, an intracellular K^{+} sensor, before ($h\nu^{-}$) and 2 h after ($h\nu^{+}$) BTP photosensitisation ($\lambda_{\text{max}} = 450$ nm, $10 \text{ J}\cdot\text{cm}^{-2}$). Data are presented as mean \pm s.d. Source data are provided as a Source Data file.

Q 5-2.

Figure 7c- Authors should consider moving the cancer cell line data (7c) that demonstrates the effectiveness of BTP photocatalysis in a cancer cell line to the main figures. Perhaps end of figure 5 or if the authors have data showing caspase 4/5 induced GSDMD cleavage is the mechanism for decreased cancer cell viability then it could be a new figure.

Response 5-2:

This is a great suggestion. We have added a new main figure (Fig. 5) containing cell viability assays.

Revised:

Fig. 5 | BTP photocatalysis induced cytotoxicity. a, Live or dead assay with Calcein AM and propidium iodide (PI). HeLa cells with BTP photocatalysis were stained by Calcein AM and PI 24 h after light exposure ($\lambda_{\max} = 450 \text{ nm}$, $3.7 \text{ J}\cdot\text{cm}^{-2}$). **b,** MTT assay of HeLa cells with BTP photocatalysis ($\lambda_{\max} = 450 \text{ nm}$, $10 \text{ J}\cdot\text{cm}^{-2}$) ($n = 4$). **c.** MTT assays for normoxic/hypoxic pancreatic cancer cells (Panc-1 and MiaPaca-2). ($n = 4$). All data are presented as mean \pm s.d. * $P < 0.05$. Student's two-tailed t -test. Source data are provided as a Source Data file.

Minor points

Q 5-3.

Figure 3- Authors should make more clear in the legend and text what the log2 is comparing (i.e. hv- over hv+ in hela cells)?

Response 5-3:

We apologise for the lack of clarity regarding the Fold Change values in Fig. 3. We have revised the figure legends and text accordingly. The log₂ Fold Change compares the BTP+ hv+ group to the control conditions (BTP+, hv-; BTP-, hv+; BTP-, hv-). For a better understanding of the comparison, we have added Fig. 3 that illustrates the experimental conditions.

Revised:

Fig. 3 | Comprehensive proteomic profiling of membrane protein oxidation induced by BTP photocatalysis.

a, Schematic illustration of the proteomic analysis workflow used to investigate the extent of oxidative modifications induced by BTP photocatalysis. The process involves a multistage search strategy for identifying O-Met and FPOP (fast photochemical oxidation of proteins) modifications. In the first search, 2,120,870 peptide-spectrum matches (PSMs) were found, and in the second search, 1,959,844 PSMs were found. Samples subjected to BTP photocatalysis and control samples were analysed and compared based on the fold change in the average precursor intensity of oxidative modifications (inset). **b**, Proteins categorized by GO subcellular annotations into 'membrane-specific', located exclusively on membranes, and 'membrane-cytosol', found on both membranes and cytosol. The remaining cytosolic proteins were labelled 'cytosolic' proteins. **c**, Volcano plot of O-Met (+16 Da) proteome showcasing oxidation focused on membrane-specific proteins versus cytosolic proteins. Proteins with a p -value < 0.05 and fold change > 2 were defined as potential oxidation targets of BTP photocatalysis. **d**, The proportions of oxidised membrane proteins across different organelles based on the O-Met proteome. 'Others' include plasma membranes and unidentified locations. **e**, Overview of the 2nd search based on FPOP modifications, showing average oxidation intensities for different amino acids. 'All AAs' represents the aggregated intensities of oxidative modifications of these 17 amino acids. The averaged oxidation intensities of 'membrane-specific', 'membrane-cytosolic', and 'cytosolic' proteins for the corresponding amino acids were presented to compare the degree of oxidation of membrane proteins and soluble proteins for each type of amino acid. The averaged oxidation intensities of three control conditions were normalised to 1. **f**, Volcano plots of the proteome other than O-Met, contrasting membrane-specific proteins with cytosolic proteins. Stricter criteria than sole O-Met analysis (p -value < 0.01 and fold change > 4) were applied for robust identification of oxidised proteins.

Reviewers' Comments:

Reviewer #1:

Remarks to the Author:

The authors present an extensively revised manuscript that, in my opinion, is significantly stronger than the original submission. In particular, the conclusion that the photoinduced damage is indeed the cause of downstream responses is now backed up by more experimental evidence. My other concerns have also been addressed. I applaud the authors for the systematic and thorough way in which they address the reviewers' concerns, and I am impressed by the additional experiments and analyses presented in the revised manuscript. I think that the manuscript is well worthy of publication in Nature Communications.

Reviewer #2:

Remarks to the Author:

The authors have responded my previous concerns about proteomics.

Reviewer #3:

Remarks to the Author:

The authors have fully addressed reviewers' detailed comments. So, the current revised version can be acceptable for publication in Nat Comm as it is.

Reviewer #5:

Remarks to the Author:

The authors have adequately addressed my concerns in the revised manuscript